TOOLS

Reproducibility

# Quality assessment in light microscopy for routine use through simple tools and robust metrics

Orestis Faklaris[1], Leslie Bancel-Vallée[1], Aurélien Dauphin[2], Baptiste Monterroso[3], Perrine Frère[4], David Geny[5], Tudor Manoliu[6], Sylvain de Rossi[1], Fabrice P. Cordelières[7], Damien Schapman[8], Roland Nitschke[9], Julien Cau[1], and Thomas Guilbert[10]

**Although there is a need to demonstrate reproducibility in light microscopy acquisitions, the lack of standardized guidelines monitoring microscope health status over time has so far impaired the widespread use of quality control (QC) measurements. As scientists from 10 imaging core facilities who encounter various types of projects, we provide affordable hardware and open source software tools, rigorous protocols, and define reference values to assess QC metrics for the most common fluorescence light microscopy modalities. Seven protocols specify metrics on the microscope resolution, field illumination flatness, chromatic aberrations, illumination power stability, stage drift, positioning repeatability, and spatial-temporal noise of camera sensors. We designed the MetroloJ_QC ImageJ/Fiji Java plugin to incorporate the metrics and automate analysis. Measurements allow us to propose an extensive characterization of the QC procedures that can be used by any seasoned microscope user, from research biologists with a specialized interest in fluorescence light microscopy through to core facility staff, to ensure reproducible and quantifiable microscopy results.**

## Introduction

Quality control (QC) is often neglected in the field of light microscopy because it is considered complex, costly, and time-consuming (Nelson et al., 2021). It is, however, essential in research to ensure quantifiable results and reproducibility among laboratories (Baker, 2016; Nature Methods, 2018; Deagle et al., 2017; Heddleston et al., 2021). Biologists aiming at publishing top-quality images in light microscopy need to understand not only the fundamentals of microscopy but also the limitations, variations, and deviations of microscope performance.

In the past, different groups have developed measurement protocols and published methods for a variety of wide-field and confocal microscope performance aspects (Kedziora et al., 2011; Murray et al., 2007; Petrak and Waters, 2014; Zucker et al., 2007; Zucker and Price, 1999). The majority of these QC studies were recently reviewed (Jonkman et al., 2020; Jost and Waters, 2019; Montero Llopis et al., 2021). Some of the studies were carried out on an international scale involving a larger community in the frame of the Association of Biomolecular Resource Facilities (ABRF; Cole et al., 2013; Stack et al., 2011). In a few of these

studies, automation of the limit values and analysis workflows was proposed. The ISO 21073:2019 norm for confocal microscopy was recently published (ISO, 2019; Nelson et al., 2020), providing a fixed minimal set of tests to be performed. This norm is descriptive, no experimental values are shown, nor are limiting values proposed, but it is the first step toward standardization of QC in microscopy.

Due to a lack of commonly accepted QC guidelines, we made a significant advance by collecting experimental data from 10 light microscopy core facilities and developed tools, fast and robust acquisition protocols, and automated analysis methods that can render the monitoring of light microscope performance easily accessible to a broad scientific community.

We checked 117 objectives, 49 light sources, 25 stages, and 32 cameras, and additionally automated many of the acquisition and analysis procedures. Most of the major microscope manufacturers were represented, with the majority coming from Carl Zeiss Microscopy (50%), followed by Leica Microsystems (20%), Nikon (14%), Olympus/Evident (9%), Andor–Oxford Instruments (7%), and Till Photonics (2%). The facilities involved are

...........................................................................................................................................................................................................................
[1]Montpellier Ressources Imagerie, Biocampus, University of Montpellier, CNRS, INSERM, Montpellier, France;   [2]Unite Genetique et Biologie du Développement U934, PICT-IBiSA, Institut Curie, INSERM, CNRS, PSL Research University, Paris, France;   [3]Prism, Institut de Biologie Valrose, CNRS UMR 7277, INSERM 1091, University of Nice Sophia Antipolis – Parc Valrose, Nice, France;   [4]Plate-forme d'Imagerie de Tenon, UMR_S 1155, Hôpital Tenon, Paris, France;   [5]Institut de Psychiatrie Et Neurosciences de Paris, INSERM U1266, Paris, France;   [6]Gustave Roussy, Université Paris-Saclay, Plate-forme Imagerie et Cytométrie, UMS AMMICa. Villejuif, France;   [7]University of Bordeaux, CNRS, INSERM, Bordeaux Imaging Center, UMS 3420, US 4, Bordeaux, France;   [8]Université of Rouen Normandie, INSERM, Plate-Forme de Recherche en Imagerie Cellulaire de Normandie, Rouen, France;   [9]Life Imaging Center and Signalling Research Centres CIBSS and BIOSS, University Freiburg, Freiburg, Germany;   [10]Institut Cochin, INSERM (U1016), CNRS (UMR 8104), Universite de Paris (UMR-S1016), Paris, France.

Correspondence to Orestis Faklaris: orestis.faklaris@mri.cnrs.fr.

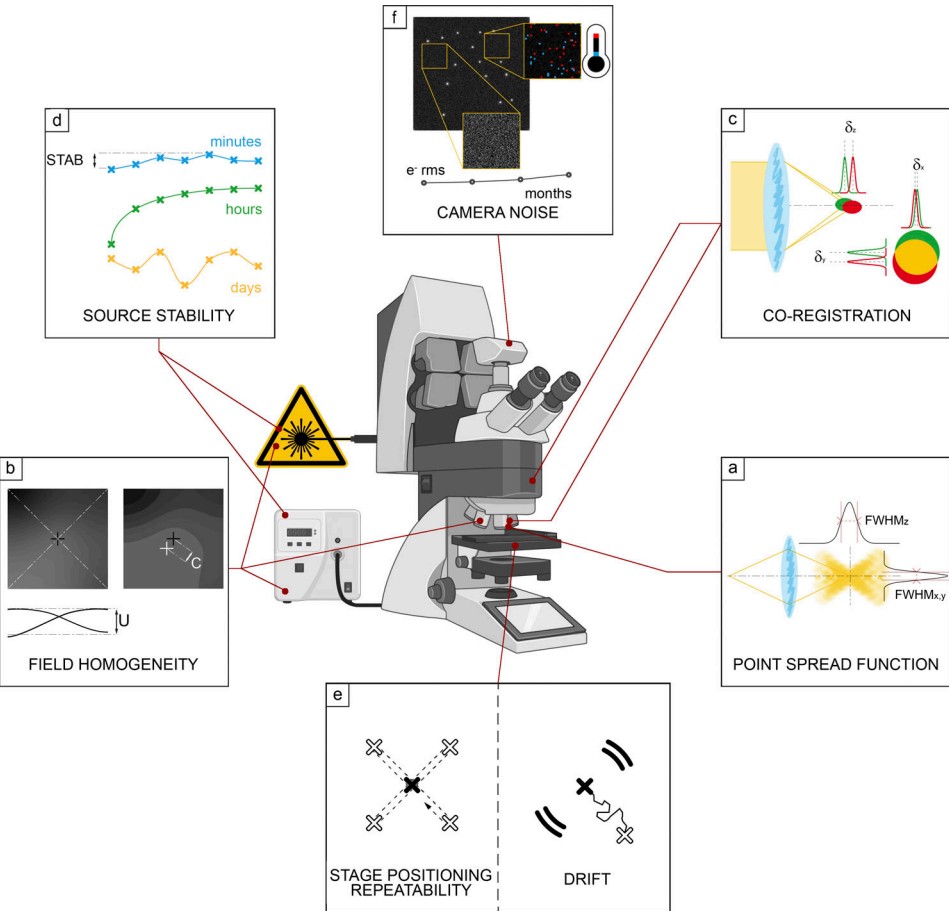

Figure 1. **Schematic overview of the seven proposed QC guideline fields, together with the corresponding MetroloJ_QC tool icon.** (a) PSF is a simple way to characterize the resolution power in the *x*, *y*, and *z* dimensions. (b) Field illumination determines the intensity quantification over the whole field of view of the system, tile-scan reconstructions, and inhomogeneous sample bleaching. (c) Coregistration characterizes the chromatic mismatch between different color channel images of the same fluorescent emitter in the *x*, *y*, and *z* directions. (d) Illumination power stability at different time scales determines the reproducibility and quantification of the experiment. (e) The imaging quality depends on the stability of the sample in *x*, *y*, and *z* characterized by drift monitoring and stage positioning repeatability. (f) The noise and offset characterization of the camera sensor influences quantification, especially for weak signals.

part of the French microscopy technological network RTmfm. This study results from collaborative work over the past 10 yr of the QC working group (GT3M; https://rtmfm.cnrs.fr/en/gt/gt-3m/) of this network.

We first employed a qualitative approach followed by specific QC metric calculations. The paper is divided into six chapters that define guidelines to characterize (1) lateral and axial resolution, (2) field illumination, (3) coregistration, (4) illumination power stability, (5) stage drift and positioning repeatability, and (6) temporal and spatial noise sources of camera sensors (Fig. 1). Each chapter begins with an introduction that serves as the chapter background in which key questions are presented. The main conclusions are given in the discussion part of every chapter, with examples emphasizing the influence of QC in biological imaging.

These six chapters will be of interest to all biologists willing to acquire meaningful and reliable images on a well-tuned microscope. More precisely, the metrics and tools that we propose will be greatly useful for (1) microscopy users with access to core facilities, (2) core facility staff taking care of the routine

operation of the microscopes, and (3) biologists with irregular seasonal usage of their microscope. A deeper knowledge of microscopy theory will help in reading the full article. Recent scientific reviews describe good practice guides, which are very useful to biologists interested in the fundamentals of fluorescence microscopy (Cuny et al., 2022; Swift and Colarusso, 2022).

We developed the MetroloJ_QC (ImageJ/Fiji Java plugin) software tool that fully automates the QC data analysis based on existing MetroloJ plugin tools (Cordelières and Matthews, 2010; Fig. 2). Processing was automated to minimize user actions. We also implemented multichannel image processing to avoid channel splitting and separately configuring the plugin for each color channel. The plugin incorporated metrics defined in this study, tools for camera characterization and drift monitoring, and permitted the creation of report documents to quickly identify parameters within/outside tolerances.

We highlighted the importance of QC measurements during new microscope setup installations and their crucial role in building up a requisite vocabulary to interact with the microscope and device manufacturers and service technicians. We

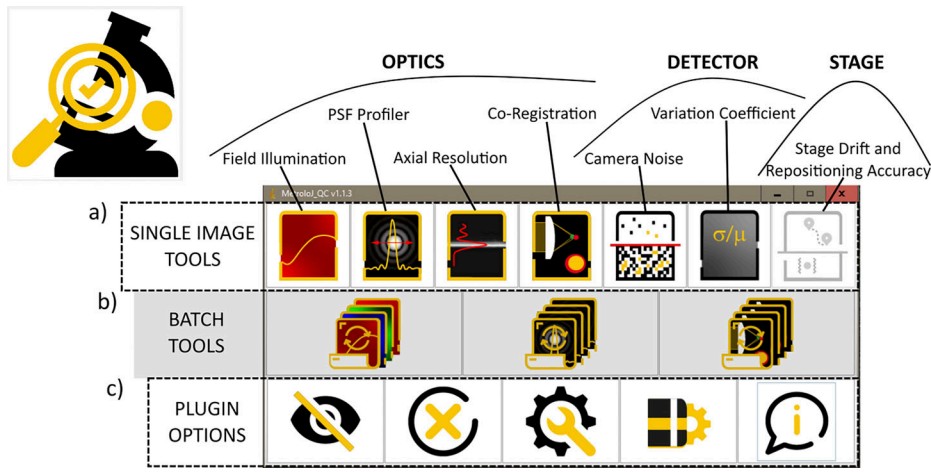

**Figure 2.** **Brief description of MetroloJ_QC ImageJ/Fiji plugin.** The plugin analyzes image data to characterize the optics components, the detector and the stage properties of a light microscope. The active window gives access to three types of tools: (a) Single image set. (b) Batch mode for automation. (c) Plugin options allowing from left to right: to toggle the main ImageJ window, to close the bar, to customize the bar and remove some above-mentioned tools, and to open the current version manual.

also recommended acquiring these metrics before and after any scheduled revision of a system to compare QC data.

We establish QC guidelines, propose tolerance values, and recommend how often microscopes should be subject to QC testing, parameters that provide valuable indicators of the setup health state and experimental reproducibility (see Table 1 for a summary and troubleshooting).

Based on many of the experiences and results presented in that work, the initiative "Quality Assessment and Reproducibility for Instruments & Images in Light Microscopy" (QUAREP-LiMi) was started in April 2020 as an international joint approach with many stakeholders including the industry in the field of light microscopy to further improve QC in the major light microscopy techniques (Boehm et al., 2021; Nelson et al., 2021).

### Lateral and axial resolution of the microscope
#### Background
One of the main missions of fluorescence microscopy is to visualize spatially cellular structures using mainly fluorescent proteins and fluorescent dyes coupled with antibodies. The resolution of a microscope, i.e., its ability to distinguish two close objects is of primary importance and is limited by diffraction. Images from single-point emitters (e.g., single fluorescent molecules in a sample) do not appear as infinitely small points on a detector of a conventional microscope. The emitted photons are scattered due to their interaction with lenses, filters, and other optical components in the imaging system and the sample itself, and therefore the resultant image is blurred. The limited aperture of the microscope objectives induces a spatial profile with a central spot and a series of concentric rings, known as the Airy diffraction pattern, referred to as the point spread function (PSF; Juškaitis, 2006; Keller, 1995; Nasse et al., 2007; Zucker, 2014).

Ernst Abbe described the blur effect and defined the resolution limit as the minimal distance allowing for two closely spaced Airy disk patterns (subdiffraction points) to be distinguished (Abbe, 1882). In biology, molecules of a few nanometers are typically organized within aggregates below Abbe's limit, which is roughly 200 nm for a high numerical aperture (NA) objective. For QC measurements, it is convenient to treat the notion of a single point emitter by measuring and characterizing the PSF (Inoué, 2006) because it gives a robust QC characterization of the objectives and it is relatively easy to prepare samples for this kind of measurement.

Evaluating and monitoring the PSF of a microscope over time is the first key step in determining its performance stability, and it has been well studied in the past (Cole et al., 2011; Goodwin, 2013; Klemm et al., 2019; Zucker, 2014) by using dedicated software tools (Hng and Dormann, 2013; Theer et al., 2014). The size, shape, and symmetry of the PSF, as compared to the theoretical ideal resolution, characterize the entire optical setup, including the objective, and affect the image quality and the subsequent quantification analysis, especially for advanced microscopy techniques. Here, we defined two metrics, a measured to theoretical PSF ratio and the lateral asymmetry ratio (LAR). We automated the PSF analysis of subresolution fluorescent beads, with a tool calculating the two metrics directly, and proposed experimental tolerance values.

#### Materials and methods
##### Sample
We used subresolution fluorescent beads for measuring the PSF of high NA objectives (Hiraoka et al., 1990). On the same slide, we also included larger beads (1 and 4 µm diameter) for easy detection of the focus plane and coregistration and repositioning measurements. The aim was to have one simple, low-cost slide containing different-sized beads that can serve for multiple types of measurements.

We used 175 nm diameter blue and green microspheres for the PSF measurements for the DAPI and GFP channels, respectively (beads PS-Speck, #7220; Thermo Fisher Scientific; $3 \times 10^9$ beads/ml). Beads were vortexed to avoid aggregates and diluted

**Table 1.  Summary of metrics and troubleshooting**

| Monitoring points (measuring frequency) | Metrics | Limit values | Possible reasons for out of limit values: Proposed solutions |
|---|---|---|---|
| Point spread function (once per mo) | FWHM XY exp/theory | <1.5 | Dirty or damaged objective: Meticulous front lens cleaning. Damaged objective: Close examination of the front lens (scratches, lens detachment); send to repair. Pinhole and collimator lens misalignment (LSCM): Check the alignment. Optical aberrations: Check the manufacturer correction specifications (APO, PlanAPO...); remove DIC or additional lens from the light path; clean optical elements in the light path; regulate correction collar for NA, temperature, or cover slip thickness. Sample: Adjust stage and sample flatness; choose #1.5 (0.17 mm thick) cover slip; image beads near cover slip; use appropriate mounting or immersion medium at the right temperature; make sure there are no bubbles in the immersion oil. Vibrations-drifts: Use an anti-vibration optical table and regulate air pressure properly; do a short time-lapse of a single bead. Galvanometric scanners (LSCM): Calibrate bidirectional mode; check at mono-directional mode. Excitation light alignment and polarization: Check sources coalignment and inspect light guide (polarization, bends). |
| | FWHM Z exp/theory | <1.5 | |
| | LAR | Case-dependent illumination source | |
| Field illumination (twice per yr) | Uniformity (U) | >50% | Damaged objective: Close examination of the front lens (scratches, lens detachment); send to repair. Large camera sensor chip size (WF, SDCM): Crop the central field of view of the camera sensor chip. Incorrect position of the field diaphragm (WF): Centering of the field diaphragm. Sources and system alignment: Adjusting the collector lens or liquid light guide position; checking possible bending of the optical fibers; coalignment of the light sources; pinhole alignment; beam size at the back focal plane of the objective. UV misalignment: Check the manufacturers' correction specifications of the optical path components. |
| | Centering (C) | >20% | |
| Co-registration (twice per yr) | Ratio $r_{exp}/r_{ref}$ | 1 | Chromatic aberrations: Check the constructor correction specifications of the considered lens (APO, PlanAPO...). Damaged objective: Close examination of the front lens (scratches, lens detachment); send to repair. Sources and system alignment: Dichroic mirror and filter positions; additional lenses and light sources alignment. Micro-lenses disk (SDCM): Check the constructor correction specifications of the optical path components. |
| Illumination power stability (once per yr) | Stability $STAB_{power}$ | >97% | Laser dying: Thorough check by the constructor. Electrical instabilities: AC-powered light source; check the power grid stability of the room. Environmental variations (thermal, air flow, humidity...): General verification of environmental and vibration conditions. Fiber polarization: Check the bends of the optical fibers; proper alignment of laser in fiber. Defective optical components (AOTF, AOM, fiber): Thorough check by the constructor. |
| | $SD_{normalized\ intensity}$ | <0.2 | |
| Stage drift (once per yr) | $V_a$: Mean velocity after stabilization | [0–50] nm/min | Environmental variations (thermal, air flow, humidity...): General verification of environmental conditions; protect the stage from thermal shifts (cover, temperature-controlled chamber, or incubator); let the stage stabilize before starting experiments. Stage control components not properly working: Check joystick at "no move" position, check controller status. Mechanical instability: Tighten the mechanical elements; thorough check by the constructor. |
| | $V_b$: Mean velocity before stabilization | [0–100] nm/min | |
| | $\tau_{stab}$ | <120 min | |
| Stage positioning repeatability (once per yr) | $\sigma_{positioning\ deviation}$ | Stage specifications | Environmental variations (thermal, air flow, humidity...): General verification of environmental conditions. Mechanical instability: Clean pieces of broken cover slip; tighten mechanical elements; thorough check by the constructor. Objective touches sample insert if position far from center: Change insert or limit movements to central positions. Not enough oil on the objective: Add enough oil beneath the whole sample before starting an experiment. Stage drift: Let the stage stabilize before starting a multiposition experiment. |
| Camera read noise (once per yr) | VAR | >90% | Electronics issue; detector temperature; detector aging; cosmic rays: Thorough check by the constructor. |
| | $STAB_{noise}$ | >97% | |

Summary of the basic metrics, the recommended measuring frequency, the experimental limit values, the possible reasons associated with their exceeding values, and troubleshooting advice.

in distilled water to achieve a density of $10^6$ beads/ml. A 50 µl aliquot of the diluted solution was dried overnight at room temperature in the dark on ethanol-cleaned type #1.5 high-performance, 18 mm square coverslip (Carl Zeiss Microscopy GmbH). The desired bead density on the slide was around 10 beads in a 100 × 100 µm field. The coverslips were mounted on slides with 10 µl of ProLong Gold antifade mounting medium (#P36930; Thermo Fisher Scientific, refractive index [RI] 1.46). As beads were directly juxtaposed to the coverslip, any spherical aberration/distortion induced by the potential RI mismatch between the mounting medium and lens immersion medium is minimized (Hell et al., 1993). Other configurations may be used to control the depth of the beads within the mounting medium and monitor the effect of the RI mismatch on image quality (to reproduce more realistic in-depth conditions of a typical biological sample). The slides were left for 3 d at room temperature in the dark to let the mounting medium fully cure. Then, samples were sealed with Picodent dental silicone (Picodent twinsil, picodent, Dental Produktions und Vertriebs GmbH). 10 bead slides were prepared in the same way and distributed to the different microscopy facilities participating in this study.

To compare the effects of the bead size on PSF measurements, we also mounted green 100 nm beads (FluoSpheres, #F8803; Thermo Fisher Scientific) using the same preparation method as described above.

### Acquisition protocol

Before any measurements, dry and immersion objective lenses must be cleaned carefully. We tested immersion and some 20× dry objectives with NA higher than 0.7. For the blue channel, the beads were imaged on wide-field microscopes with settings used for DAPI imaging (typically a bandpass 350/50 nm excitation filter, a 400-nm long pass dichroic beamsplitter, and a 460/50 nm emission filter). On laser scanning confocal microscopes (LSCM), the beads were excited with a 405-nm laser line, and fluorescence was collected between 430 and 480 nm. For the green channel, the beads were imaged with the wide-field microscopes using setups similar to GFP imaging (typically with 470/40 and 525/50 nm bandpass excitation and emission filters combined with a 495-nm long pass beamsplitter) and for LSCM, the beads were excited with a 488-nm laser and fluorescence was collected between 500 and 550 nm.

The spatial sampling rate is crucial for accurate PSF measurements. At least the Shannon–Nyquist criterion should be fulfilled to ensure that the PSF image is not deprecated by the lack of spatial sampling. This criterion defines that the pixel size should be equal to at least one-half of the resolution of the optical system (Pawley, 2006a; Scriven et al., 2008; Shannon, 1949).

In our study, most of the collected images were slightly oversampled to provide a more precise fitting. In some cases, we were limited in the choice of the adapted pixel size, thus influencing the analysis. This is the case for the lateral X/Y sampling and 20× dry objectives with NA > 0.7 coupled with cameras used in classical wide-field setups, lacking any other magnification relay lens. Whereas the typical theoretical lateral resolution is around 350 nm for the GFP channel, the typical minimum camera pixel size is around 6.5 µm in a wide-field setup, corresponding to a pixel size of 325 nm in the image. For the same

reason, respecting the Shannon–Nyquist criterion with EMCCD (electron multiplied charge-coupled device) cameras having a physical pixel size of 13–16 µm was difficult in some configurations. Concerning the lateral $z$ sampling, the Shannon–Nyquist criterion was also considered which gave for instance 0.15 and 0.2 µm step size for the highest NA objectives (1.4) for confocal and wide-field microscopy respectively (Webb and Dorey, 1995).

For LSCM imaging, the pinhole was set to 1 Airy unit (A.U.), a setting at which the pinhole size matches the diameter of the central Airy disk of the diffraction pattern to assess the standard performance of a confocal microscope. Thus, using 1 A.U. is a good compromise for image collection between signal intensity and resolution (Cox and Sheppard, 2004). In the case of a degraded PSF (not straight or not symmetric along the $z$ axis for instance), the pinhole was widely opened to record all of the aberrations. Pinhole alignment is necessary before or after the tests. Moreover, as noise affects the PSF measurement, a high signal-to-noise ratio is essential for the analysis fitting and the precise full width at half maximum (FWHM) calculation. FWHM is the width of a line shape at half of its maximum amplitude. For the Gaussian line shape (as we assume that is the case for the PSF profile), the FWHM is about 2.4 SDs. We found that single color beads, compared to multicolor ones, provide a higher signal and should preferably be used. A tutorial video summarizing the above protocol and parameters that influence PSF acquisition was produced by the working group GT3M of the RTmfm French microscopy network (https://youtu.be/ll4X_e8_mo8).

We first investigated if the proposed procedure was robust enough by examining (1) the image processing: for one $z$-stack of a single-bead five repetitions of the image processing with our plugin gave strictly identical results, (2) the image acquisition: for five acquisitions (five different time points) of the same bead we studied the variability of the measured FWHM for a single bead (Fig. S1). We found for the WF case $x = 0.256 \pm 0.002$ µm; $y = 0.26 \pm 0.002$ µm; and $z = 0.673 \pm 0.02$ µm, (3) the field of view containing several beads (Fig. S1 B). For the beads that were in the central area of the image (the central area is the area that we defined in our protocol that contains the beads to be taken into account for the FWHM calculations), we found $x = 0.261 \pm 0.004$ µm; $y = 0.266 \pm 0.003$ µm; and $z = 0.685 \pm 0.02$ µm. For the entire field of view, we found $x = 0.267 \pm 0.016$ µm; $y = 0.274 \pm 0.02$ µm; and $z = 0.676 \pm 0.032$ µm. From the above results, we concluded that image processing is repeatable and robust. The repeatability accuracy of the different acquisitions showed variability of <1% in $xy$ and <3% in the $z$-direction. When processing several beads from the central part of the field of view, we found variability of <2% in $xy$ and <3% in the $z$ axis. For the whole field of view, the variability increased to 8% in $xy$ and 5% in the $z$ axis. These results show that a small variability exists among different acquisitions and that acquisition variability is higher for the $z$ axis. When the whole field of view was taken into account, spatial aberrations that are highly linked to the objective type and quality caused a higher variability.

Important considerations when acquiring and analyzing PSF images are the signal-to-background ratio (SBR) and the signal-to-noise ratio (SNR). The SBR was calculated directly from the

MetroloJ_QC plugin, using the segmented bead mean intensity value (signal) and the mean value of a 1-μm-thick background annulus around the segmented bead (background). The SNR was calculated by using the square root of the maximum pixel intensity normalized to a 12-bit dynamic range, noting that the value describing the intensity of a pixel is only proportional to the number of photons (Stelzer, 1998). We observed that if the SBR or SNR is low, then the precision of the FWHM calculation is low (Fig. S2, A and B). Paying attention to background is therefore highly recommended, as shown by Stelzer (1998).

Another important consideration for PSF evaluation is the sampling density (i.e., pixel size). One has to collect images with a sampling density high enough (at least more than two times the resolution according to the Shannon–Nyquist criterion) in the *xy* and *z*-dimension to get the most accurate PSF Gaussian fitting. We consider that PSF can be sufficiently fitted by a Gaussian function (Zhang et al., 2007). To experimentally determine the adequate sampling rate/density, high SBR (SBR > 3 as measured by the plugin MetroloJ_QC) PSF acquisitions were performed using different voxel sizes. We did the image recording with a point scanning confocal setup since the pixel size can be easily adjusted (compared to a wide-field setup, where, due to their fixed physical camera pixel size, the options, such as camera binning or use of a magnification relay lens, are limited). The Shannon–Nyquist criterion was considered as met whenever pixel size was equal to or lower than λex/8*NA (Sheppard, 1986; Wilson and Tan, 1993). Using a 1.4 NA lens and an excitation wavelength of 488 nm, the criterion value is 43 nm. This is the case for a closed pinhole (near to 0.25 A.U.). For a 1 A.U. pinhole, the Nyquist criterion allows a 1.6× bigger pixel size (https://svi.nl/NyquistRate; Piston, 1998). In our case, with a pinhole of 1 A.U., we observed that the pixel size has an influence on lateral FWHM values when the pixel size value is higher than a threshold of 80 nm or higher than 1.16× the Nyquist criterion value (Fig. S2 C). It is statistically significant (P < 0.05) when performing Dunnett's multiple comparison test, setting the 80 nm pixel size as control.

The fourth consideration for accurate PSF measurements is the choice of fluorescent microspheres. Their size and brightness can alter the FWHM calculation. The brightest beads should be used, and this is the reason we recommend using single-labeled beads, as described earlier. Ideally, the bead diameter should be below the resolution limit of the objective. Fig. S2 D shows measurements carried out with a WF setup using a 1.4 NA objective. For 100 or 175 nm diameter beads, FWHM values were significantly different (lateral *xy* or axial *z* FWHM values show P < 0.0001 for Kolmogorov–Smirnov test for each size pair and in the lateral and axial direction). We conclude that when looking for the most accurate PSF measurements, 100 nm beads are recommended. However, for PSF monitoring over time, brighter beads are needed. They are also more convenient if a wide magnification range of objectives is to be examined using the same slide. Hence, with 100 nm microspheres, dimmer than the 175 nm beads, and with low magnification/NA lenses, detecting the beads and achieving a correct FWHM estimation may prove quite challenging. For these reasons, we recommend using

single-color 175-nm-diameter beads for reliable PSF monitoring. A mix of different bead colors may be used to speed up image acquisition and enable, in a single *z*-stack, measuring FWHM for different wavelengths.

**Metrics**

We calculated the experimental PSF values using the MetroloJ_QC plugin with the Fiji software (see Fig. S3 for the workflow). The plugin calculates the FWHM along the *x*, *y*, and *z* axis of all beads in the field of view (FOV). It applies a Gaussian fit for each of the three axes. Before analyzing the images with the plugin, we visually inspected the acquisitions. Some images were needed to be cropped to remove saturated beads. An alternative option can be used to automatically discard any saturated beads from the image using the MetroloJ_QC plugin. Only beads that were close to the center of the FOV (30% of the full chip of a sCMOS sensor, zoom of 6 for confocal) were analyzed to avoid aberrations (Hell and Stelzer, 1995). Experimental data shown in Fig. S1 confirm that aberrations and the deviation of the theoretical FWHM increase as we approach the corners of the FOV, especially for high NA objectives. At least five beads were analyzed per objective.

The experimental FWHM was compared with the expected theoretical FWHM.

(1) Wide-field:

$$\text{Lateral resolution}: res^o_{x,y} = \sqrt{2 \times \ln 2}\, \frac{1.22 \times \lambda_{em}}{2NA} = \frac{0.51 \times \lambda_{em}}{NA}$$

$$\text{Axial resolution}: res^o_z = \frac{1.77 \times n \times \lambda_{em}}{NA^2}$$

(2) Scanning confocal (pinhole ≥1 A.U., NA > 0.5):

$$\text{Lateral resolution}: res^o_{x,y} = \frac{0.51 \times \lambda_{ex}}{NA}$$

$$\text{Axial resolution}: res^o_z = \frac{0.88 \times \lambda_{ex}}{n - \sqrt{n^2 - NA^2}}$$

(3) Spinning disk confocal:

$$\text{Lateral resolution}: res^o_{x,y} = \frac{0.51 \times \lambda_{em}}{NA}$$

$$\text{Axial resolution}: res^o_z = \frac{\lambda_{em}}{n - \sqrt{n^2 - NA^2}},$$

where $\lambda_{em}$ is the emission wavelength, $\lambda_{ex}$ is the excitation wavelength, and $n$ is the refractive index of the immersion liquid. Resolution formulas are the FWHM of the PSF and not the distance of the maximum to the first minimum of the intensity profile of the PSF (Amos et al., 2012; Wilhelm, 2011). For spinning disk confocal microscopy (SDCM), we consider that the pinhole size does not fill 1 A.U. (it is the case when we use a circular pinhole size close to 50 μm; Toomre and Pawley, 2006).

The LAR is also calculated and is defined as

$$LAR = \frac{min\left[FWHM_x, FWHM_y\right]}{max\left[FWHM_x, FWHM_y\right]}.$$

**Analysis**

The MetroloJ_QC plugin works as follows: first, the *xy* coordinates of the beads are identified using a find maxima algorithm on a maximum intensity projection of the stack. Beads too close either to the image edge or to another bead are discarded

from the analysis (the user in the plugin can modify this distance). Subsequently, FWHM is measured, where the maximum intensity pixel within the entire 3D data set is determined and an intensity plot along a straight line through this maximum intensity pixel point is extracted in all dimensions. Finally, the bell-like plots/curves are fitted to a Gaussian function using the built-in ImageJ curve fitting algorithm to determine the FWHM. After automation of the FWHM measurements on several beads/datasets, average values are extracted in all three dimensions, and SDs are calculated. A batch mode enables the automation of the analysis. If the algorithm detects spots that do not originate from the beads, they can be removed computationally using the Gaussian fitting parameter $R^2$ for each FWHM measurement. We recommend a value superior to 0.95. Further inspection of each PSF, using the computed bead signal-to-background ratio, helps in removing aberrant values. Additional parameters are also measured, like lateral symmetry, or whether the acquisition meets the Shannon–Nyquist criterion.

### Biological imaging

An example of the influence of the PSF in biological imaging was a LLC-PK1 (pig kidney) cell with nuclei staining (DAPI), a pericentrin staining (ab polyclonal primary antibody ab4448 coupled to the secondary antibody Alexa Fluor 488 – A11034; Invitrogen), and a γ-tubulin staining (ab monoclonal primary antibody by sigma GTU-88 clone coupled to the secondary antibody Alexa Fluor 568 – A11031; Invitrogen). The microscope was an SDCM (Dragonfly, Andor–Oxford Instruments), and images were taken with a Plan-Apo 100×/1.45 Nikon objective and an EMCCD iXon888 camera (Andor–Oxford Instruments).

### Statistical analysis

The figures and statistical analysis were prepared using GraphPad Prism software. For LAR metric, significance was tested for the WF, SDCM, and LSCM modalities. First, the assumption of normality was tested using the Shapiro–Wilk test, with a P value <0.0001, 0.104, and 0.0813 for WF, SDCM, and LSCM modalities, respectively, which indicates that the WF distribution was not normal. Therefore, a Kruskal-Wallis test was used that determines whether the median values of the modalities are different. The test gave an $H$ value of 42.7 (out of 141 values) and a P value <0.0001 (****), indicating that there is a significant difference in the LAR metric result of the three modalities.

### Results

In total, we collected data from 117 objectives from WF, SDCM, and LSCM microscopy modalities. The technique and the setup and objective quality determine the PSF shape (Fig. 3 A). We showed that the ratios of experimental to theoretical lateral and axial PSF FWHM are better for the GFP compared with the DAPI channel (Fig. 3 B). The measured lateral FWHM mean value for the LSCM modality stayed close to the expected theoretical one with impressively low dispersion (Fig. 3 C), although it showed a large dispersion along the z axis. Many cases of elongated PSF profiles along the z axis are due to spherical aberrations, especially when the PSF is taken with water (black-capped line in Fig. 3 B) or air at low magnification and NA (25×/ 0.8) objectives (black arrows in Fig. 3 B), as Cole et al. (2011) have shown experimentally and Hell et al. (1993) theoretically. The WF and

SDCM FWHM measurements show similar behavior. The measured FWHM values are not as good along the xy axis as on the z axis, compared to the theoretical ones. Of note, as the sampling criterion was met in z, the observed ratios for the axial FWHM are closer to the theoretical value. For instance, a WF case that is far from the theoretical value is shown with a blue arrow followed by "iii" (Fig. 3 B, the "iii" corresponds to the "iii" PSF profile of Fig. 3 A) and concerns a 40×/1.3 Plan-Apo objective lens, and acquisition with a pixel size of 183 nm. We measured a ratio of 2.33 for xy and 1.53 for z. Coma aberrations or astigmatism can also be present, although their influence on the measured xy FWHM stays low. A typical PSF example showing coma aberration is the one with the blue arrow followed by "ii" in Fig. 3 B ("ii" PSF in Fig. 3 A), characterized by an xy ratio of 1.65 and a z ratio of 1.31. It should be mentioned that we avoided measuring PSF for wavelength ranges with objectives that are not meant to be corrected for the used wavelength range. More specifically, this is the case for Plan Fluor or NeoFluar objectives for the DAPI channel. For instance, a 40×/1.3 NeoFluar objective gave a ratio of 3.3 and 2.1 for xy and z, respectively. These values are not shown in Fig. 3 B, as we considered that these objectives are not designed to give acceptable performances in these wavelength ranges.

From the experimental results shown in Fig. 3 B, we propose to set the tolerance value at a ratio of 1.5 between measured and theoretical FWHM. 66% of the lateral FWHM and 82% of the axial measurements are located inside the 1.5 ratio dark green square.

We studied the symmetry of the PSF by calculating the mean values of the LAR, defined as the ratio between the minimum and maximum experimental x and y FWHM of individual beads (Fig. 3 D). We observed that for LSCM the mean LAR is much lower than for the other two techniques (LSCM: 0.85, SDCM: 0.92, WF: 0.96) and it is statistically significant (P < 0.0001).

We finally measured the stability of the objective performances by analyzing PSFs acquired for 3 yr using the same microscope, objective, sample, and were also acquired by the same operator (Fig. 3 E). Variations of lateral and axial FWHM were small (<5%). In October 2019 (Fig. 3 E, red arrow), and 5 d after a planned manufacturer's maintenance visit, measurements showed a 10% increase in the x-axis FWHM, while no significant change was observed for the y-axis FWHM. We recovered the symmetry after taking out and reinstalling the camera and paying special attention to avoid any tilt between the camera, the camera adaptor, and the microscope body (measurement of December 2019 and later on).

Finally, we showed that a degraded PSF influenced biological imaging, according to the resolution level of the studied structure. To this end, we imaged the centrosome and cellular DNA in a cell undergoing mitosis (Fig. 3, F–H). For the imaging of larger structures like the nuclei of this example, PSF quality is not that important. For near-to-diffraction limited imaging and colocalization analysis, like for the centrosomal proteins γ-tubulin and pericentrin, PSF quality is of great importance, as shown in the zoomed image of the centrosome proteins (Fig. 3, F–H). In this case, an under-sampling (Fig. 3 G) or a DIC prism in the light path (Fig. 3 H) showed degradation to the PSF and the biological

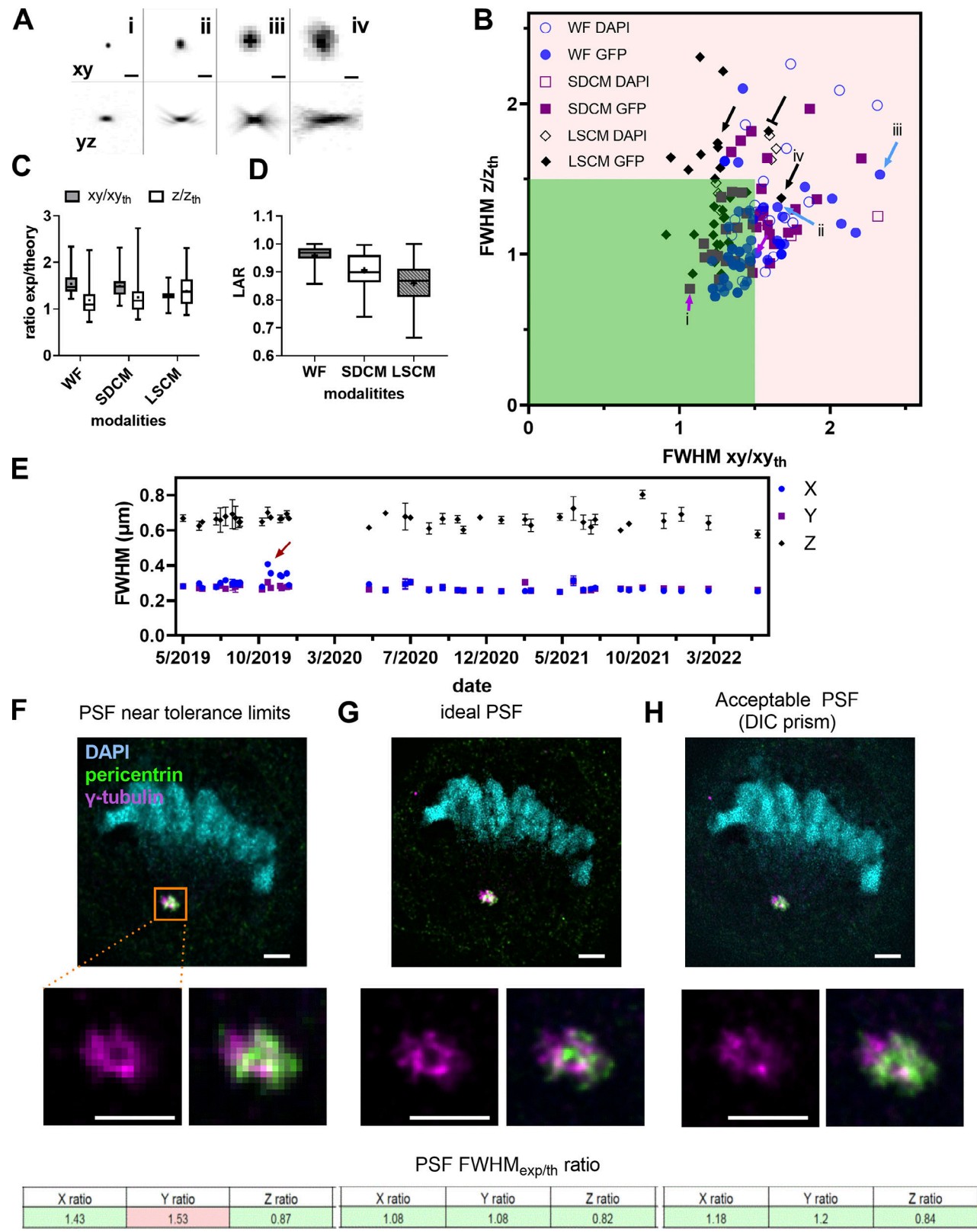

Figure 3. **PSF distribution and influence on image quality of biological structures. (A)** Square root PSF along *xy* and *yz* of four cases (i-iv) shown in B. Scale bar is 1 µm. **(B)** Ratio of experimental to theoretical lateral and axial PSF FWHM values for WF, SDCM, and LSCM techniques for the DAPI and GFP channels. The red arrows show PSF in the limit values, the blue arrows show PSF of medium quality, the black-capped arrow shows a 40× water immersion objective with an elongated *z*-axis PSF, and the black arrows show PSF cases from dry 20× objectives. **(C)** Statistical analysis with median (horizontal line), mean values (dot), SD (box) and min/max values (whiskers) of B. **(D)** Box plots summarizing the distribution of the LARs for the PSFs shown in B. Statistical significance was determined by using Kruskal–Wallis test (P value was <0.0001 [****]). For C and D, the independent *n* data points were 61, 43, and 37 for WF,

SDCM, and LSCM, respectively. **(E)** Stability evolution of PSF over 32 mo for an upright WF microscope. The red arrow shows when the PSF is significantly different along the *x* axis. The gap in the dates of January 2020 corresponds to the COVID-19 lockdown when no experiments could be carried out. Error bars represent the SD. At least five PSFs were analyzed per date. **(F–H)** Degradation of image quality on a biological sample depends on PSF quality. The biological sample is a cell in division (anaphase state). The cell nucleus is labeled with DAPI (cyan), the pericentrin protein of the centrosome is labeled with Alexa Fluor 488 (green), and the γ-tubulin is labeled with Alexa Fluor 561 (magenta). The images are acquired with a SDCM, Plan-Apo 100×/1.45 objective. For each PSF case, we show the acquisition of the cell in three colors, the zoom of the centrosome region for one color and two-color overlay, and the PSF summary results of the mean FWHM$_{exp/th}$ for the three axes. Scale bar is 2 μm. **(F)** Imaging with a PSF FWHM ratio along the *y* axis that is out of the tolerance values due to oversampling (big pixel size). **(G)** Imaging with an ideal PSF (use of additional lenses inducing a 3× magnification to respect Nyquist criterion). **(H)** Imaging with an acceptable PSF: the added DIC prism introduces coma aberrations.

---

image, highlighting the importance of optimal PSF settings to study diffraction-limited objects.

### Discussion

Results showed the high variability of the PSF measurements due to a large variety in the quality of objectives used. Variability is also introduced by the users whose capacity to fully follow a strict procedure may vary. Besides any objective-induced aberration like coma, astigmatism, or distortion (Sanderson, 2019; Thorsen et al., 2018), special care should be given to following the same acquisition protocols, using the same slides, and paying special attention to avoid errors such as forgetting a DIC prism within the optical path, not cleaning the objectives before the measurements, a non-flat sample or stage insert, or not properly setting the objective correction collar, when necessary. Variability was associated with suboptimal sampling rate (e.g., inevitable for WF or SDCM for the lateral values for low magnification objectives and big pixel size cameras) or blue-emission dyes. Most conventional microscopes and commercially available optics are designed for optimal performance at visible wavelengths (Bliton and Lechleiter, 1995), and adding aberration correction within the UV or near-UV wavelength range is challenging. Spherical aberrations (due to refractive index mismatch or incorrect adjustment of the lens correction collar) and pinhole misalignment are possible sources of the high variability of LSCM axial FWHM values.

We also showed that while asymmetry was negligible in WF and SDCM, the average asymmetry ratio of LSCM is 0.86. This is expected, as FWHM is smaller in the direction perpendicular to the direction of the linear polarization of the excitation laser in high NA objectives (NA > 1.3; Li et al., 2015; Micu et al., 2017; Otaki et al., 2000).

We proposed experimentally defined limit values. Objective resolution performances can be considered within limits if the lateral and axial measured/theoretical ratios are kept below 1.5 (dark green zone of Fig. 3 B). Although such limit values may seem quite high, one has to keep in mind that these tolerance values rely on some theoretical formulas, considering optimal, though unobtainable, conditions such as shot-noise free confocal imaging or subresolution point sources, while in reality, we used 175 nm beads. It is important to specify that a ratio value slightly higher than the tolerance values defined here does not necessarily mean that a return of the objective to the manufacturer for a check is necessary. The user of the microscope can nevertheless continue monitoring the PSF quality of the objective while keeping in mind that imaging of structures with sizes near to the optical resolution of the microscope should be avoided.

For regular QC monitoring of the microscope, we recommend measuring PSF once per month. Our data showed a high dispersion of variability of the measurements over time for objectives with NA > 1. Improper use of the objective and generally a misalignment of the whole microscope system can result in a measured PSF value far from the expected theoretical one. Therefore, regular monitoring of the PSF with a frequency of once per month, if the human resources of the facility or research team allow that, is highly recommended. In cases of demanding experiments, like in super-resolution microscopy, PSF should be measured more frequently, even before each experiment. The same PSF measurement before the acquisition can be used for image deconvolution processing using experimental PSFs (McNally et al., 1999).

When the experimentally defined tolerance value is not met, the microscope user should use all available tools for troubleshooting (look at Table 1 for a summary and proposed actions). For instance, an adjustable pinhole at the LSCM is a possible solution. First, one has to be sure that the pinhole is well aligned before performing any confocal PSF measurements. Whenever the performance significantly diverges from theoretical values, we recommend using a fully opened pinhole or checking the objective on a WF microscope. These comparisons are helpful for precisely identifying the origin of the issue (i.e., objective-associated or related to some other component). For SDCM, the most often fixed pinhole size increases the variability, especially when we use objectives with lower magnification than 60× or 100×. Alternatively, the method described by Zucker et al. (2007) using a metal-coated mirror and detecting the reflected light of the laser can be performed to ensure pinhole alignment and measure the axial resolution.

To sum up, the impact of a low-quality PSF on the result of biological imaging is relative to the structure to be imaged. For objects in the cell near the resolution limit, like in Fig. 3, F–H, it will induce aberrant or artefactual structures and influence colocalization studies. For high- or super-resolution techniques (like, for instance, Airyscan, STED, SIM) the quality of the PSF should be optimal. For bigger structure imaging (e.g., nuclei imaging of Fig. 3 F), the quality of the PSF is not that crucial and imaging can still be satisfactory.

### Field illumination
#### Background

Quantitative optical imaging requires a perfectly even field illumination. For example, if the center of the field of view is

much brighter than the corners, two cells with an identical level of GFP expression will appear with a difference in intensity. Ideally, all the pixels in the image of a uniform sample should have the same gray level value across the field of view, considering the intrinsic noise of detectors (Murray et al., 2007). In the real world, illumination light source uniformity, alignment, and optical aberrations (from the objective and additional optics included in the light path) can affect the homogeneity of the field illumination.

Regarding light source uniformity, for WF microscopes the bulb-type sources are mercury, metal-halide, or xenon arc lamps. They contain a central bright source of illumination that is inherently non-uniform and often requires a light diffuser to homogenize the emitted light (Ibrahim et al., 2020). More recently, LED sources with their long lifetimes and excellent stability offer a highly improved field flatness due to their coupling with the microscope with a liquid light guide (Aswani et al., 2012). For SDCM and LSCM, the light sources are lasers that are coherent, with an intensity distribution described by a Gaussian function. The Gaussian profile is naturally inhomogeneous, presenting a maximum central spot of intensity. Lasers are coupled to the microscope with optical fibers that can introduce another source of non-uniformity. Relay optics and spatial filtering can be used to homogenize the beam Gaussian profile (Gratton and vandeVen, 1995). Scanners that are not perfectly calibrated can also influence the laser illumination uniformity as well as the detection uniformity of LSCMs (Stelzer, 2006). Furthermore, the dichroic mirror and the filter positioning in the filter cubes (for WF) or the emission filter wheel have an impact on the observed field illumination for each channel (Stelzer, 2006).

Accurate image intensity quantification over the entire field of view requires characterizing the field illumination pattern and correcting for heterogeneities if necessary. It is essential for image stitching, ratiometric imaging (Fura2, FRET), or segmentation applications. When non-uniform illumination occurs in experiments involving tile scans (acquisition of adjacent XY planes), it leads to optical vignetting and unwanted repetitive patterns in the reconstructed image (Murray, 2013). An image of non-uniformity, as a reference, is then a prerequisite to correct this shading effect. In live-cell experiments localized phototoxicity, but also with fixed samples, inhomogenous bleaching of the sample might be observed. Previous studies investigated the field illumination measurements by defining theoretical metrics and measurement protocols (Bray and Carpenter, 2018; Brown et al., 2015; Model and Burkhardt, 2001; Stack et al., 2011; Zucker and Price, 1999). Here, we proposed the centering and uniformity metrics using simple tools and we defined limit values.

## Materials and methods
### Sample
For the field illumination flatness assessment, fluorescent plastic slides can be used (blue, green, orange, and red, provided by Chroma Technology Group #92001 or #FSK5; Thorlabs). Their inconvenience is the low signal obtained in the far-red channel (>650 nm emission). For a complete field illumination characterization, measurements should be performed for all four main channels (DAPI channel through Cy5 channel). For a quick and regular characterization, a field illumination measurement at the DAPI and GFP channels can be enough. A glass coverslip can be sealed on the plastic slide with immersion oil to protect the plastic slide from scratches (Zucker, 2006b). An alternative to the plastic slide is to use a thin layer of a fluorescent dye (Zucker, 2014). The thin layer of a fluorescent dye gives a more precise illumination pattern and can be more convenient for shading corrections for tile scan acquisitions (Model and Burkhardt, 2001). We compared the plastic slides with a self-made homogeneous thin fluorescent specimen. We used a highly concentrated (1 mM) dye solution (rhodamine G6; Sigma-Aldrich) between a glass slide and coverslip for this specimen. We tested both plastic slide and dye solution and found that for the defined metrics, the plastic slides are convenient (Fig. S4), although, for SDCM, the pinhole crosstalk detection is well seen with plastic slides and may alter the uniformity measurement (Toomre and Pawley, 2006). We also observed that the centering measurement is independent of the sample type.

### Acquisition protocol
For WF and SDCM microscopy, the full camera chip was used. For confocal microscopy, we used the recommended zoom 1× from the manufacturer to yield a uniform field of view, as suggested in the ISO 21073:2019 norm (ISO, 2019). We placed the slide on the stage and focused on the surface. We measured the field illumination slightly deeper than the surface level (around 5–10 μm) to avoid recording scratches or dust (Fig. S5 B). The acquisition parameters were set to take full advantage of the detector dynamic range and avoid saturation. Only one frame is required. A supplementary step of averaging the intensities of several images can be used for calibrating the flat field correction and creating a shading correction image, necessary for tile scan imaging (Model, 2014; Young, 2000). In our case, we focused on characterizing the field illumination, and a single image was sufficient.

### Metrics
We defined the following two metrics to evaluate the field illumination:

(1) The uniformity of the illumination at the observed FOV, U (%), is expressed as follows:

$$Uniformity(\%) = 100 \times \frac{Intensity_{min}}{Intensity_{max}},$$

where intensity$_{max}$ and intensity$_{min}$ are the maximum and minimum intensity acquired in the FOV, respectively. The size of the FOV influences the value of this metric. A ratio of 1 (100%) represents an ideal case.

(2) The centering of the maximal zone of the intensity of the image, C (%), is expressed as follows:

$$Centering~(\%) = 100-$$
$$\left[ 100 \times \sqrt{\left(X_{max} - X_{center}\right)^2 + \left(Y_{max} - Y_{center}\right)^2} \times \left(\frac{2}{\sqrt{x^2 + y^2}}\right) \right],$$

where X$_{max}$ and Y$_{max}$ are the coordinates of the maximum intensity zone, X$_{center}$ and Y$_{center}$ are the coordinates of the center

of the FOV, and $x$ and $y$ determine the width and the height of the image, respectively. The size of the FOV does not influence the metric value (normalization by the geometrical center position of the image). A value equal to 1 or 100% is the ideal case.

The uniformity and the centering metrics are included in the ISO 21073:2019 norm (ISO, 2019).

The MetroloJ_QC plugin was used, and the analysis workflow is shown in Fig. S5. The algorithm starts by applying a Gaussian blur filter (sigma = 2) to smooth the homogeneity image to avoid isolated saturated pixels (e.g., hot pixels) and to smooth local imperfections (Bankhead, 2016). Then, the plugin locates both minimum and maximum intensities. It also finds the center of the intensity. As with the MetroloJ plugin, a normalized intensity image is calculated. Next, using the normalized image, the maximum intensity location is determined considering a reference zone (either the 100% intensity, i.e., the geometrical center of all pixels with a normalized intensity of 1, or any other zone, such as 90–100%, i.e., the geometrical center of all pixels with a normalized intensity between 0.9 and 1).

The user is prompted to divide the normalized image intensities into categories (bins). This value will be used for the computation of the reference zone and generation of the iso-intensity map. A value of 10 will generate isointensity steps of 10%. A threshold is then used to locate all pixels with the maximum 100% intensity or the last bin window. The geometrical center of this reference zone is then located. Distances of these points to the geometrical center (center of intensity, maximum intensity, center of the thresholded zone) are computed, and the field illumination metrics are calculated.

The user can choose to discard or not a saturated image. Whenever saturation occurs in a few isolated pixels, noise may be removed using a Gaussian blur of sigma = 2. Note that saturation computation is carried out after the Gaussian blur step. Hence, whenever aberrant saturated isolated pixels are polluting the channel, if Gaussian blur removes them, the image will no longer be considered as saturated, as no saturated pixels will be found. In the case of "clusters" of saturated pixels, the applied Gaussian blur is not strong enough to eliminate the cluster center's saturated pixels and the channel will still be recognized as saturated and skipped if the discard saturated sample option is selected. A batch mode enables the analysis automation.

### Biological imaging

An example of the influence of the field illumination in biological imaging was MCF-7 cells with a nuclei staining (Hoechst 33342; Invitrogen) and an actin staining (phalloidine Alexa Fluor 488, A12379; Invitrogen). The microscope was a SDCM (Dragonfly, Andor–Oxford Instruments), and images were taken with a Plan-Apo 60×, 1.4 NA Nikon objective and an EMCCD iXon888 camera (Andor–Oxford Instruments).

### Results

We studied the uniformity (U) and centering (C) distribution for the three main microscopy modalities by performing measurements for 130 objectives for a wide range of magnifications (5×–100×) mounted on 35 microscopes (Fig. 4 A). The U–C mean values were 61–76 and 53–82% for WF and SDCM techniques

respectively, followed by a higher dispersion of both metrics for the WF technique (Fig. 4 B). To note, 60% of the SDCM microscopes were CSU X1 model and 40% W1 model. The former shows better U–C values due to the smaller FOV. For LSCM, the U–C mean values of all measurements were at 65–51%, with a higher dispersion compared with the two other modalities (Fig. 4 B). We studied the influence of the wavelength on the U–C values. We observed that for the DAPI channel, the uniformity showed a higher dispersion than for the GFP channel with a lower mean value (56% for DAPI and 68% for GFP). The results for the centering were similar, with a slightly higher mean value for the GFP channel (68% DAPI and 73% GFP). Two typical field illumination pattern examples of the DAPI channel low U values are selected in Fig. 4 A (acquisitions with a 10× and 63× objective on a LSCM).

We also studied the U–C metrics distribution according to the detector type. We showed that when using cameras, the sensor chip size influenced the observed U values (Fig. S6 and example of Fig. 4 A for a 10× objective, GFP channel). Field illumination images acquired with large FOV (e.g., 13.31 × 13.31 mm) sCMOS cameras showed a lower U, as most of the optics of the microscope were originally designed for the use of smaller imaging area CCD (e.g., 8.77 × 6.60 mm) or EMCCD (e.g., 8.19 × 8.19 mm) devices.

We also showed that the source type influences the field illumination (Fig. 4 C). One can clearly see that arc-lamp–based light sources result in U–C values with high variability, while LED sources result in less dispersed U–C values and notably excellent centering (mean value at 87%). Laser sources show a high dispersion of C values for LSCM with a mean value of 53%, while for SDCM and WF techniques the U–C value dispersion is smaller and the mean C value significantly higher (78%). This can be explained by the difference in the detector type (Fig. S6).

From the above measurements, we defined the experimental tolerance values as follows: U > 50% and C > 20% (Table 1). In total 27% of our measurements were out of the limits for both metrics. For biological applications, low U–C values can result in inaccurate quantification studies. Fig. 4, D and E shows the field illumination of a 63×/1.4 objective for a SDCM microscope using a plastic slide and a labeled biological sample (nuclei with DAPI and actin with Alexa Fluor 488 for GFP channel). We clearly see that for the DAPI channel, the uniformity values are less good and below the previously defined tolerance values (U: 36.7% <50% which is the limit value). Apart from a significant dimmer detected signal on the corners of the FOV, the low U values influence the quality of a tile scan when this is applied. When the central region of the FOV was acquired, the field illumination was significantly improved (U: 61.5% and C: 68.8%, Fig. 4 F).

### Discussion

For techniques using a detection camera (WF and SDCM), our results demonstrate that the sensor chip size influences the U metric. Although smaller chips have been used in the past, most manufacturers recently introduced larger field corrections for objectives, taking into account the whole internal microscope light path. Hence, whenever homogeneity illumination is affected in the case of a wide camera sensor chip, besides choosing

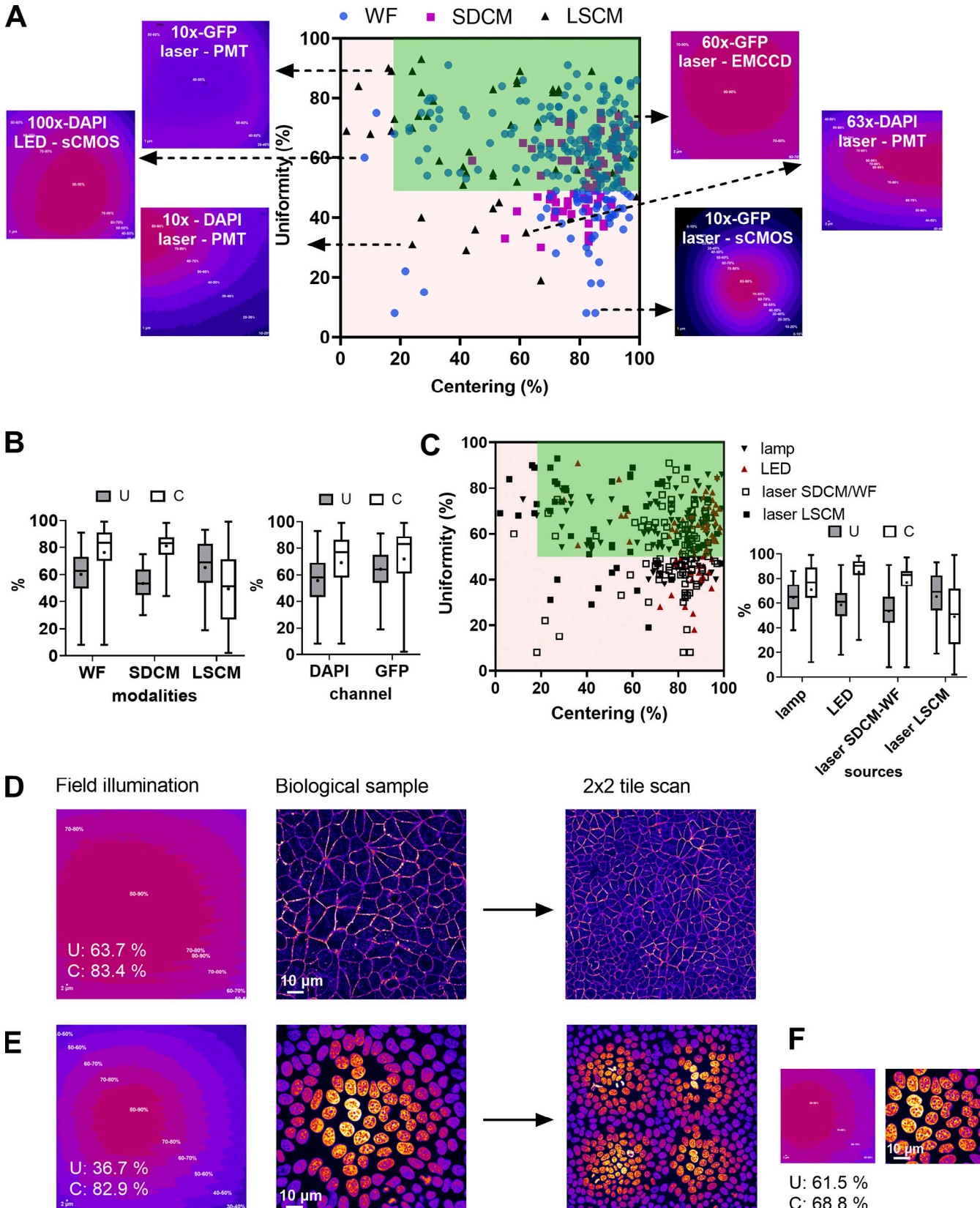

Figure 4. **Distribution of uniformity and centering field illumination flatness metrics and their influence on biological imaging. (A)** Uniformity (U) and centering (C) distribution are classified into microscopy modalities, with representative field illumination pattern examples. **(B)** Statistical analysis of A, according to the modality and the excitation channel. The mean value is shown as a cross, the SD as a box and the whiskers correspond to the min and max. **(C)** U–C distribution classified into excitation source types with the statistical analysis study. **(D–F)** Influence of field illumination flatness on image quality of

biological structures. The objective is a Plan-Apo 60×/1.4 (Nikon) on a spinning disk microscope (Dragonfly-Andor). **(D)** Field illumination pattern and U–C values for a plastic slide extracted by MetroloJ_QC, the corresponding image for a DAPI stained nuclei sample, and a 2 × 2 tile scan. **(E)** Same configuration as D for the GFP channel and actin Alexa Fluor 488 labeling. **(F)** The 50% central area of D with the U–C calculated metrics.

better-corrected objectives, in the case of a WF system, it could be advisable to adjust the collector lens or the liquid light guide position. For other cases, there remains the possibility of cropping the image to a central area of the sensor chip or using digital shading correction. The latter should be used with caution because it strongly modifies the value of the acquired intensities.

The excitation wavelength affects mostly uniformity. Most LSCM setup designs involve the 405 nm laser line passing through a different fiber/lightpath than "visible" laser lines, resulting in a different alignment between DAPI and visible channels, which can explain the U differences (Pawley, 2006b). For LSCM modality, the pinhole alignment can influence drastically uniformity. Such adjustment is not always accessible to the user, thus talking with the service technician and the constructors is necessary to ensure a properly aligned microscope.

Results also showed that lamp sources result in the least favorable U–C values, most often due to the less accurate autoalignment procedure. Laser sources are more accurately aligned (high C values) and present a better Gaussian profile, even though for SDCM and WF techniques their coupling to a large area sensor chip often results in low U values.

We propose a measuring frequency of twice per year for each objective and all channels detected by the microscope. In addition, after changing arc lamps, laser replacements or any other manipulations on the coupling of the light source to the microscope or the scan heads, field illumination needs to be checked.

For biological imaging, we defined the experimental limit values of U > 50% and C > 20%. When these values are not met, the microscope user should use all available tools for troubleshooting (Table 1). An after-sales service visit is not necessarily essential, but special attention should be paid when images are used for quantitative analysis, like colocalization studies, ratio imaging, deconvolution, and segmentation. In these cases, one should carefully characterize the field illumination. Values far from the limits that we give previously do not allow trustful quantitative analysis. Depending on the application, low values can have serious (quantification of the intensity of objects located over the entire FOV) or manageable consequences (segmentation, tracking). Especially when the tile scan option is used for imaging of large samples (e.g., slices of tissues), inhomogeneous illumination can deteriorate the resulting image (Fig. 4 E). Shading correction can be applied to correct the tiled image, but should be acquired with an appropriate uniformly fluorescent sample (slide containing a high concentration of a dye, like fluorescein, for better results).

## Co-registration in *xy* and *z*
### Background
When specimens are labeled with different colored fluorophores, the non-experienced microscope user expects an image in which the position of these different markers is absolute and not influenced by chromatic effects or other distortions (barrel, pincushion, dome, and 3D-rotation). Under ideal conditions, imaging of multicolor microspheres can lead to excellent colocalization in *x*, *y*, and *z*. However, a microscope image can suffer from non-optimal corrected objectives that induce a chromatic shift. In the *z* direction, this aberration is the result of the modification of the position of the focal point of the objective according to wavelength (Nasse et al., 2007). Depending on the objective corrections this shift may not be the same in the whole FOV. The chromatic shift may also be introduced by a misalignment of the excitation sources (if several lines are used in the confocal scanning scheme), insufficient chromatic, or other aberration-corrected optical elements, misaligned filter cubes (filters or dichroic mirrors not correctly positioned in the cube), the use of non-zero-pixel shift multiwavelength dichroic mirrors, stage drift, pinhole or lens collimator alignment, or misaligned camera(s) (Comeau et al., 2006; Zucker, 2006a).

The chromatic shift may influence quantitative imaging, like colocalization studies of proteins. It is thus important to characterize the chromatic shift and when needed to correct it. In this study, we proposed a protocol to evaluate coregistration for an objective in its central field, where the best chromatic corrections are expected. We do not seek calibration for chromatic differences and correction for the entire FOV. Calibration is often useful as coregistration is not linear across the FOV (Kozubek and Matula, 2000). A spatially resolved analysis of coregistration might help to identify the region of interest of the FOV and correct the shifts (e.g., for colocalization studies, ratiometric, or FRET imaging). To evaluate the coregistration in *xy* and *z* directions, we used multilabeled microspheres (Goodwin, 2013; Mascalchi and Cordelières, 2019; Stack et al., 2011) and defined a coregistration ratio as our metric.

### Materials and methods
#### Sample
For the coregistration measurements, a custom-made bead slide containing 1 and 4 μm diameter, four-color beads (i.e., beads incorporating multiple-colored dyes) is used. We diluted the 1 μm bead solution (#T7282, TetraSpeck; Thermo Fisher Scientific) in distilled water to achieve a density of $10^5$ particles/ml. The 4 μm beads (#T7283, TetraSpeck; Thermo Fisher Scientific) were diluted to a further extent to achieve a density of $10^4$ particles/ml.

#### Acquisition protocol
3D stacks of multicolor fluorescent beads in two or more channels were collected along at least 10 μm. For LSCM systems, the zoom was higher than 3 to increase the acquisition speed. The Shannon–Nyquist criterion was met whenever achievable and saturation was avoided. At least five beads were acquired, coming from at least two different FOVs. The beads should be very close to the center of the image (not more than one-fourth

of the total field of view) since objectives are usually best corrected for chromatic shifts in their central field.

While subresolution beads are mandatory for resolution assessment, coregistration studies are possible with subresolution beads (0.1–0.2 µm) up to a few micrometer-size beads. We first studied the influence of bead size on the coregistration results. We observed that beads with a size of 1 µm are ideal for coregistration studies, while coregistration results based on smaller than 0.2 µm beads give a significantly higher dispersion (Fig. S7). We attribute this difference to the difficulty of achieving a high SBR with the 0.2 µm beads, a necessary condition for estimating their center coordinates.

### Metrics

Our metric here was the $r_{exp}/r_{ref}$ ratio. In more detail, the acquisition consisted of 3D stacks with two or more emission channels. MetroloJ_QC plugin uses automation of the MetroloJ coregistration algorithm. Beads were identified and saturated beads or beads too close to the image border or another bead were discarded. The bead center of mass was subsequently calculated for each channel and the distances between the centers were estimated as $r_{exp}$. For each channel to channel distance $r_{exp}$, a reference distance was calculated as $r_{ref}$, as previously described (Cordelières and Bolte, 2008; Fig. S8 D). When $r > r_{ref}$, two points A and B are considered not colocalized. The plugin calculated the ratio $r_{exp}/r_{ref}$ for each color couple. A ratio higher than 1 indicates, in our case, a not accepted colocalization. The shortest wavelength of the color pair was used to calculate the theoretical resolution, as it is the most stringent resolution value. When analyzing the usual four-channel images, six combinations need to be considered.

MetroloJ_QC automatically generated analyses for all possible channel combinations, measured the pixel shifts, the (calibrated/uncalibrated) intercenter distances, and compared them to their respective reference distance $r_{ref}$. A ratio of the measured intercenter distance to the reference distance was also calculated. Images with more than one bead can be analyzed, and a batch mode enabled the analysis of multiple datasets. Fig. S8 and the plugin manual provide a more detailed description of our coregistration workflow.

### Biological imaging

Monocyte dendritic cells (MoDC) were used with actin (phalloidin-Alexa Fluor 405, A30104; Invitrogen) and beta3 integrin staining (primary/secondary antibody conjugated to Cy3). Acquisitions were carried out with an upright WF microscope (AxioImager Z2; Carl Zeiss Microscopy GmbH), a Plan-Apo 63×/1.4 objective, and a sCMOS Zyla 4.2 camera (Andor–Oxford Instruments).

### Results

We collected data from setups of the three microscopy techniques using 92 different objectives.

We observed that >73% of the calculated coregistration ratios show nearly perfect coregistration (i.e., ratio <1) in blue-green (B-G) and green-red (G-R) channel combinations (Fig. 5 A). Our tolerance values range below unity (green zone of Fig. 5, A and B). When looking at individual combinations, 96% of the measures were below 1 for the G-R, 75% for G-FR (green-far red),

60% for B-R, and only 61% for the B-G. A comparison of the three different microscopy techniques shows that the above statement is true with all setups (insets of Fig. 5, A and B). However, SDCM is more often associated with insufficient chromatic correction with 60% of the measured ratios below 1 compared to LSCM or WF with 70 and 83%, respectively.

A closer look at outlier values highlights interesting features. As expected, objectives less corrected for chromatic aberrations between the blue and the other channels showed high coregistration ratios when compared with the Plan-Apo series. It is the case for the 40×/1.3 Zeiss Plan-Neofluar of a WF and the Nikon Plan-Fluor objective of a SDCM setup represented with a round hollow blue spot pointed out with a blue-capped line (ratios B-G: 1.94, G-R: 0.81) and the square hollow red spot pointed out with a red-capped line (ratios B-G: 1.48, G-R: 0.98), respectively (Fig. 5 A).

Interestingly, we also showed that coregistration shifts can be due to non-objective associated issues. The blue arrow (ratios B-G: 1.44, G-R: 1.34) was obtained with a 63×/1.4 Plan-Apo Zeiss objective of a WF system. After testing the same objective on a different microscope and testing different equivalent filter cubes on that setup, we found out that one source of error came from the alignment of the filter in the cube. A similar cube equivalent case was associated with ratios of B-G: 0.19 and G-R: 0.23, confirming that the issue came from the misalignment of the cube components (dichroic mirror).

It is sometimes easier to pinpoint the origin of poor chromatic performance. The points shown with black cap end lines (B-G/G-R ratios of 2.93/0.32 for a 40×/1.3 and 1.92/0.31 for a 63×/1.4 NA objectives) were taken with a Leica SP5 confocal equipped with the classical light path design involving two laser injection fibers, one for the 405 nm diode laser and another for all other "visible" wavelengths. As correction mismatch was only observed with a combination involving the 405 nm-associated B channel, it was easy to identify a 405 nm light path correction lens, located before the objective, as the origin of these poor performances. We reduced the observed shifts after careful correction lens realignment (2.67–0.43 and 1.35–0.04 for 40× and 63× lenses respectively, black arrows).

Fig. 5 B shows the coregistration results for two other different color couples (B-R and G-FR). Compared with Fig. 5 A, we observed that the G-FR couple performs slightly less well than the equivalent G-R, which is expected as the wavelength difference is higher (Bliton and Lechleiter, 1995). Using the same arrow type/color scheme, most of the extreme cases highlighted in Fig. 5 A are outliers in Fig. 5 B. We focused on a limit case where using the same WF microscope, a first Plan-Apo 63×/1.4 objective satisfied coregistration ratios (Fig. 5 C) and then a second objective with the same characteristics, showed higher coregistration ratios (Fig. 5 D), far from the defined tolerance value of 1 (B-R: 1.54, orange arrow in Fig. 5 B).

Using the same setup, we showed the influence of bad coregistration values in biological imaging (Fig. 5, E and F). Structures that were supposed to colocalize (actin and beta3 integrin) were labeled with two different dyes (Alexa Fluor 405 and Cy3). We clearly saw that when we are interested in

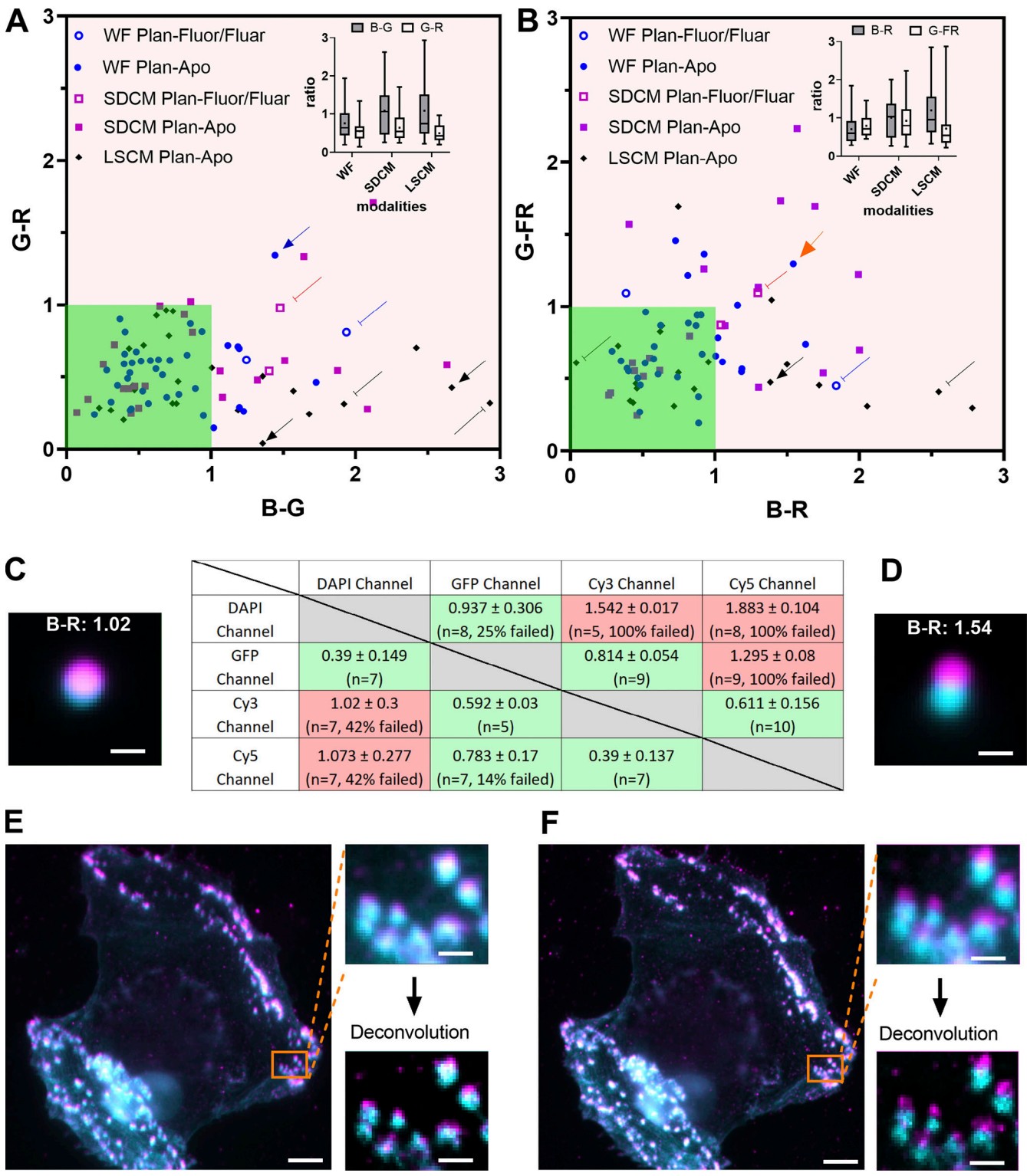

Figure 5. **Co-registration distribution and influence on biological imaging. (A and B)** Co-registration ratio of a blue-green channel combination compared to green-red (A) and green-far red (B) channel combination classified by the microscopy technique and the kind of objective (Plan-Apo or Plan-Fluor/Fluar). The arrows highlight some extreme cases. The dark green area highlights the acceptable coregistration ratio values. Insets show the box plots of the ratio distribution. The data points were 39/20, 24/23, and 28/23 for WF, SDCM, and LSCM for A and B, respectively. **(C and D)** Co-registration ratio table results for 1 µm beads for two different Plan-Apo 63×/1.4 objectives, on the same WF microscope. Totally, 7 and 10 beads were used in each case. The DAPI channel in cyan and Texas Red channel in magenta are represented on the bead images, as well as the calculated ratio B-R. The D configuration is pointed out with an orange arrow in B. Scale bar is 1 µm. **(E and F)** Influence of coregistration on biological imaging using the same objective and setup of C and D. Cell is two color labeled with actin (Alexa Fluor 405, cyan) and beta3 integrin (Cy3, magenta). The two labeled structures are supposed to colocalize. The insets are zoomed WF and deconvolved images of the areas highlighted in orange. Scale bar is 5 and 1 µm for the cell image and the zoom, respectively.

colocalization studies of micrometer size structures (zoomed images of Fig. 5, E and F), the coregistration ratio should be below unity. The effect becomes even more important when we deconvolute the images.

### Discussion

The results showed not unexpectedly that most of the objectives are better corrected for the G-R combinations, while the combinations with the blue channel (DAPI) are more often associated with poor coregistration (Bliton and Lechleiter, 1995). This ensures that the tested microscopes were most adapted for imaging at that range of the visible spectrum, often used for live-cell imaging.

WF performed better when compared to SDCM and LSCM for the coregistration of the DAPI channel with the GFP or RFP. This might be a result of the lasers used for LSCM and SDCM that have to be carefully aligned and usually follow a different optical path for LSCM. On the contrary, an alignment of a WF microscope is much simpler. For instance, the misalignment of the UV lens in the confocal setup showed extreme values and triggered a high ratio for near-UV images. The realignment of the UV lens was very efficient for the 63× objective but not for the 40×, meaning that one intermediate position has to be chosen or has to be adapted each time for the specific objective. For WF microscopy, the positioning of the dichroic mirror in the filter cubes can influence channel coregistration. SDCM were the most affected systems during our studies, suggesting that their design was more subject to chromatic shift. This may be due to different wavelengths having more chromatic shifts or being less corrected while going through the SDCM head and the microlens array of the disks. When comparing the results of the three modalities for the visible wavelengths, we observed that for LSCM the dispersion is the lowest, together with the mean values. This is coherent with the theory when the pinhole is properly aligned (Stelzer, 1998).

The limit value for the coregistration ratio $r_{exp}/r_{ref}$ is 1. In the cases when $r_{exp} > r_{ref}$, the microscope user should use all available tools for troubleshooting (Table 1). We propose a measuring frequency of twice per year for each objective and for all channels detected by the microscope and in addition as outlined for the field illumination after any major manipulations on the coupling of the light source to the microscope or the light source itself.

To conclude, coregistration shifts of a microscope should be understood by the user, who should be aware of the objective correction type and execute experiments accordingly. This is critical for biologists willing to calculate colocalization percentages between two proteins of interest at the subcellular level, as inaccurate coregistration could lead to strong misinterpretations. For experiments examining whether cells are positive for two different proteins of interest (e.g., in tissue slices), coregistration becomes less critical. Coregistration corrections should be performed preferentially at the hardware level, avoiding as much as possible image postprocessing steps.

### Illumination power stability
#### Background

Spatial- and time-resolved quantification of a labeled cell component involves fluorescence quantification. Such emission quantification relies on the fluorochrome photophysical properties and is linearly related to the dye concentration and the intrinsic power of the light source used under conventional excitation. The light source power should be kept stable to allow quantification studies of the fluorescence signals. Besides sample preparation variability, fluorochrome performance, or environmental parameters such as room temperature, and considering that the source stability is characterized, the microscope can be considered an avoidable source of variance. This illustrates the importance of light source power monitoring, especially for researchers who want to compare images from samples acquired weeks apart. Source power monitoring however requires tools that most researchers may not have available, such as power meters. Some of these tools include a sensor area that fits into the sample position on the stage, enabling power measurements at the sample plane for different wavelengths. Nowadays there is a lack of standard protocols on illumination power measurements that unavoidably increases irreproducibility. Only very recently international initiatives, like the working group 1 of QUAREP-LiMi, worked on the development of standard procedures to measure and standardize illumination power measurements, seeking a consensus from the scientific community (Gaudreault et al., 2021).

Here, we proposed to monitor laser sources fluctuations at four different time scales: warm-up time of the source to reach its stability regime; short-term monitoring for the stability of the excitation chain during the acquisition of a z-stack; mid-term monitoring for intensity fluctuations during a typical experiment duration of 2 h; and long-term monitoring for the global behavior of a light source along its life-time. Comparisons over more extended periods are quite challenging, as other parameters may also affect the experiment (e.g., sample changes, sample preparation variations/modifications, degradation of elements in the optical path). Monitoring such fluctuations is indicative of the setup "health." Compared to past studies (Murray et al., 2007; Stack et al., 2011; Swedlow and Platani, 2002; Zucker, 2006b), we defined two metrics with their tolerances based on experimental values of a wide range of microscopes.

### Materials and methods
#### Sample

While illumination intensity can be measured at any point of the light path, including directly after the light source, we recommend monitoring it close to the objective focal plane. In that way, all parameters that influence the illumination intensity are integrated, and the actual luminous flux received by the sample gets measured. A large set of tools for the evaluation of illumination power stability exist. We used the power meter console (PM100A; Thorlabs) coupled to a microscope slide power photodiode Si sensor, 350–1,100 nm sensitive, designed to measure optical powers from 10 nW–150 mW (S170C; Thorlabs) with a response time of 1 µs. This sensor fits in the microscope slide holders of upright and inverted microscopes and measures the power equivalent to the sample plane. The controller console is connected to the computer by USB. The illumination power can

be recorded over time with the Power Monitor GUI software at a defined sample rate. When we observed the instabilities of the source power over time, we monitored the room temperature with the TSP01 temperature probe (Thorlabs) and recorded it with the TSP01 Application software.

### Acquisition protocol

We chose a low magnification objective (dry 10×, NA 0.3 or 0.4) and cleaned the objective before use in case of any dried dirt (oil, glycerol, or silicone) on the lens. From our experience in imaging facilities, we often observe dry lenses that require thorough cleaning due to the presence of misplaced immersion liquid. The slide power sensor was placed on the stage and the objective was centered on it. The power meter wavelength correction was adjusted to the corresponding laser wavelength we wanted to measure. Background correction was performed in the darkened room. We set the source power at 100% or at 20% for cases where the sensors may be damaged (Gratton and vandeVen, 1995). For LSCM, we set a bleach point in the center of the FOV (stationary laser beam) and launched a time series. If no bleach point mode was available, the highest zoom was applied at the bidirectional mode. We advise using the bleach point mode, if applicable, to avoid modulation/blanking of laser beams by built-in Acousto-Optic Tunable Filters (AOTF), which can occur between the end and the beginning of a new scan line to avoid unnecessary specimen exposure. When available, the blanking function was switched off to avoid measurement disturbance.

To monitor the warm-up time of the light sources, we started recording the illumination power directly after switching on the source. We clearly see that the warm-up time is different for each system and can even differ for each laser type of the same system. The 405-nm diode laser of Fig. S9 A and the 488-nm diode pumped solid state (DPSS) laser of Fig. S9 B show a long warm-up time of nearly 2 h. This long warm-up period was unexpected and might be explained by a laser defect. A 1-h recommended waiting time before starting the stability measurements is a time that is convenient for most cases. It is, however, very important to perform a warm-up measurement of the sources of the microscope every year to define the exact warm-up time before starting the stability measurements.

For short-term stability monitoring, the illumination power was measured every second and recorded for a duration of 5 min. For mid-term stability, we measured values every 30 s for a duration of 2 h. For long-term stability, we recommend a measurement frequency of once a month.

### Metrics

For the short and mid-term time scales, we calculated the SD of the measurements and the stability factor (STAB).

The stability factor is defined as

$$STAB_{power}(\%) = 100 \times \left[ 1 - \left( \frac{P_{max} - P_{min}}{P_{max} + P_{min}} \right) \right],$$

with $P_{max}$ and $P_{min}$ the maximum and minimum recorded power. This stability factor is the one defined in the ISO 21073: 2019 norm (ISO, 2019). Long-term monitoring does not require a specific metric, and the behavior of the source power is

monitored through its lifetime and interpreted as a function of the source type.

We calculated the SD and STAB metrics using Microsoft Excel.

## Results

We studied the power fluctuations for 49 lasers and showed that room temperature, the coupling optics, and the electronic components like the AOTF may influence the stability. While mid-term stability is ensured for the majority of the setups (Fig. 6 A), for some cases, we observed cycle-like fluctuations (lasers of a TIRF setup in Fig. 6 B). The underlying reason was an irregular polarization shift in the optical fiber, influenced by the temperature instability of the room. By performing simultaneous power and temperature measurements (the temperature was measured at the optical fiber level) we showed that, in some cases, there was a correlation between the power and temperature fluctuations (Fig. 6 C).

We calculated the stability metrics and showed that most lasers have high STAB values and low SD as illustrated by the laser lines of Fig. 6 A (blue arrow in Fig. 6 D). The lasers were classified into three categories: diode lasers (including OPLS, i.e., the lasers that can be controlled directly and do not require any modulating device such as an AOTF or AOM), DPSS lasers, and gas lasers (e.g., argon or He-Ne lasers). We observed that the variability and the mean value of the STAB and SD metric were the best for diode lasers, followed by DPSS and gas lasers (Fig. 6 E). Based on the experimental data, we propose a tolerance value for STAB of 97%, below which the source is considered unstable (dark green zone of Fig. 6 D and Table 1). A normalized intensity STD of 0.02 defines the limit above which fluctuations are considered too high for short-term and mid-term time-scaled experiments. We observed that some lasers have lower STAB (<97%) and higher SD (>0.02), such as the 488 nm argon laser line of Fig. S9 A (magenta arrow in Fig. 6 D). Extreme cases like the 561 and 642 nm laser lines of Fig. 6 B (red arrows of Fig. 6 D) show very low STAB and high SD.

Short-term analysis of a subset of these lasers gave similar results (Fig. S10). The behavior of the lasers of the same setups (blue and closed orange arrows) is similar, and this suggests that short-time scale stability issues are likely linked to the stability of common electronic and/or optical components that all laser light encounters until reaching the objective plane, like an optical fiber polarization effect for example, rather than laser-related individual issues.

We studied the long-term stability of the laser sources and showed that the stability is highly related to laser beam alignment and laser replacements. For instance, the monitoring of a LSCM Leica SP5 over 9 yr showed that service technician visits or whole laser or optic component (optical fiber) replacements had a positive effect on illumination intensities (Fig. 6 F). Similar measurements on a total internal reflection fluorescence (TIRF) setup, not covered by after-sales service, showed that regular beam alignment of the laser sources results in improved long-term laser stability (Fig. S11 A). For arc-lamp light sources, we observed that aging

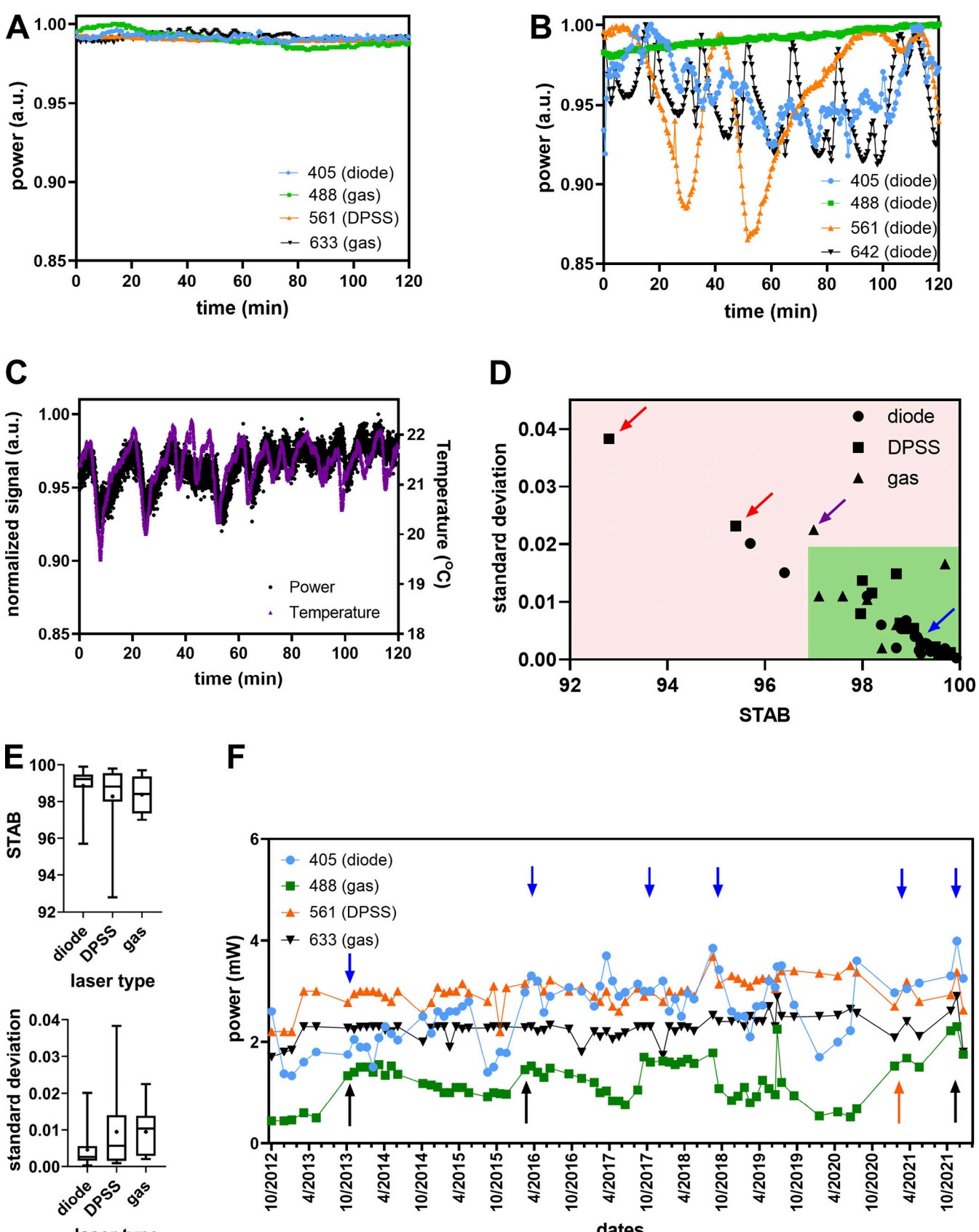

Figure 6. **Illumination power stability data summary. (A)** Power stability over 2 h for the four laser sources of a confocal microscope LSM 880 (Carl Zeiss Microscopy GmbH). The power value is normalized to 1 by dividing each value by the maximum power value. **(B)** Power stability over 2 h of the four lasers of a

TIRF microscope. **(C)** Laser power (black dot) and room temperature (magenta square) variation over time (2 h) of the 561 nm laser of B. The minimum and maximum value of temperature was 19.5 and 22°C respectively. **(D)** Laser power mid-term stability STAB versus SD of normalized to 1 (max value) laser power values over 2 h. The blue arrow points to the 488 nm laser of A, the magenta arrow points to the 488 nm laser of Fig. S9 A, and the red arrows point to the 561 and 642 nm lasers of B. The dark green area highlights the acceptable values. **(E)** Statistical analysis of D of the STAB and SD metrics. Mean values are represented with a cross, SD with boxes and min/max with whiskers. Data points are 24, 15, and 9 for diode, DPSS, and gas lasers respectively. **(F)** Illumination laser power monitoring over a microscope (Leica SP5) lifetime (long-term stability). Intensities of a 405 nm diode, 488 nm argon, and 633 nm He-Ne gas lasers and a 561 nm DPSS laser were recorded for 9 yr. The 488 nm laser was replaced three times (black arrows). The orange arrow indicates when the optical coupling fiber of the 488 nm laser to the microscope was replaced. The manufacturer service visits are indicated with blue arrows and are often associated with power increases as the system was realigned.

influences the source power, although this is highly related to the bulb type (Fig. S11 B).

### Discussion

We showed, by defining two simple metrics, that once the microscope system is warmed up, any further fluctuations are likely to arise because of issues such as laser aging (especially for gas lasers), inadequate polarization stability, or misalignment of elements such as optical fibers (Table 1). Determining the light source warming time is not trivial as some sources can show a warm-up period of some hours (Fig. S9).

We focused our studies mainly on laser sources. Most measurements involving solid-state lasers were within limits as soon as the source was warmed up. External source parameters, like insufficient fiber coupling or temperature variations, influenced their stability. Gas lasers, typically aging argon laser tubes, showed strong power fluctuations.

During long-term monitoring, we observed that the aging of the source, optics alignment stability, fiber coupling, or some damage of optical elements (e.g., deterioration of an AOTF by a 405-nm laser line) provoked stability fluctuations. Hence, whenever fluctuations are observed, we recommend, if accessible, a supplementary measurement directly at the laser output to check the stability of the source before going through all the optical elements. However, this is only possible in some cases and not compatible with most closed commercial systems. Long-term monitoring is instrumental in deciding if a laser needs to be replaced. The above tolerance values may need to be adjusted depending on the laser type used. Gas lasers tend to show high mid-time scale fluctuations, and their aging is often associated with a global intensity decrease (Fig. S9). A drop by a factor of three is a good indication that the laser needs to be replaced or the tube current has to be adjusted. Nevertheless, a third of the original value is usually still sufficient for most experiments (excluding FRAP). For solid-state lasers, a long-term power decrease is most often due to some misalignments.

We propose measuring the laser stability of a microscope system twice per year. For experiments that need absolute and not relative quantification, the measurement of the illumination power before each experiment is necessary.

We defined the experimental limit values of our metrics. Of course, some microscopy experiments do not require that degree of source stability (e.g., nuclei counting). Nonetheless, users have to be aware of the fluctuations and monitor the source power, especially if quantification is involved. A drop in the specific labeling signal can be due to the source power instability or aging of the source and, hence, unrelated to the sample itself.

In practice, users aiming at reproducing a quantitative experiment performed a week or a month ago should always pay attention to laser stability before comparing the datasets.

### Stage drift and positioning repeatability
#### Background

Stage stability is of great importance when performing multidimensional acquisitions involving multiple planes, positions, and time points. The precision of the motorized components may significantly impact the conclusions of the experiment as they may introduce inaccuracies in measurements. The specifications of the mechanical components and more precisely of the microscope stage need to be verified and validated through appropriate controls. For microscopy experiments, the stage stability can be inherent to the stage characteristics and can also often be influenced by external parameters. During time-lapse live-cell imaging, multiposition imaging experiments, or recordings of large z-stacks (hundreds of planes), instabilities can result in a pseudo movement of the observed structures over time and lead to false conclusions. The setup can be affected by lateral (xy) and axial drift (change of the defined observation z focal plane). Importantly, when using high NA objectives, small axial drifts are likely to be detected due to the shallow depth of field (Inoué, 2006).

The reasons for drift can be complex and caused by room temperature variations, mechanical thermal expansions, a thermal gradient in a temperature control chamber, mechanical movements along the z axis, the stage itself, the sample holder, or the sample in the holder or even vibrations (Burglin, 2000; Nikon MicroscopyU; Price and Gough, 1994). Using custom-made bead slides, we characterized the 3D drift of various microscope setups by defining a stabilization time and calculating the 3D drift rate before and after stabilization. We experimentally defined tolerance values for these three metrics.

For multiposition experiments, a key parameter is the stage repeatability, i.e., the system's ability to reposition to the same point. Experiments such as those tracking moving objects in multiple positions over time are greatly impacted by the precision of the lateral movement of the stage. The ideal stage should have high accuracy and repeatability. The two parameters are important but independent. Systematic errors that influence stage accuracy can be taken into account and compensated for, but after such corrections, repeatability will be the ultimate limit of the device. We calculated the repeatability and defined our metric as the SD of the position coordinates during successive moves to a set of predefined positions. Protocols exist in

literature about these measurements, but unfortunately, the stage performance is often considered isolated (ISO, 2014; Serge et al., 2016) and not as a part of a whole system, something that we take into account. We compared the acquired experimental values to the stage specifications found in the manufacturer's datasheets.

## Materials and methods
### Stage drift
*Sample.* The custom-made fluorescent slide with the 1 µm diameter beads or the 4 µm beads was used (the same slide used for coregistration studies).

*Acquisition protocol.* For both stage drift and repeatability acquisitions, the microscopes were set up on active optical tables, as for all other measurements in this study. We checked the flatness of the stages by using spirit levels for WF and SDCM and by imaging a mirror in reflection mode for LSCM. The horizontality of the stage can be adjusted by inserting layers of paper or tape between the mounting frame and the stage (You et al., 2021).

The stage drift characterization is a fairly straightforward procedure to follow. A fluorescent bead is convenient, but bright-field observation of various features, like dust as a sample, is possible as well, implying that the dust is not located on the faceplate of the camera or in the optical path. This condition can be easily verified: in the presence of dust in the FOV of the microscope in bright-field mode, if the dust moves with the *xy* stage, then the dust is on the sample. Turning the camera slightly around its connecting axis in live mode allows knowing if the dust is in the optical path (in the live-image streaming the dust moves) or on the camera (in the live-image streaming the dust does not move). One should pay attention to stitching applications, where a shift of the camera angle with the *xy* stage directions requires a new calibration.

We placed the slide on the stage and the whole microscope setup was switched on at least 30 min before the experiment to equilibrate the temperature between the sample and the microscope to reduce sample drift. We centered one bead in the illumination field and adjusted the *z*-position on focus. If present, it can be very useful to activate hardware and software focus corrections for the calibration of experiments. However, to properly monitor the *xy* behavior of stages and the *z* stability, the *z* correction has to be deactivated to allow raw drift evaluation and thus maintain a sensitivity to the environmental parameters, which must be taken into account. We started *z*-stack recordings over time at a range of 15 µm. We used 60/63× high NA oil immersion objectives to stay closer to experimental conditions where a high numerical aperture makes it possible to precisely detect drifts in the three dimensions. We evaluated the long-term drift (overnight acquisitions) for a time-lapse of 15 h with an acquisition frequency every 10 min at room temperature set at 20–22°C (depending on the microscope room the defined temperature values can be slightly different).

*Metrics.* We measured (1) the stabilization time $\tau_{stab}$ of the stage, (2) the mean velocity $V_b$ of the beads in nm/min in *xyz* before the stabilization time, and (3) the subsequent velocity $V_a$ after this time. The stabilization time is the one required for the stabilization of the "sample on stage" system. This duration is measured from the start of the acquisition until the moment when any *xyz* drift is lower than a distance corresponding to the spatial resolution of the system.

*Analysis.* We used the TrackMate plugin of Fiji (Tinevez et al., 2017). The TrackMate plugin is widely used for single particle tracking purposes. The plugin finds the center of mass of the bead for each time-point in the three dimensions and it calculates the mean velocity of the bead and the 3D displacement. We exported the bead *xyz* displacement values and traced them over time. This plot is convenient to find the stabilization time. Finally, we recalculated the mean velocity for the time values before and after the bead stabilization.

*Biological imaging.* We used osteosarcoma cells (ATCC 143 B - CRL-8303) expressing the construction CD63phluorin/scarlet (Addgene pLenti-pHluorin_M153R-CD63-mScarlet/Plasmid #172118). Images were acquired with a Nikon Ti2 microscope using TIRF illumination, a Plan-Apo 100×/1.45 objective, and a Prime 95B sCMOS camera (Photometrics).

### Stage positioning repeatability
*Sample.* Any samples that can be well-distinguished in fluorescence or transmitted mode would work for this kind of measurement. In our cases, we used the 1- or 4-µm fluorescent multicolor beads of the self-made bead slide (the same slide as used for the coregistration studies).

*Acquisition protocol.* Microscope stage specifications include maximum travel range, (maximum) speed, repeatability, or accuracy. We focused on the repeatability and determined the repositioning of the *xy* stage movement by moving between multiple fixed points over time and observing the variability of the position precision. Repeatability can be performed unidirectionally (the stage returns to a given point with only one axis movement along *x* or *y*) or bidirectionally (the stage returns to a reference point coming from a previous point, involving *x* and *y* movements). The above diagonal movements are the displacements that could result in the less optimal repositioning since they involve movement in both the *x* and *y* axes.

For our acquisition protocol, the stage was first initialized (whenever this option was available in the software). The microscope with all its components was switched on at least 30 min before the experiment and 10× or 20× dry objectives were used to avoid any oil pressure/surface tension influence. The sample was firmly fixed on the mounting frame of the stage (usually the stages offer a clipping mode that fixes the mounting frames and slides securely in place). To avoid drift, the setup stayed at this state for the stabilization time found during drift characterization. A reference stage position, where a fluorescent bead sits in the center of the FOV, was recorded as $P_{ref}$. The stage was then shifted 3 mm away in both *x* and *y*-axes toward the top-right direction (i.e., a maximum of 4.2 mm diagonal movement). This position (referred to as top/right position P1) was recorded. The stage was then returned to the initial bead position, before being further shifted away in the opposite position along the *x* axis (bottom/right, 4.2 mm away from the bead position, referred to as P2). Opposite positions along the *y* axis were also recorded (referred to as P3 and P4), resulting in a total diagonal movement of 8.4 mm. Then the acquisition software of the microscope was

set to acquire a 20-cycle time-lapse of the nine positions ($P_{ref}$-P1-$P_{ref}$-P2-$P_{ref}$-P3-$P_{ref}$-P4-$P_{ref}$), as shown in Fig. 1 E.

We chose a 4.2-mm diagonal distance as most often imaging of a coverslip/slide mounted sample does not involve more than 3 to 5 mm stage movements. This value has to be adapted depending on the application the setup is intended for. To fully characterize a stage for all kinds of applications, we performed repeatability measurements for two more distances along the $x$ and $y$ axes, for 0.3 and 30 mm when achievable. The latter distance corresponds to experiments on histological slides or multiwell plates. All experiments were carried out at room temperature set at 20–22°C (depending on the microscope room the defined temperature values can be slightly different).

*Metrics.* The $x$ and $y$ coordinates of the bead reference positions are calculated for each cycle (values in μm). The SDs $\sigma_x$ and $\sigma_y$ of the measurements give the repeatability values along the $x$ and $y$ axes. The values are then compared to the repeatability value given by the manufacturer. This value usually appears in the datasheet of the stage which is a document containing all parameter values which are measured before the purchase for that item ideally or for this item category. This information is useful to properly characterize the device at installation and over time.

*Analysis.* Bead positions were analyzed using the Fiji tracking plugin TrackMate (Tinevez et al., 2017). We then calculated the SDs of the measurements. Fig. 8 A shows a time projection of the reposition experiment for a bead on the left. Cyan and magenta represent the two extreme positions of the bead. On the right, the calculated trajectory is illustrated with a red line.

*Biological imaging.* The yeast cells were W303 strain, named ySP16782. They expressed mCherry–septin protein (mCherry-cdc3::URA3). For testing the stage repositioning repeatability, a SDCM (SpinSR10, Olympus) was used with an ASI $xyz$ stage. The acquisition was recorded through a 60×/1.5 UPLAPO objective (Olympus) and a Fusion BT camera (Hamamatsu Photonics).

## Results
### Stage drift
We characterized first qualitatively the drift by observing the time projection of the maximum intensity 3D projection of a typical bead track (Fig. 7 A). The 3D displacement (increased $z$ drift compared to $xy$) decreases over time leading to stabilization (Fig. 7, B and C). The $z$-axis drift was lower than the axial resolution of the system, so it can be considered negligible, for standard microscopy experiments.

The stabilization time $\tau_{stab}$ (time from the start of the acquisition until the $xyz$ drift is lower than a distance corresponding to the spatial resolution) in Fig. 7 B was 105 min. Before stabilization, the mean 3D speed $V_b$ was 56 nm/min, and after stabilization $V_a$ = 8 nm/min. The $\tau_{stab}$ depends mainly on environmental conditions, particularly related to temperature variations that can imply large drifts (Mason and Green, 1975).

We plotted the distribution of the 3D drift speed before and after the calculated stabilization times of 20 stages. When a short stabilization time (<45 min) is observed, $V_b$ and $V_a$ values are close. For longer stabilizations (45 min < τ < 120 min or τ > 120 min), the two-speed values are uncorrelated, showing that

once the stabilization is achieved under our criterion, the whole system is stable. Finally, for one of the 20 tested setups, we found no stabilization during the complete 15-h time-lapse period.

We experimentally classified drift behavior into three categories according to the specificity of experiments and the need for precision (Table 2). When $V_b$ and $V_a$ values are under 15 nm/min and close to each other, the drift is negligible (high precision experiments, e.g., super-resolution purposes). Accumulated over time, this amount of drift may be problematic for localization microscopy (acquisition time of minutes or hours). When 45 min < $\tau_{stab}$ < 120 min and 15 nm/min < $V_b$, $V_a$ < 50 nm/min, drift can be considered acceptable (e.g., time-lapses at the cell scale). Any other drift parameter combination should be earnestly examined because it can lead to failed experiments, especially when the focus is lost.

A representative example of stage drift in biological imaging is shown in Fig. 7 E. Living cells, labeled with CD63PhLuorin (CD63 is a membrane protein and PhLuorin a GFP variant), were imaged for 600 s at a time interval of 1 s. As shown in the overlay image of Fig. 7 F, which is a zoom region of Fig. 7 E, we observed a mechanical drift between the first (image in cyan) and last (image in magenta) acquired frames. The drift occurs along one single axis and is constant during the acquisitions (no stabilization achieved, and the drift velocity is 400 nm/min, in the critical range of Table 2). The reason for that drift was a damaged joystick (when at "no move" position, the stage kept moving along one axis) of the stage controller and not the stage itself.

Additionally, we investigated the drift speed fluctuations over days. Three different measurements on the same system were performed on three consecutive days. The drift values were very close and the velocity dispersion was low. However, during a month or even a year, the drift of a system can change significantly if environmental parameters are not stable, like the room temperature or even the temperature flow in the room.

### Stage positioning repeatability
We tested 25 different stages on 21 microscopes. Stage suppliers were Märzhäuser, ASI, Nikon, and Olympus, and the majority of the stages had linear two-stepper motors and 5 stages had linear encoders. The variability for the 3 mm $xy$-stage displacements was always below 0.4 μm (Fig. 8 B). This value is within the specifications of the manufacturers, which are usually lesser than 1 μm for stepper motors and 0.7 μm for linear encoders. A distribution analysis highlights a population with values below 0.2 μm at both axes (green zone in Fig. 8 B). For most multiposition experiments on tracking fluorescence localization within a living cell, the 0.2 μm value can be a critical one. It is near the $xy$-resolution limit for high NA objectives, and we define it as our tolerance value.

We examined some parameters that could influence repeatability. The stage speed, for the ranges that we tested (7–20 mm/s) did not significantly affect the values (Fig. S12, A and B). The stage acceleration did not affect the repeatability as well (Fig. S12, C and D). The waiting time was not the same for all experiments as it cannot always be controlled and adjusted via the

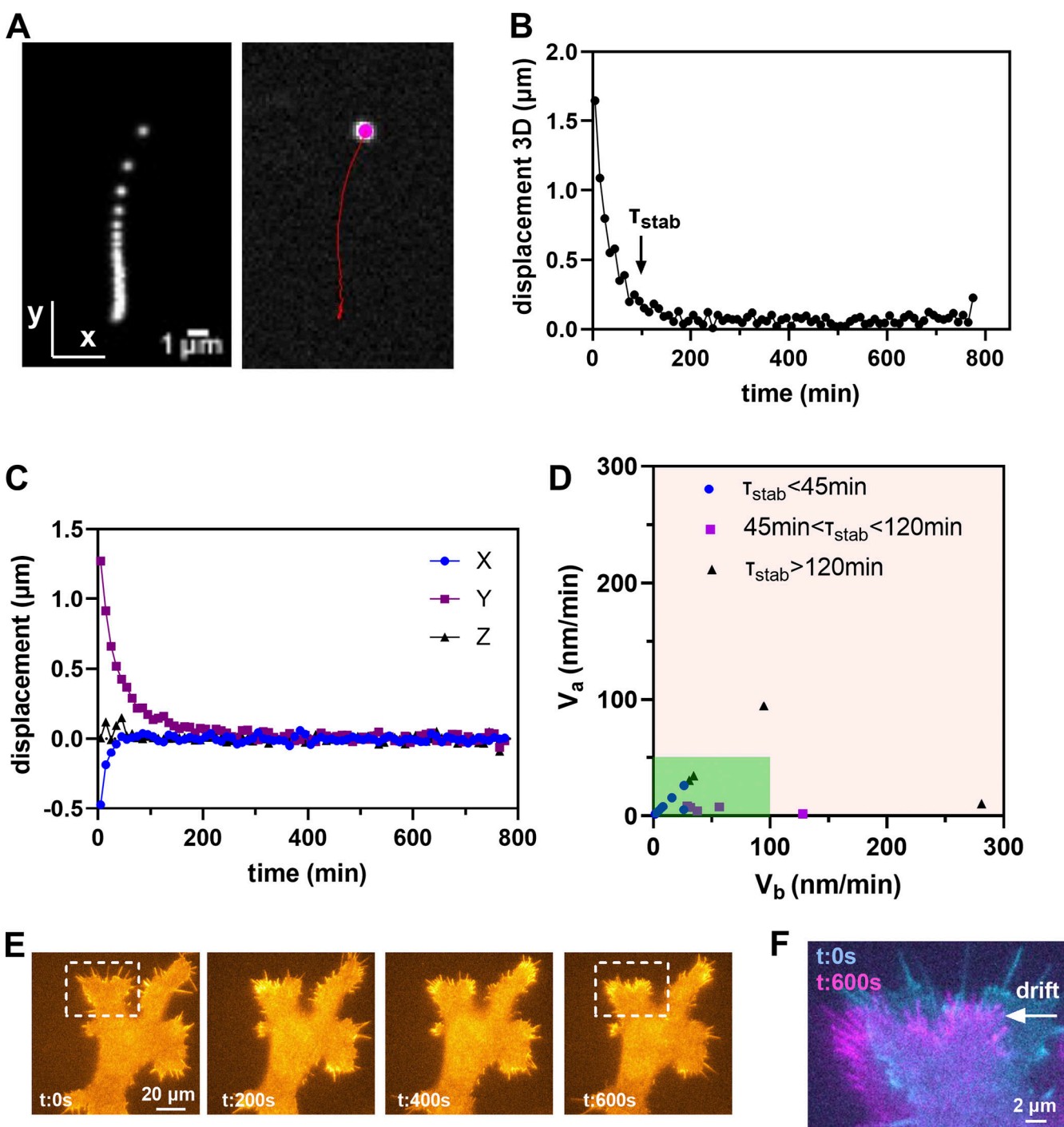

Figure 7. **Stage drift data summary. (A)** Max Z projection of the time projection of a 15 h time-lapse of a fluorescent bead, and trajectory of the bead, with the initial segmented bead position shown in magenta. **(B)** Global 3D displacement over time. The arrow indicates the calculated stabilization time $\tau_{stab}$. **(C)** Relative displacement along the $xyz$ axes. **(D)** Calculated velocity before ($V_b$) and after ($V_a$) stabilization of different microscope stages grouped in three categories upon $\tau_{stab}$. The dark green zone highlights the acceptable values. **(E)** $xy$ time lapse over 600 s of living cells labeled with CD63PhLuorin (membrane labeling). **(F)** Overlay of the initial $t = 0$ (cyan) and final $t = 600$ s (magenta) frame of the series of E to highlight the relative drift along the $x$ axis.

acquisition software. Typical waiting times were in the range of 10–300 ms. The proper fixing of the sample is important for repeatability measurements, but it depends very much on the used insert type for each microscope, which is quite diverse. We then examined the reproducibility of the measurements by performing acquisitions for five consecutive times on five

different stepper motor stages (Fig. 8 C). We observed that the variability of the results was stage dependent (stages 2 and 4 showed lower variability than the rest of the stages in Fig. 8 C).

When we compared two identical model stages on the same setup (ASI stages with linear encoders), we observed different repositioning values. We identified that the first stage, which had a

**Table 2. Drift tolerance values**

| | Standard | Acceptable | Critical |
|---|---|---|---|
| $\tau_{stab}$: stabilization time | <45 min | (45–120) min | >120 min |
| $V_b$: mean velocity before stabilization | <15 nm/min | (15–100) nm/min | >100 nm/min |
| $V_a$: mean velocity after stabilization | <15 nm/min | (15–50) nm/min | >50 nm/min |

The drift is negligible when the velocities before and after stabilization are identical and kept under 15 nm/min and the stabilization time $\tau_{stab}$ is lower than 45 min. The drift becomes acceptable for velocities before and after stabilization in the range of 15–100 and 15–50 nm/min respectively and $45 < \tau_{stab} < 120$ min. For higher values the drift is considered critical and not acceptable. These values are experimental and serve to guide microscopy users to apprehend the orders of magnitude for their experiments.

z-piezo issue and had to be replaced, showed $\sigma_x = 0.05$ µm and $\sigma_y = 0.27$ µm (capped line of Fig. 8 B). The replacement stage showed much better results ($\sigma = 0.03$ µm for both xy), far under the system spatial resolution (arrow of Fig. 8 B). This example illustrates the diversity in quality that can occur while purchasing such devices.

We next examined the influence of the stage displacement distance using three different values (0.3, 3, and 30 mm) for both axes for five different stages equipped with two stepper motors from the same manufacturer (Märzhäuser) and with similar datasheets. Although there is a high variability (measurements not taken on the same microscope and the stages were not exactly the same model), we can distinguish that the repeatability deteriorated for longer displacements (Fig. 8 D). The 30-mm xy displacements correspond to typical distances used during multiposition experiments in a multiwell plate.

We examined the influence of position repeatability on biological imaging. Fig. 8 E shows a maximum intensity projection of a 3D time-lapse of a single position of two zoomed yeast cells under division. The time series lasts 30 min with an acquisition time interval of 2 min. This series of frames belongs to a 6-position time-lapse experiment using a 60×/1.4 objective with a SDCM. *Saccharomyces cerevisiae* yeast cells expressed mCherry coupled to the septin protein Cdc3 (the protein forms various filaments containing structures such as rings, as we see in Fig. 8 E when the two cells are divided). We clearly see that the repeatability is not good, as some frames are displaced along the y direction. In Fig. 8 F, we decided to visualize the frames of the 12-min time point (in cyan) and 14 min (in magenta) and show the vertical drift that is in the order of >1 µm. For the whole 30 min time-lapse, the SD of the position repeatability is calculated at 0.07 and 0.25 µm for the x and y axes respectively. The 0.25 µm value is out of the defined limit value and makes it impossible to treat the images. When we characterized the repeatability of the same stage without sample and with the correspondent protocol (20× dry objective), the values of the SD were 0.11 and 0.15 µm on the x and y axes, respectively.

A method to overcome the stage repeatability issue is to post-process the images. We used StackReg ImageJ plugin (Thevenaz et al., 1998) and corrected the drift (bottom row of Fig. 8, E and F).

### Discussion
### Stage drift

The stage drift measurements are simple to perform, but their interpretation can be complex. Drift measurements only make sense if environmental conditions (e.g., temperature, humidity, air flow, and warming time of the stage) are kept stable. As external changes (e.g., period of 1 yr) influence the drift results, time monitoring of the drift is necessary. We applied a straightforward protocol to measure the drift during overnight experiments. We defined three metrics and proposed their tolerance values (Table 2). Here again, these are values that we recommend and should not be considered as an absolute reference. We propose to measure stage drift once per year in the frame of preventive maintenance. For super-resolution microscopy, the frequency should be adapted to the modality.

Within the framework of a given experiment, the user must have general control over the orders of magnitude to be quantified. For example, if distances between two fluorescent signals in two different channels are measured, the relevant time domain is the acquisition delay between both channels, especially if the two signal sources drift together.

Our results showed that a stabilization time is mandatory before starting the acquisition. Furthermore, it is very important to consider that once the sample is fixed on the stage, it becomes an integral part of this device. To preserve the intrinsic heat of the stage, it is possible to leave the controller under permanent voltage, avoiding additional drift due to stage heating cycles. A z drift can also arise due to bad tightening of its drive axis, which is possible on aging microscopes (Price and Gough, 1994). For biological imaging, minimizing the stage drift is critical, especially for time-lapse experiments. As shown with the biological example of Fig. 7, E and F, drift can be problematic for live cell imaging, even for short time scales (10 min in our case). Effective software solutions exist to overcome the drifts after the acquisitions (e.g., StackReg [Thevenaz et al., 1998]), but it is always preferable to understand and master the drifts at the hardware level (Table 1 for troubleshooting).

### Positioning repeatability

Stage repeatability is directly linked to stage drift. When parameters involved in drift are controlled and the drift is characterized, the ability of the stage to reposition at the same xy location can be measured. Monitoring repeatability must then be performed after the stage stabilization time and the sample should be well secured to avoid a sample-associated source of drift. Parameters that may influence the repeatability should be controlled when possible (often the acquisition software does not give access to these parameters). We propose to measure positioning repeatability once per year.

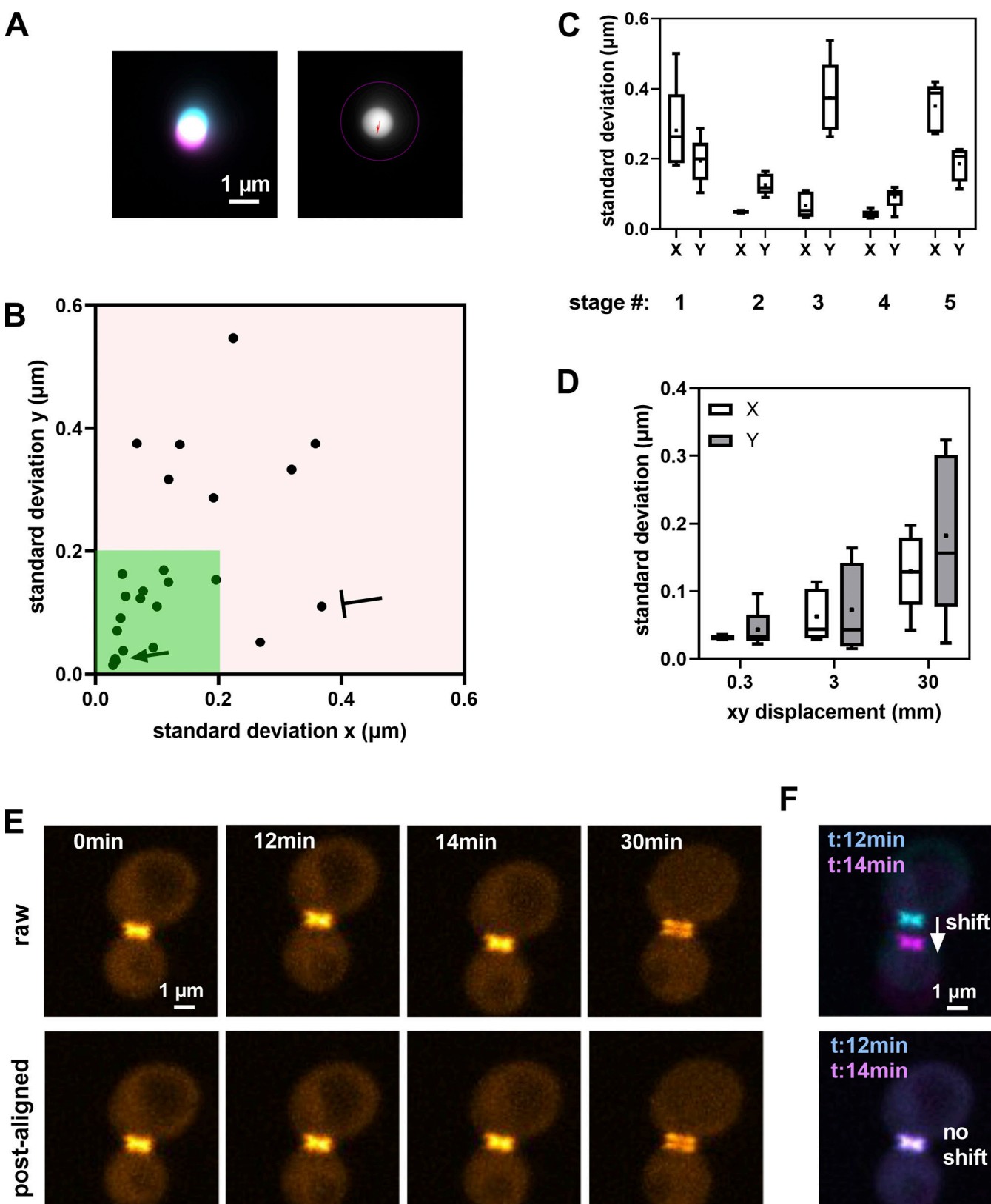

Figure 8. **Stage positioning repeatability data summary. (A)** Left: Time projection of the repeatability experiment for a fluorescent bead. Cyan and magenta represent the two extreme positions of the bead. Right: The calculated trajectory is shown with a red line. **(B)** Bidirectional stage repeatability for 3 mm displacements in both *xy* axes (4.2 mm diagonal). The arrow and capped line show the results of two linear stages (same model with encoders) tested on the same microscope stand. The dark green zone highlights the acceptable values. **(C)** Box plots summarizing the reproducibility in time (five different measurements per stage) of stage *xy* repeatability for five stages equipped with two stepper motors. **(D)** Repeatability dispersion increases for the five stages

of C with the *xy* displacement. Whiskers indicate the min/max values of the box plots of C and D and dots indicate the mean values. **(E)** Raw and postprocessed aligned max intensity Z projection of 30 min time-lapse of a yeast cell under division expressing mCherry-septin Cdc3 protein. The experiment included six positions with 2 min time intervals, here we highlight the time series of a single position. **(F)** Overlay of the 12 min (cyan) and 14 min (magenta) timepoint of E showing the mechanical drift at the raw case and the zero drift at the postprocessed series.

Our experimental repeatability values were always within the manufacturers' specifications. We defined a tolerance value that is close to the spatial resolution of the system. For multi-position imaging, the user can expect positioning repeatability to be accurate at the scale of a focal volume, although this precision is not necessary for all experiments. We found that repositioning is less precise when the stage displacements increase.

Repeatability can be improved when a backlash correction (backlash is the systematic error created by lost motion in the drive mechanism that appears when changing direction) is applied or when a waiting time is added and/or when the stage has linear encoders (Stuurman and Thorn, 2015).

Last but not least, our measurements were performed considering the stage as part of the microscope and under "real" conditions, which is not always in accordance with the manufacturers' protocols and datasheet values where the stage is considered as an isolated item. In the latter case, the stage is not subjected to the conditions of temperature change in the context of live cell imaging or drifts. The way to measure stage repeatability can also influence the results. For instance, when using a dry objective, we measure the stage position repeatability. If we want to approach the biological imaging conditions, immersion objectives can deteriorate the repeatability results. In the case of the example of Fig. 8, E and F where a 60× oil immersion objective was used, the repeatability values are out of the defined experimental limit values. The microscopy acquisitions can serve only for qualitative studies (observe cell division and creation of the septin rings in that case) but not for quantifications (study the size and the structure of the septin ring during cell division or study its colocalization with other proteins for two-color imaging). In that case, postprocessing correction is necessary to correct the drifts.

### Camera noise
#### Background
The last element in the acquisition chain of a light microscope is the detector. It can be a 2D array sensor chip (mainly CCD, EMCCD, or sCMOS cameras) or a point detector (mainly PMT, APD, HyD, or GaAsP). By construction, all these devices generate different noises, which can influence the contrast of the generated image, especially for the detection of weak signals, and thus determine whether the signal will be detected (Stelzer, 1998). The quality of an image depends mainly on the Nyquist sampling of the sample illuminated with the least possible luminous flux while obtaining as much contrast as possible. It is thus important to monitor the contribution of detector noises (Deagle et al., 2017). Due to the different detector technologies, it is particularly complex to assess a single protocol that would apply to all cases. In this study, we chose to focus on array sensor chips (CCD, EMCCD, and especially sCMOS cameras). Metrics

for single-point detectors were proposed in the past (Murray, 2013; Zucker and Price, 1999). The European Machine Vision Association has published standards on noise characterization of cameras (EMVA Standard 1288, 2010). In this study, we proposed a simple protocol to measure read noise-related metrics, including dark offset and dark signal non-uniformity (DSNU) values that can be crucial for further fine processing of the data (Watanabe et al., 2017).

Compared to previous studies (Ferrand et al., 2019; Mullikin et al., 1994; Watanabe et al., 2017; Zucker, 2006b; Zucker and Price, 1999), we performed measurements among a wide variety of CCD, EMCCD, and sCMOS cameras for which simple noise metrics are missing since these kinds of cameras are quite recent. These past studies also evaluate noises within the framework of a complete detection system where the whole optical path influences the captured signal. Although this method is beneficial and necessary for microscopy experiments when using actual samples, it is very dependent on microscope components. The influence of the upstream elements may hinder the ability to characterize the sensor accurately. We thus evaluated the noise of the sensor as an isolated element to have some easily reproducible values. With the proposed protocol, we do not aim to calibrate a sensor, but simply to characterize it and monitor its noise over time after the installation day.

### Materials and methods
#### Sample
No sample is needed for this measurement.

#### Acquisition protocol
We monitored three parameters: (1) the read noise of the camera sensor, (2) the dark offset value, and (3) the DSNU. The basic unit used by manufacturers to characterize the different sources of noise is not the gray level but the electron. The cameras are characterized by variable photon/electron conversion factors, so the use of the electron unit allows more direct comparison between cameras.

(1) The read noise is created during the conversion of photons to electrons and originates mainly from on-chip amplifiers. It is measured in dark conditions to avoid noise contributions due to uncertainties of photon arrival (Janesick, 2007). Read noise is often a technical bottleneck when a low signal is to be quantified. This sensor parameter is usually provided in the manufacturer's datasheet, mainly as "Read out noise," in electron RMS (root mean square) for CCD cameras or in electron med (median value) for sCMOS. Each pixel in CCD and sCMOS cameras accumulates a charge proportional to the intensity of the localized illumination. At the end of the exposure time, the CCD camera sequentially transfers the charge of each pixel to a common bus. The conversion of the charge into voltage takes place in the printed circuit of the camera and the read noise is therefore constant across all pixels. In a sCMOS chip, the charge-to-voltage

conversion takes place in each individual pixel, resulting in pixel-to-pixel variation of the read noise. This difference in the architecture of the reading method results in lower average photodiode read noise for sCMOS compared to CCD cameras, while for sCMOS, read noise is no longer constant across the chip (Brouk et al., 2010; Hain et al., 2007; Lambert and Waters, 2014; Mandracchia et al., 2020). This must be taken into consideration when using image analysis algorithms (e.g., single-molecule localization) on images acquired with sCMOS cameras (Babcock et al., 2019; Huang et al., 2013). Also, some quantitative image data analysis, like maximum likelihood estimation methods, requires individual pixel characteristics, such as variation of the read noise (Huang et al., 2013). In our protocol, the read noise is measured by the fluctuation of the value of a pixel along a temporal acquisition of 100 images, without any light influence. Read noise is then considered as a temporal noise. While the quantity of 100 frames may seem overestimated for CCD read noise measurement, it is a fair compromise for sCMOS read noise (obviously not to characterize every pixel's attributes; Huang et al., 2013), and DSNU measurements, as explained below. An open debate in the field is one in which noise needs to be taken into account, RMS or median. Our protocol calculates both. According to theory, for CCD and EMCCD cameras, the RMS and median model gives similar read noise values as the readout in these cameras is serial, and an identical read noise across the whole chip is expected. In sCMOS sensors, the noise distribution is not symmetrical. Thus, both approximations, Gaussian (RMS) and median, are justified. Our interest here is to follow the calculated values over time and compare those with the ones in the camera datasheet. We recommend comparing with the value that is given in the datasheet, and in any case contact the manufacturer to request missing values. The median value provides information about the general magnitude of the noise. Together with the RMS value, they indicate the spread of the read noise distribution.

(2) The dark offset value is a baseline arbitrarily adjusted by the manufacturer to allow access to the complete read noise fluctuation and thus avoid negative gray values (Lambert and Waters, 2014).

(3) The DSNU (also called Offset Fixed Pattern Noise) is measured to establish the stability and uniformity of the camera offset. It refers to the fluctuation between pixel values of the chip under the same dark condition, is measured in electrons and is the SD of all pixels' dark offset (Long et al., 2014). As seen previously, the gain of the amplifiers may vary from pixel to pixel (and column to column) and thus contribute to the DSNU. This noise is often observed to a greater extent in sCMOS cameras than in CCD cameras, although recent improvements in sCMOS design tend to reduce fixed pattern noise (El Gamal et al., 1998; Snoeij et al., 2006). EMCCDs often have additional noise distributed over the image sensor chip area (Rieger and Stallinga, 2014). For precise DSNU evaluation, thousands of frames should be acquired to eliminate the read noise. To define the optimal dataset size, we introduced a quality criterion that compares the DSNU obtained with datasets of different sizes. A threshold of 10% yielded an optimal dataset size of 100 frames (i.e., the DSNU value difference obtained with larger datasets is <10% of the DSNU value calculated with

100 frames, Fig. S14 D). Hence, we recommend using a 100 dark frames dataset for DSNU evaluation.

To resume, read noise is an inherent property of camera sensors, regardless of whether or not light reaches it. Thus, the measurements were carried out without excitation light and sample. When possible, the camera shutter was closed to avoid any unwanted light contributions. When available, the cameras were cooled according to the manufacturer's recommendations (for sCMOS 0 or −10°C, EMCCDs −70 to −80°C). The acquisition software was set to acquire a 100-cycle time-lapse. The sensor exposure time was set to short values (e.g., 10 ms) to avoid DSNU contribution, and pixel binning was not used (Orzanowski and Wachulak, 2018).

For this study, the CCD cameras were all Coolsnap HQ2 (Photometrics); the EMCCD cameras were iXon 888, iXon 897 (Andor–Oxford Instruments) and Evolve512 (Photometrics); the sCMOS cameras were Zyla 4.2 or 5.5 (Andor–Oxford Instruments); Orca Flash 4 LT, LT+, V2, V3, Fusion BT (Hamamatsu); Prime 95B (Photometrics); and Edge 4.2 (PCO). Most often we measured the read noise for the different readout modes of the cameras (30, 100, 200, or 500 MHz for sCMOS, 10 and 17 or 30 MHz for EMCCD, 10 and 20 MHz for CCD). For EMCCD cameras, we set the EM gain at the minimal value of 0, which is not the common condition that these cameras are used for, but this setting allows comparison with the manufacturers' values. When available, we also acquired data with blemish corrections to replace cold and hot pixels (defective pixels not influenced by incoming light, with constant zero value and saturated value of the sensor dynamic, respectively) with the average intensity of surrounding pixels (McNally et al., 1999). This also must be taken into consideration when using image analysis algorithms (e.g., likelihood-based analysis methods) since in this case, noise statistics are violated (Huang et al., 2013).

**Analysis**

The data analysis was performed using the MetroloJ_QC plugin. The main workflow is shown in Fig. S13. The plugin generates both intensity average and SD projections across the 100 frames. The average intensity projection was used to calculate offset and DSNU values. The offset is the mean average intensity across all pixels of the average intensity projection. The DSNU is the SD of the pixel intensities of the mean average intensity projection multiplied by the electron to analog-to-digital units (ADU) conversion factor (or gain), usually provided in the manufacturer's datasheet. To find the read noise map, we calculated the SD image of the 100 frames in electrons (the conversion factor e⁻/ADU should be used). The duration of each frame was 10 ms. The root mean square (RMS) noise refers to the RMS value of all pixels read noise. The median noise is the median value of all pixels read noise (dark noise is considered equal to zero for this short acquisition time; Lambert and Waters, 2014).

We defined two metrics: VAR consists of measuring the variation in the percentage of the manufacturer's specification as indicated in the datasheet to the measured read noise experimental value:

$$VAR = 100 \times \frac{N_{theor}}{N_{exp}},$$

with $N$ the read noise value (RMS or median, depending on the given theoretical value) of the camera. The second one consists of the stability of the read noise over a long time and is defined as:

$$STAB_{noise} = 100 \times \left[ 1 - \left( \frac{N_{max} - N_{min}}{N_{max} + N_{min}} \right) \right],$$

with $N_{max}$ and $N_{min}$ the maximal and minimal read noise values. The timescales here are long-term (in correspondence to the source stabilization studies) and are on a scale of months to years.

### Biological imaging

We used yeast cells W303 strain, namely ySP16782. They expressed mCherry–septin protein (mCherry-cdc3::URA3). For testing the camera noise and offset uniformity, we used a SDCM (SpinSR10; Olympus) microscope, a 60×/1.5 UPLAPO objective, and a Fusion BT camera (Hamamatsu Photonics).

### Results

The read noise was calculated from three CCD, four EMCCD, and 25 sCMOS cameras from all main manufacturers and most often for both standard and fast camera readout modes.

We characterized the read noise maps (typical cases in Fig. 9 A or extreme cases with glow effect and offset calibration in Fig. S14 A). We observed that blemish corrections considerably decreased the measured noise RMS values (Fig. S14 B). DSNU map also gave useful information on the camera sensor chip corrections (Fig. S14 C). For each sCMOS camera, we measured the read noise and DSNU values in electrons (Fig. 9 B). The measured DSNU was never higher than the read noise, as expected from theory for this kind of cameras (Janesick, 2007).

For all types of cameras, we calculated the VAR metric for both RMS and median read noise values (Fig. 9 C). We calculated this metric for both noise values as camera manufacturers often provide a single readout noise value (either RMS or median). The majority of the VAR values were close to 100%. This shows that measured readout noise values were close to the manufacturer's specification. We found 84% of the calculated RMS values within a ±10% variation from the manufacturer's values. Almost half of the values are higher than 100% (better noise evaluation compared to specifications) and were most often observed when a blemish correction was applied. For CCD and EMCCD cameras (dots in blue and magenta), the RMS or median readout noise values are almost identical, and VAR distribution is kept in a narrow ±10% variation window. We propose then experimental tolerance values for VAR 90%.

We also monitored the read noise evolution over months for sCMOS cameras to calculate the $STAB_{noise}$ metric (Fig. 9, D and E). We define 97% as the limit value, which is always the case for the monitored cameras.

We studied the influence of the read noise and a non-homogeneous noise pattern on biological imaging. Fig. 9 F shows the read noise map (no illumination) in electrons of a back-illuminated sCMOS camera and the image of a biological sample (same sample as in Fig. 8 D, yeast cells, expressing mCherry coupled to septin protein). On the noise pattern, we distinguish vertical stripes (which is due to a pixel offset issue of

the sensor). The overall read noise is 0.78e⁻ (RMS value) but the inhomogeneous pattern can influence imaging. On the biological image, especially when the signal is dim, noise variations in the FOV may hinder quantifications. In our case, we zoom on the region of a vertical stripe of the noise pattern. Relative quantification studies of the intensity signal among cells can be influenced by the different offset and noise levels among cells.

### Discussion

The 32 tested cameras proved to be particularly reliable to their respective datasheet characteristics. The calculated metrics were most often in the limit values that we defined (98% of the cases for RMS and 91% for mean read noise). Although, a visual inspection of the noise map is often necessary to understand the origin of an issue (e.g., glow effect, offset calibration, or camera sensor chip corrections shown in Fig. S14).

VAR metric values showed mainly a lower than 10% variation of the manufacturer's values, and we considered the 90% VAR value as our experimental tolerance. When applicable, on-chip blemish correction should be activated as it yields lower noise values. For deeper sensor chip characterization, it is preferable to avoid pixel correction. It can hide some information on the number of warm pixels and their evolution in time. It should be also noted that the number of warm pixels may increase with exposure time due to the dark noise influence or cosmic rays. Cameras are constantly subjected to cosmic rays, which implies an increase in the number of hot pixels blocked at the saturation values (Theuwissen, 2007; Nampoothiri et al., 2011; Niedzwiecki et al., 2019). For QC studies, it is necessary to keep the same settings during the different measurements on different dates, so the results are comparable.

We propose measuring the camera noise once per year (Table 1) as our data show very low variability.

It is important to note that camera manufacturers apply protocols to correct the homogeneity of the chip through the DSNU (column or pixel correction for sCMOS), average hot pixels, or adjust the linearization of the sensor (Mandracchia et al., 2020). Other postacquisition alternatives exist to correct compromising noises for demanding image processing tasks (Liu et al., 2017).

Even though all the studied cameras were within the noise limit values, some of them had to be replaced, sometimes twice. Indeed, the major defect that we observed within our various core-facility measurements that lead to a replacement of the camera lies in the reliability of the vacuum seal carried out to protect the sensor chip of the cameras. When the seal fails, we observe condensation forming on the faceplate protection of the sensor chip depending on atmospheric conditions, as highlighted in Fig. S15. This situation has no simple solution to our knowledge and the camera must be replaced or returned for repair.

A feature of the sCMOS cameras is the rolling shutter readout option, in which different rows of the pixel array are exposed and read at different time frames. Therefore, the generated image contains parts exposed at different absolute times. At short exposure times or for very dynamic samples, it is advisable

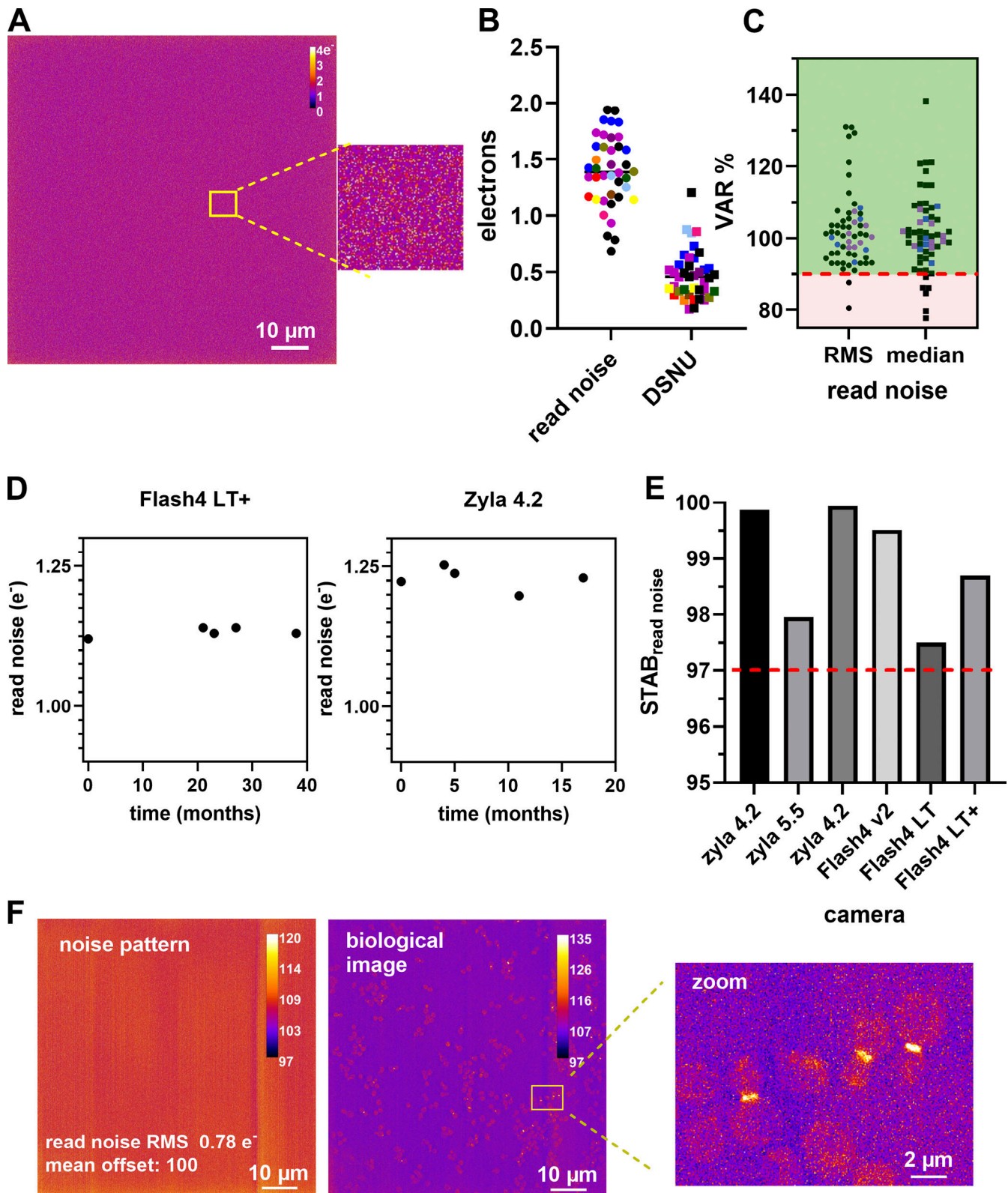

Figure 9. **Read noise and DSNU characterization of cameras. (A)** Read noise map in electrons of an Orca Flash4 v3 (Hamamatsu) and a zoom zone of 100 × 100 pixels. For display reasons, the maximum of the noise map is set to four electrons. **(B)** Read noise (RMS) and DSNU distribution as expressed in electrons of sCMOS cameras. Each color corresponds to a single camera, with different readout speed modes and the color stays the same for the read noise and DSNU representation. Total 40 datapoints for each case. **(C)** Read noise RMS/median variability (VAR) metric distribution for CCD (blue), EMCCD (magenta), and sCMOS cameras (black). The green zone and the red dashed line highlight the acceptable values. Total 58 points for each case. **(D)** Read noise monitoring over time for two sCMOS cameras. **(E)** Stability read noise metric (STAB) for six cameras over at least 1 yr of monitoring. The red dashed line defines the minimal

97% VAR tolerance value. **(F)** Influence of noise pattern in biological imaging. Noise map of an sCMOS back-illuminated camera (Fusion BT; Hamamatsu) for read noise and offset value characterization. Biological imaging of yeast cells expressing mCherry coupled to septin protein (zoom image shows imaging of three cells on an inhomogeneous noise-offset pattern). Scale bar in intensity (digital) values.

to check that the rolling shutter effect does not affect the acquired data (Lambert and Waters, 2014).

For biological imaging, read noise inhomogeneity on the sensor can influence quantification studies for low signal applications (e.g., fluorescent protein low expression levels as is often the case for yeast cells, shown in Fig. 9 F, or for single molecule applications).

It should be mentioned that for proper camera characterization, the linearity and the non-uniformity measurements of the photon response are of great importance. Of equal importance is the measurement of the sensitivity, which implies considering the whole microscope and its components. These measurements require reference samples, uniform illumination, and a reference calibrated sensor to compare (Dixon et al., 2008). They also require costly equipment and entail skill and time commitment levels that cannot be reasonably expected from most microscope users or facility staff. We thus do not recommend these protocols in the type of study performed here.

### Conclusion

We described the essential aspects of the QC tools, the acquisition process, the analysis, and gave a vision of how helpful an image database that collects data from different laboratories could be (Fig. 10). We tried to use simple tools, as affordable as possible, such as custom-made bead slides, plastic fluorescent slides, and power meters to assess QC of light microscopes over time. Other QC tools exist, like the Argolight slide (Argolight SA, France), GATTAquant beads (Gattaquant GmbH, Germany), nanorulers, or PSFCheck slides. The drawbacks of these tools (i.e., price or guaranteed performance for a limited duration of time) influenced our choice for using simple and cost-effective tools.

We defined metrics through simple and robust acquisition protocols that do not take >10 min to acquire or launch. Nevertheless, the acquisitions require that the user is well trained to perform experiments in a reproducible manner. A future direction will be to establish a single sample slide and the adapted software for an automated acquisition, eliminating associated human error here.

Acquired data needs to be analyzed in a reproducible manner as well. We thus developed and proposed the ImageJ/Fiji plugin MetroloJ_QC. When possible, we compared the measured values with the theoretical ones, such as the resolution for PSF or the coregistration. For some other cases, we referred to manufacturers' specifications like the stage repositioning or the sensor read noise. We also associated the limit values summed up in Table 1 with the limits for biological experiments when possible. We gave recommended measurement frequency, which depends on multiple parameters like the per year occupation time of the microscope, the number of users (major difference if the system is in a core facility with dozens or hundreds of users per year or in a research team, used by the same people the whole year), the

human resources that take care of the microscope, and the kind of biological experiments carried out.

The values are experimental, not fixed, and the fruit of the experience of 10 core facilities over the last years. Even more importantly, the limit values should serve as a basis for any biologists wishing to understand the orders of magnitude involved in their microscopy experiments. For instance, the QC requirements will not be the same for nucleus count imaging, subcellular protein colocalization, or single-molecule localization. We encourage researchers to contact core facilities or imaging scientists and discuss their specific QC needs, as these groups often dispose the expertise and the tools for QC measurements. When this kind of interaction is not possible, another way is to contact the microscope suppliers, who might have established their own specific QC measurements. This article will act as a starting point for the most often used QC measurements in light microscopy.

It should be noted that the datasets of that work were not acquired to compare the performance of different systems. This kind of comparison may be influenced by parameters difficult to standardize, like the microscope environment or the association of the different microscope components which are by nature different, even if they can have similar datasheets, and thus lead to different results. Our datasets and the resulting guidelines aim to characterize the microscope's health state over time, using relative and not absolute values.

We consider that the metrics that we define in that work complement the ISO 21073:2019 norm (ISO, 2019) and hopefully will be further enriched by the international initiative QUAREP-LiMi, whose role is to establish commonly accepted guidelines for quality assessment in light microscopy (Boehm et al., 2021; Nelson et al., 2021). This need for standardization accepted by the whole scientific community and principal actors from the academic and private sectors has been recently highlighted (Marx, 2022a, 2022b).

The recording of the metric values is the last link in the chain of the QC. Monitoring over time the different measured values is necessary, and a simple spreadsheet file may be sufficient. Ideally, the solution is a QC management database for processing the analysis and obtaining the metrics as quality control images and data are imported (Huisman et al., 2020; Linkert et al., 2010; Swedlow et al., 2021). Furthermore, such a database would be able to integrate these values into the metadata of the acquired images (Hammer et al., 2021). In the end, these metrics would allow an open and accessible inspection of the quality of the used microscope and of every single image, within a collection of microscopy metadata, as recently proposed by Rigano et al. (2021), with the Micro-Meta App approach, and that would be a key contribution to an automated image quality assessment method.

### Online supplemental material

Fig. S1 shows PSF variability with image processing, the image field of view, and the PSF repeatability accuracy. Fig. S2 shows

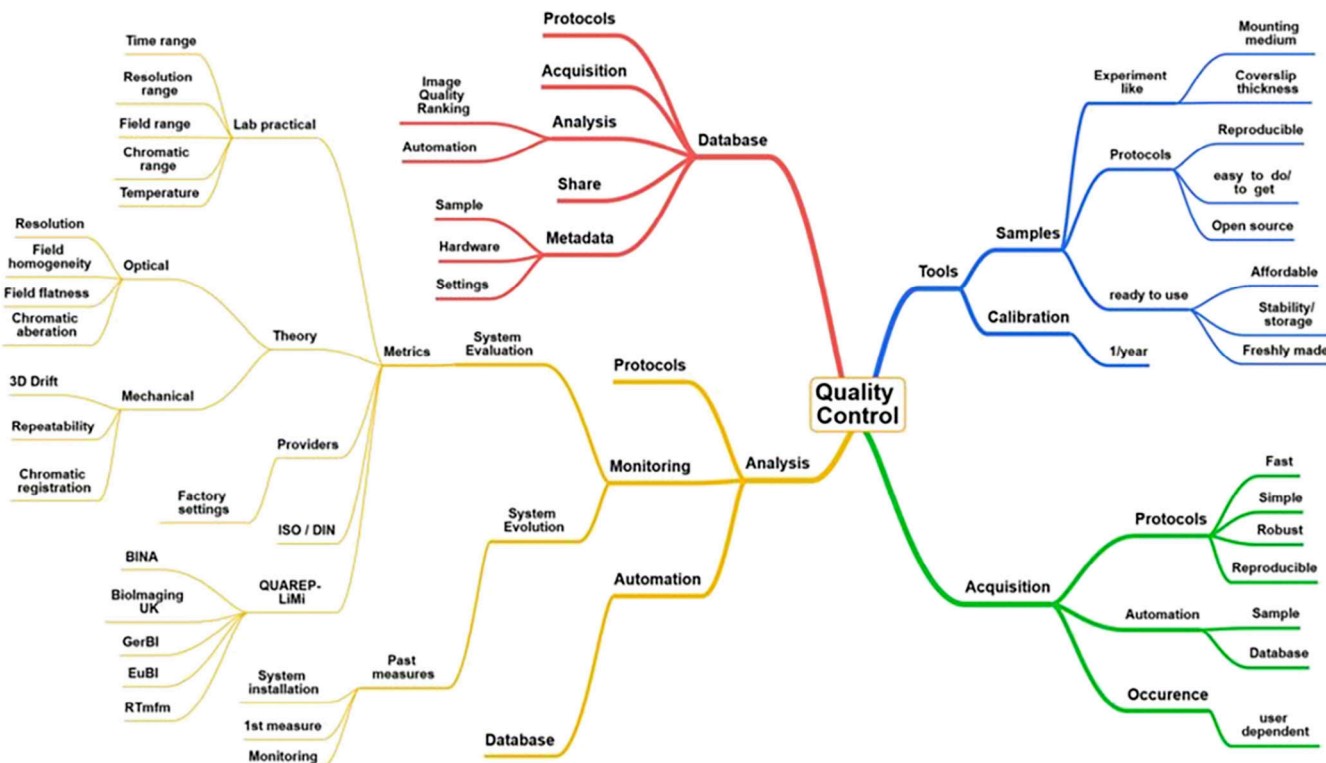

**Figure 10.** **Heuristic map of light microscopy QC.** QC needs tools (blue branch), acquisition protocols (green branch), image analysis (yellow branch), and clever data storage (red branch).

PSF variability with SBR and SNR ratios, pixel size for LSCM, and bead size for WF. Fig. S3 shows the PSF analysis workflow with the MetroloJ_QC plugin. Fig. S4 shows the field illumination metrics using either a green chroma plastic slide or a glass slide/coverslip configuration with a fluorophore (rhodamine) layer for a spinning disk microscope. Fig. S5 shows the field illumination workflow with the MetroloJ_QC plugin. Fig. S6 shows the distribution of uniformity and centering field illumination metrics classified by the detector type (CCD/EMCCD, sCMOS, PMT). Fig. S7 shows the influence of the bead size on the coregistration accuracy calculated by the MetroloJ_QC plugin. Fig. S8 shows the coregistration workflow with the MetroloJ_QC plugin. Fig. S9 shows the warm-up time of lasers of a LSCM and a SDCM. Fig. S10 shows the short-term power stability of the illumination source and Fig. S11 shows the long-term stability for two different cases (one for lasers and another for a X-Cite light source). Fig. S12 shows the stage positioning repeatability dependence on stage speed and acceleration. Fig. S13 shows the camera noise specifications (read noise, dark offset, DSNU) workflow calculation with the MetroloJ_QC plugin. Fig. S14 shows the camera read noise and DSNU map and distributions. Fig. S15 shows three examples of condensation effects on the camera chip.

**Data availability**
The download link of the plugin, direct access to its dependencies, its manual, and simple sample preparation and acquisition protocols can be found at: https://github.com/MontpellierRessourcesImagerie/MetroloJ_QC. We provide the majority of the datasets used for this work, with the maximum of metadata available under the principle of accessibility of research open source data compatible with the FAIR principle. The data represent 360 GB of data and are accessible under the following DOI: https://doi.org/10.57889/OD_METROLOJ_QC.

**Acknowledgments**
The authors thank Sebastian Beer and Bruno Emica (Hamamatsu Photonics) and Gerhard Holst (Excelitas PCO GmbH) for their valuable support and discussion on the metrics of the QC of camera sensors. We would like to thank Kees van der Oord (Nikon Europe B.V.), Christian Schulz (Leica Microsystems CMS GmbH), and Ralf Wolleschensky (Carl Zeiss Microsocpy GmbH) for their helpful input on PSF issues, and Arnd Rühl (Märzhäuser Wetzlar GmbH) for his invaluable advice on the stage part of this manuscript. We thank Yea-lih Lin for the help on Fig. 1 and Azam Aslemarz, Nathalie Morin, Raissa Rathar, Daniel Bouvard, and Maritzaida Varela for providing us with the biological samples for the QC metrics validation. We thank Claire Dupont for the measurements and feedback on the stage repeatability. We are deeply indebted to Ulrike Boehm, Pierre Bon, Mark Scott, Alain Trautmann, Raphael Gaudin, and Dimitris Liakopoulos for their critical review of the manuscript. We finally thank the French RTmfm community for the fruitful exchanges throughout the past years.

The Montpellier Ressources Imagerie (BCM), PICT (Institut Curie), Bordeaux Imaging Center and IMAG'IC (Institut Cochin) are part of the National Infrastructure France-BioImaging supported by the French National Research Agency (ANR-10-INBS-04). This work was supported by the French technological network RTmfm of the CNRS "Mission pour les Initiatives Transverses et Interdisciplinaires".

The authors declare no competing financial interests.

Author contributions: O. Faklaris and T. Guilbert conceptualized the study. O. Faklaris, A. Dauphin, B. Monterroso, P. Frère, D. Geny, T. Manoliu, D. Schapman, and T. Guilbert designed the protocols. O. Faklaris, L. Bancel-Vallée, A. Dauphin, B. Monterroso, P. Frère, D. Geny, T. Manoliu, S. de Rossi, D. Schapman, and T. Guilbert collected the data. O. Faklaris and T. Guilbert performed data analysis. F.P. Cordelières developed the original software analysis tool and J. Cau developed it further for this study, integrating the proposed metrics and automating the analysis. O. Faklaris, A. Dauphin, J. Cau, and T. Guilbert designed the figures. O. Faklaris wrote the original draft and supervised the project. All authors reviewed the manuscript. O. Faklaris, A. Dauphin, R. Nitschke, and T. Guilbert revised the manuscript.

Submitted: 15 July 2021

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

# Supplemental material

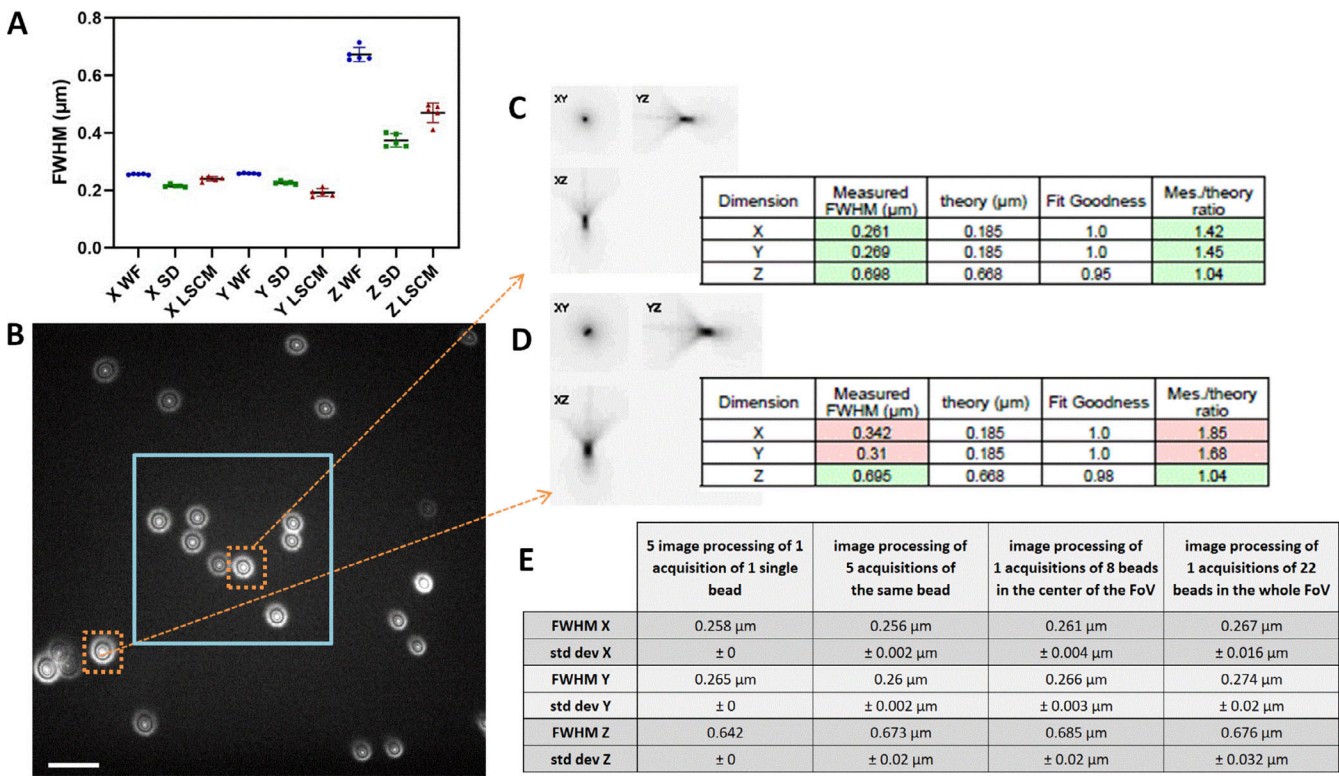

Figure S1. **PSF variability with image processing, acquisition repeatability, and FOV. (A)** PSF repeatability accuracy for wide-field (WF), SDCM, and LSCM along the *x*, *y*, and *z* axis. PSF were acquired five consecutive times. Error bars represent SD. **(B)** One *z* plane image of a 175 nm bead sample, slightly out of focus to observe the symmetry of the PSF pattern. Beads near the corners of the FOV show a significant asymmetry. Image taken on a WF-TIRF microscope (TiE Nikon) at zero angle mode, with the 488 nm laser, with a 100×/1.49 objective and use of an additional lens 1.5× to respect the Nyquist criterion (final pixel size, 73 nm), with a Prime95B Photometrics camera (FOV 87.6 × 87.6 μm). Scale bar is 2 μm. In blue square: the central area of the FOV that is taken into account for the PSF calculations for this objective. **(C and D)** PSF profiles along *xy*, *yz*, and *xz* as extracted by the MetroloJ_QC plugin (square root intensity) of the beads in dashed orange squares in B. Although along the *z* axis the FWHM is similar, along the *xy* axis the FWHM is significant different and higher for beads far from the center of the image. **(E)** Calculated FWHM with the SD for three repeatability studies (processing, acquisition, FOV).

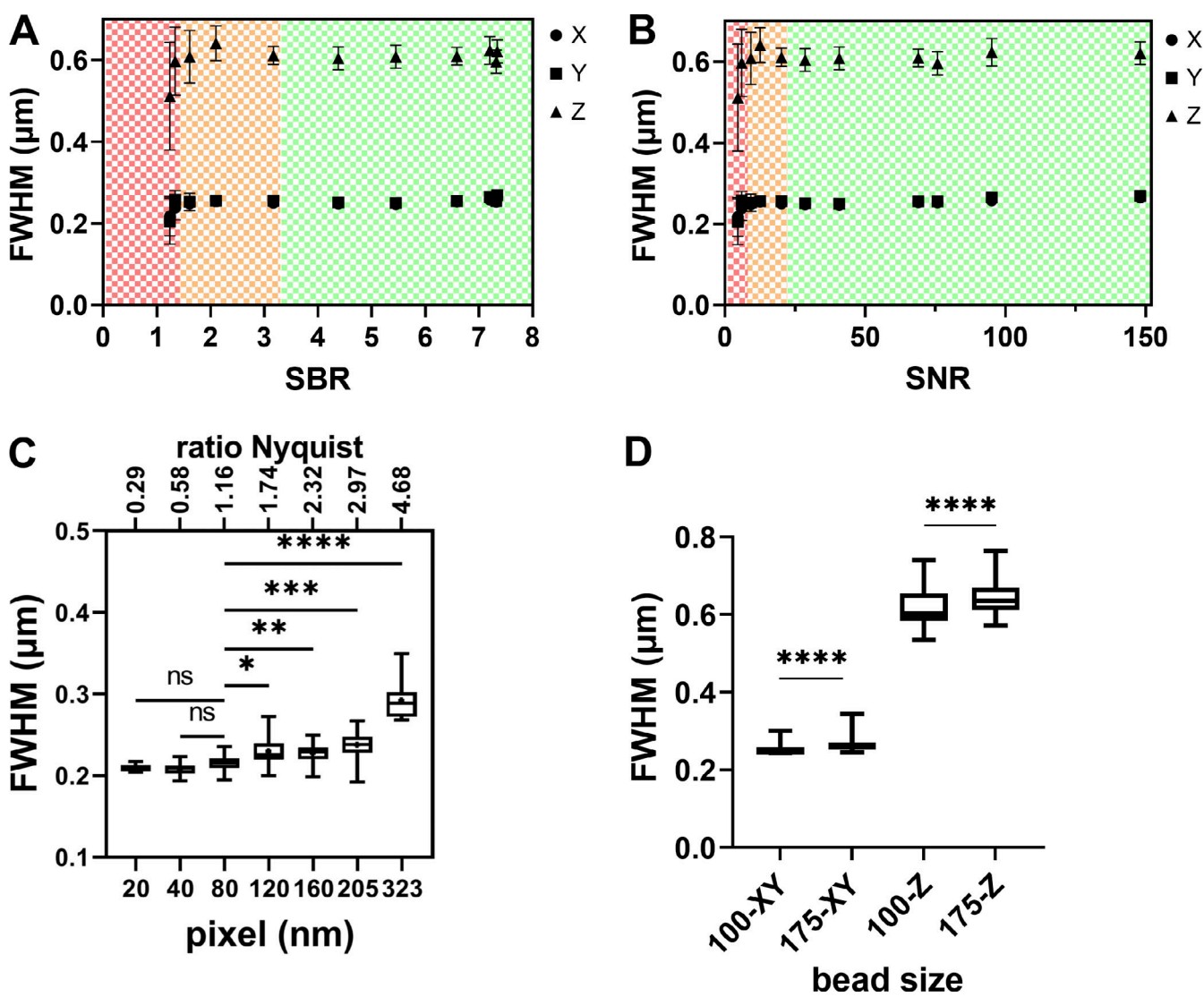

Figure S2. **PSF variability with SBR, SNR ratios, pixel size for LSCM, and bead size. (A and B)** Effect of the SBR and SNR (for 12-bit dynamics) ratio on FWHM estimation precision for a wide-field microscope. The same three FOVs of 29 175-nm one-color beads were acquired for different exposures and excitation intensities on an upright Carl Zeiss Microscopy WF microscope, GFP channel, 63×/1.4 objective. The SBR ratio is estimated using the segmented bead mean intensity value (Signal) and the mean value of a 1-µm thick background annulus around the segmented bead (Background) as calculated by the plugin MetroloJ_QC. As SBR is increasing, the FWHM estimation accuracy gets better. For very low SBR the FWHM values decrease. We distinguish three tolerance zones. The green zone is the one presenting the lowest errors and the best fit (R2 equal to 1). The orange zone presents a higher error and a worse fit (0.93 < R2 < 0.99). The red zone presents both high error and bad fit ($R^2 < 0.9$). **(C)** Lateral FWHM dependence on the sampling rate on a LSCM microscope (LSM880, Carl Zeiss Microscopy GmbH) with pinhole at 1 A.U. The normality of the distributions for each condition was first confirmed with Shapiro-Wilk test. Statistical significance was determined using Dunnett's multicomparison test, with the 80 nm setting chosen as the control value. The P value was non-significant (ns), *, P < 0.05; **, P < 0.005; ***, P < 0.0005, or ****, P < 0.0001. Data points varied from 8 to 22. **(D)** PSF dependence on microsphere size for WF and LSCM microscopes (WF in A and B and LSCM in C). WF-100 and WF-175 stand for WF PSFs for 100 and 175 nm beads, respectively. Distribution normality was tested with Shapiro-Wilk test and showed a not normal distribution (P < 0.0001). Statistical significance was determined applying a Kolmogorov-Smirnov test for each pair and each condition (one pair: 100 and 175 nm beads). The P value for each pair was <0.0001. The data points were 87 and 156 for 100 and 175 nm beads respectively. All measurements in C and D were carried out with a 63×/1.4 lens, at the GFP channel (525 nm emission).

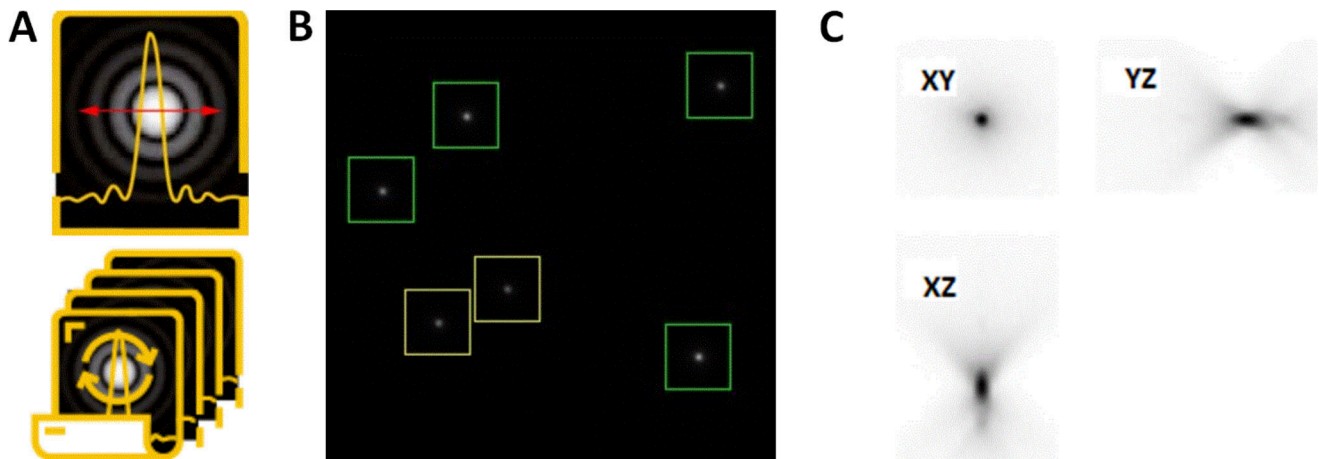

**D** Resolution table:

| Channel | Sig/Backgnd ratio | Dimension | Measured FWHM (µm) | theory (µm) | Fit Goodness | Mes./theory ratio |
|---|---|---|---|---|---|---|
| Channel 0 (em. 525.0nm) | 16.3 | X | 0.262 | 0.191 | 1.0 | 1.37 |
| | | Y | 0.265 | 0.191 | 1.0 | 1.39 |
| | | Z | 0.592 | 0.718 | 0.98 | 0.82 |

**E** Lateral asymmetry ratios:

| Channel | Ratio |
|---|---|
| Channel 0 (em. 525.0nm) | 0.99 |

**F** Summary

| Channel | X | Y | Z |
|---|---|---|---|
| Channel 0 | 0.264 +/- 0.005 µm<br>16.0 beads<br>(0.191 µm)<br>mean R2: 1.0<br>mean SBR: 13.08 | 0.268 +/- 0.004 µm<br>16.0 beads<br>(0.191 µm)<br>mean R2: 1.0<br>mean SBR: 13.08 | 0.6 +/- 0.014 µm<br>16.0 beads<br>(0.718 µm)<br>mean R2: 0.98<br>mean SBR: 13.08 |

Figure S3.   **PSF analysis workflow with the MetroloJ_QC. (A and B)** PSF or PSF batch icon (A) and a "beadOverlay" image (B) is generated taking into account the declared user parameters (ROI size, prominence value). Beads that are taken into account for the FWHM calculation are in the green squares. Yellow square means that the beads are too close. **(C)** Square root PSF image of one bead that helps visualizing the PSF and detecting aberrations. **(D)** Resolution table: the measured FWHM along x, y, and z of the bead in C. Theoretical values are provided, along with the calculated fit goodness (R²) and the ratio FWHM measured/FWHM theoretical. If values are within specs they are highlighted in green; if not, they are highlighted in red. **(E)** xy asymmetry is monitored by the LAR. **(F)** Summary table for the multibead acquisition presenting the mean FWHM with the SD for the three axes, the number of beads taken into account, the mean R², and the mean SBR value.

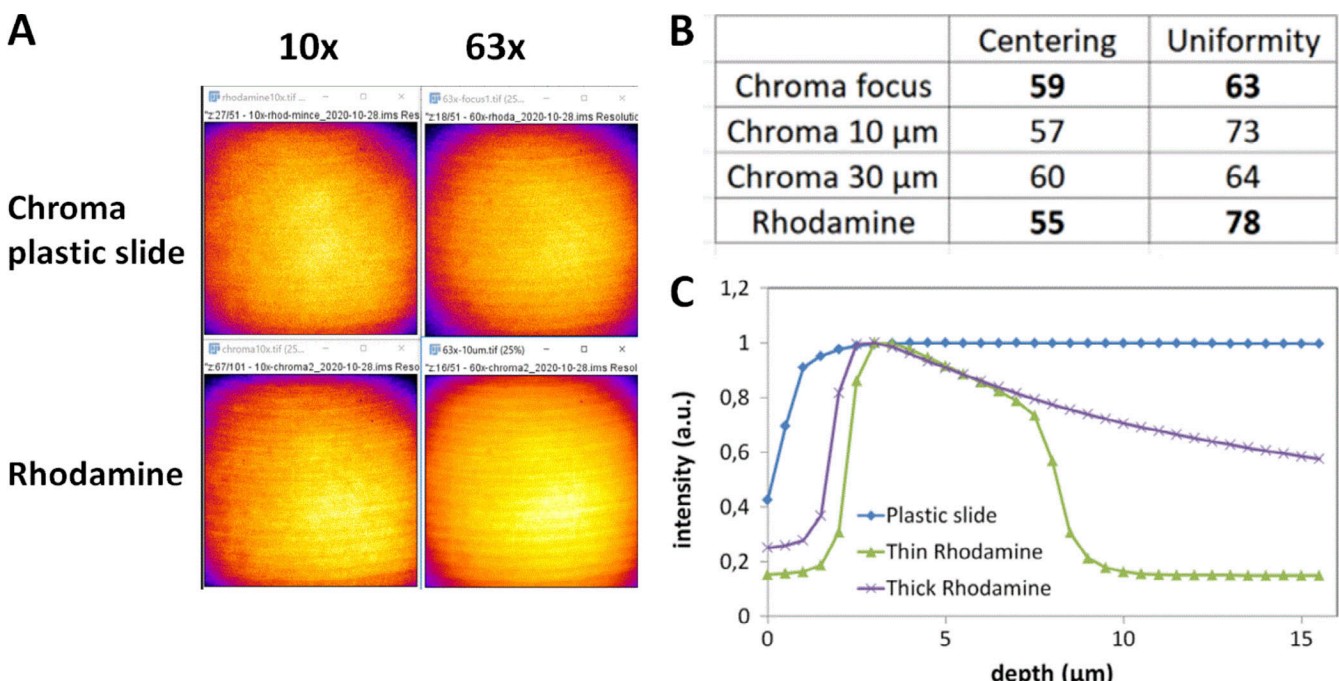

Figure S4. **Field illumination using either a green chroma plastic slide or a glass coverslide/coverslip configuration with a rhodamine layer for a spinning disk Dragonfly microscope (Andor–Oxford Instruments). (A)** Field illumination images at 488 nm excitation for a 10× and 63× with a plastic and rhodamine slide, detected with an EMCCD iXon888 camera. Artifacts shown on these images come from a synchronization artifact between Nipkow disk rotation and camera acquisition. The image projected on the sensor chip results from the integration of multiple individual scans by the Nipkow disk. A short exposure time can result in these artifacts, as explained in (Chong et al., 2004). **(B)** Centering and uniformity metrics for A. The acquisition was set at the plastic/glass or rhodamine/coverslip interface, or 10/30 µm deep within the plastic/rhodamine layer. **(C)** Effect of focal plane depth on emission intensity using either a plastic slide or thin/thick layers of fluorescent rhodamine dye (PLAN APO 63×/1.4 objective, ex. 488 nm).

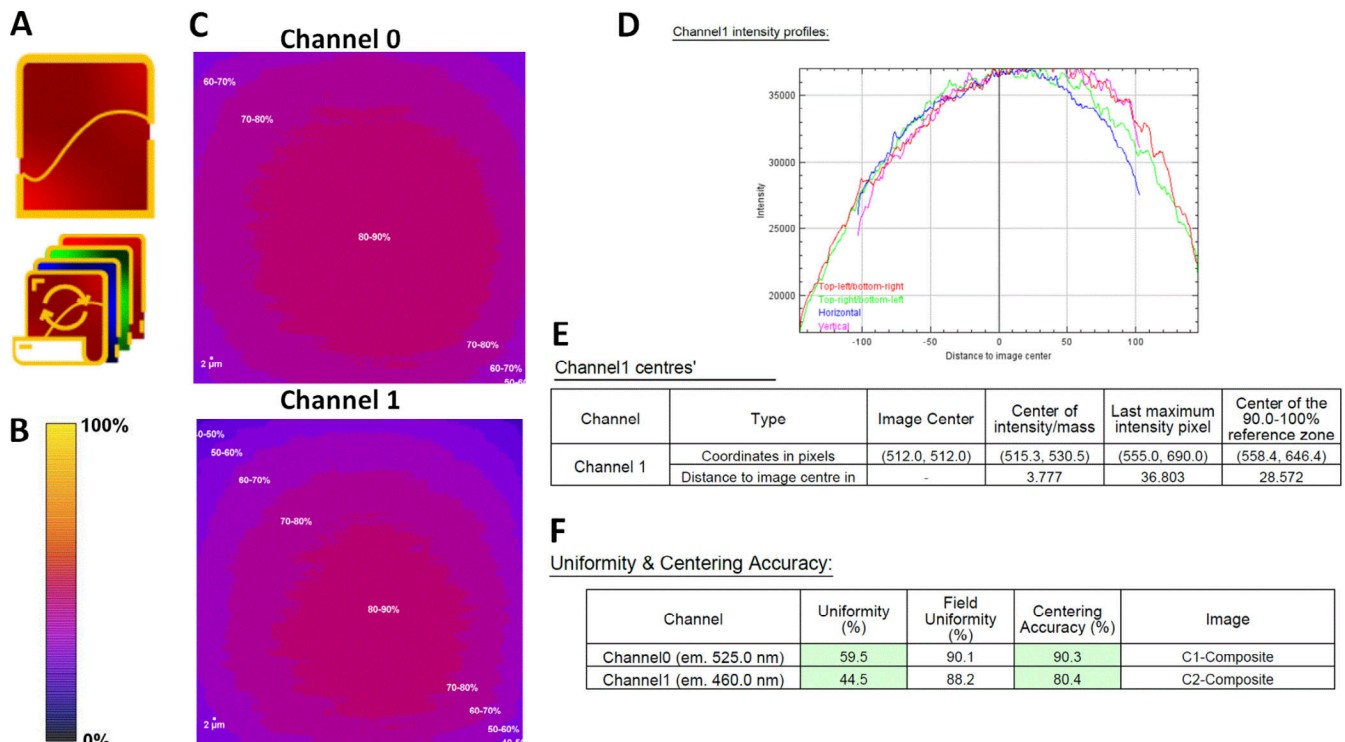

Figure S5.   **Field illumination workflow with the MetroloJ_QC. (A)** Field illumination single and batch icon. **(B)** Intensity scale bar. **(C)** Normalized intensity profile of channels 0 and 1. **(D)** Intensity profile along horizontal/vertical/diagonal lines going through the image geometrical center. **(E)** The location of the image center (geometrical center), the center of intensity (Center of Mass of the page channel), the maximum intensity pixel for the page channel) and the reference zone are provided, along with the distances to the geometrical image center. **(F)** Uniformity and centering metrics for both channels of C. Centering accuracy is computed using the 90–100% zone as reference rather than the maximum intensity pixel position. If values are within specs they are highlighted in green if not in red. Specs can be modified by the user.

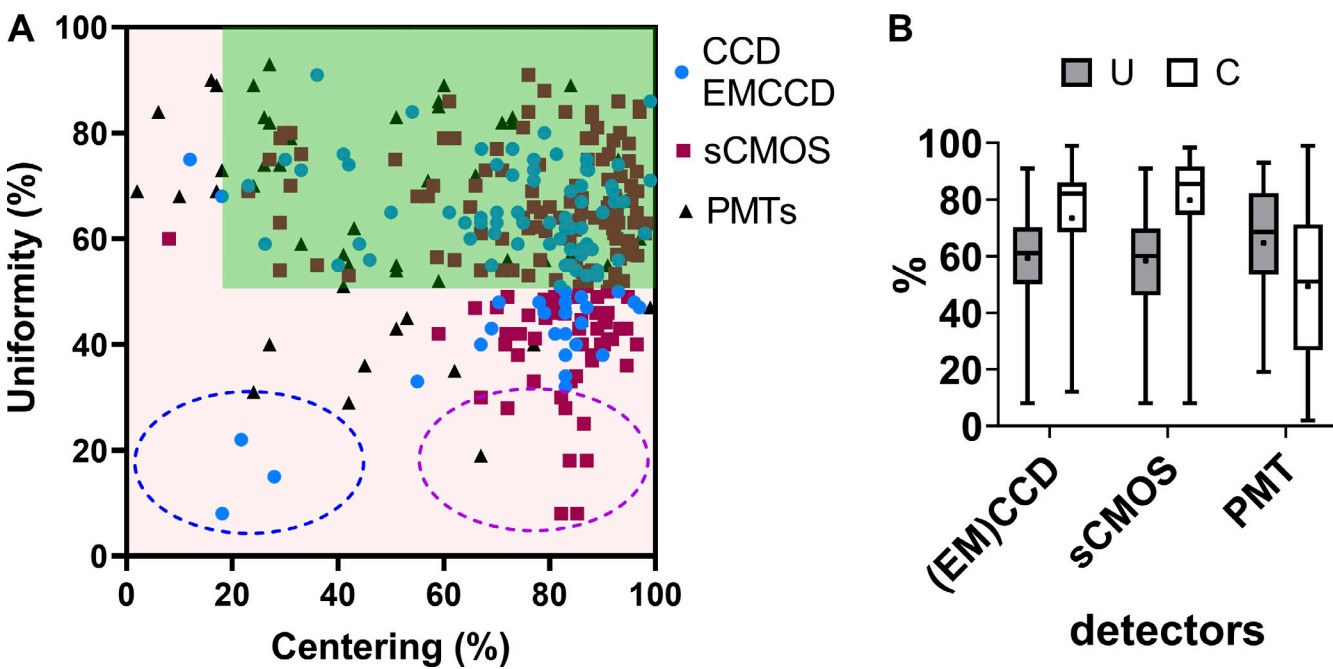

Figure S6.   **Distribution of uniformity (U) and centering (C) field illumination flatness metrics classified by the detector type. (A)** U–C distribution. The tolerance area is in dark green. Representative examples are shown. Inside the blue dashed circle are three values measured with an EMCCD camera. The magenta dashed circle highlights a subpopulation of low U/high C cases, only associated with sCMOS camera images. **(B)** Box plots summarizing all the U–C distributions of A. Whiskers show the min/max values, the dot in the box shows the mean value, and the box shows the SD. We found that the sCMOS cameras show a higher dispersion compared to the CCD-EMCCD sensor chips. We indeed observed that low U/high C accuracy combinations were mostly associated with larger sensor sizes (magenta circle in A). The only low U and C combinations (blue circle in A) were associated with misaligned TIRF microscopes, on which proper laser alignment had to be manually set before each experiment. The reference is the GFP channel, although many of the measurements were performed for both the DAPI–GFP channels or for all of the basic four channels.

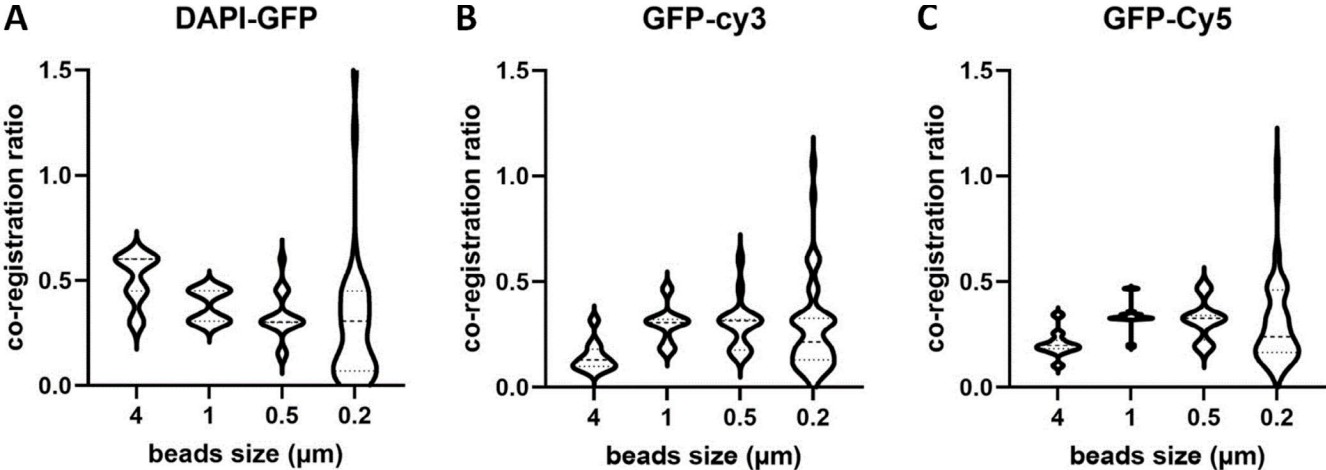

Figure S7.   **Influence of the bead size on the coregistration accuracy calculated by MetroloJ_QC. (A–C)** Co-registration ratio distributions for 4, 1, 0.5, and 0.2 μm multicolor beads. For all cases, more than five beads per condition and two FOV were examined, for a 100×/1.4 objective, on an upright WF microscope (AxioImager Z2, Carl Zeiss Microscopy GmbH). We observe that 0.2 μm beads present a high dispersion, with a coefficient of variation higher than 65% for all channel combinations. The 1 μm beads show the smallest dispersion for all cases and are thus the most adapted beads to use for this kind of measurements.

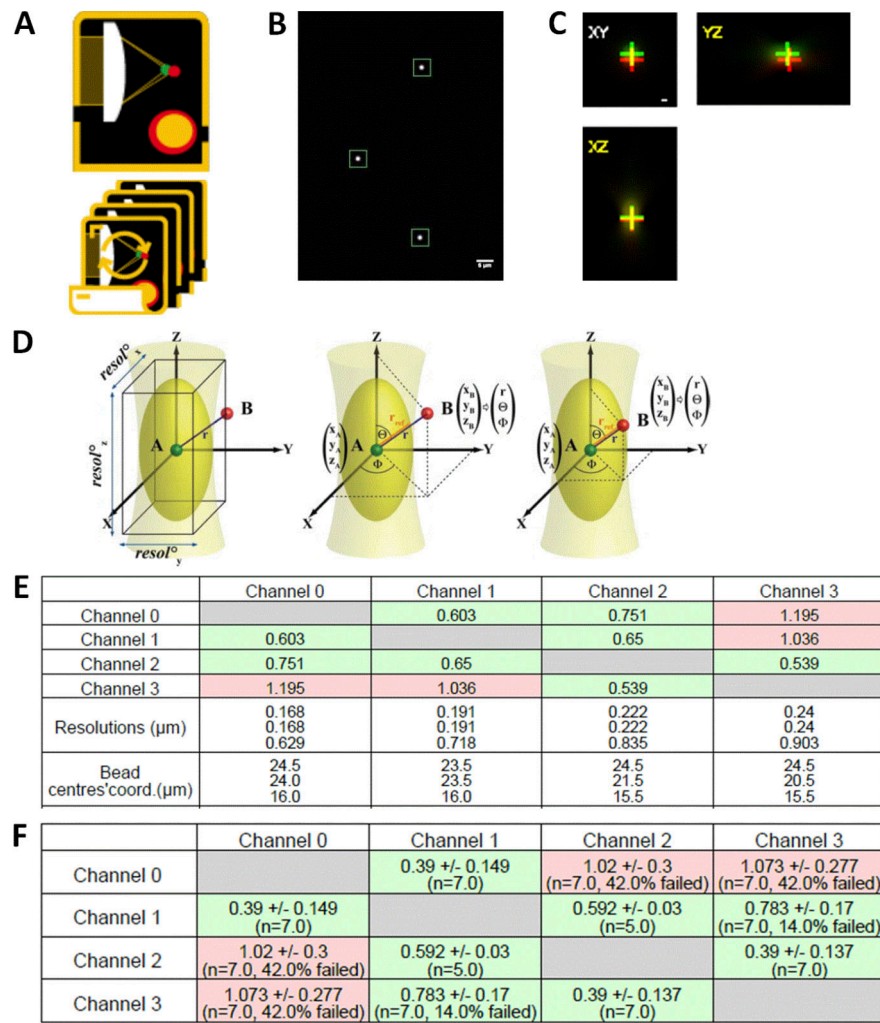

**E**

| | Channel 0 | Channel 1 | Channel 2 | Channel 3 |
|---|---|---|---|---|
| Channel 0 | | 0.603 | 0.751 | 1.195 |
| Channel 1 | 0.603 | | 0.65 | 1.036 |
| Channel 2 | 0.751 | 0.65 | | 0.539 |
| Channel 3 | 1.195 | 1.036 | 0.539 | |
| Resolutions (µm) | 0.168<br>0.168<br>0.629 | 0.191<br>0.191<br>0.718 | 0.222<br>0.222<br>0.835 | 0.24<br>0.24<br>0.903 |
| Bead centres'coord.(µm) | 24.5<br>24.0<br>16.0 | 23.5<br>23.5<br>16.0 | 24.5<br>21.5<br>15.5 | 24.5<br>20.5<br>15.5 |

**F**

| | Channel 0 | Channel 1 | Channel 2 | Channel 3 |
|---|---|---|---|---|
| Channel 0 | | 0.39 +/- 0.149 (n=7.0) | 1.02 +/- 0.3 (n=7.0, 42.0% failed) | 1.073 +/- 0.277 (n=7.0, 42.0% failed) |
| Channel 1 | 0.39 +/- 0.149 (n=7.0) | | 0.592 +/- 0.03 (n=5.0) | 0.783 +/- 0.17 (n=7.0, 14.0% failed) |
| Channel 2 | 1.02 +/- 0.3 (n=7.0, 42.0% failed) | 0.592 +/- 0.03 (n=5.0) | | 0.39 +/- 0.137 (n=7.0) |
| Channel 3 | 1.073 +/- 0.277 (n=7.0, 42.0% failed) | 0.783 +/- 0.17 (n=7.0, 14.0% failed) | 0.39 +/- 0.137 (n=7.0) | |

Figure S8. **Co-registration workflow with the MetroloJ_QC. (A)** Co-registration single and batch icon. **(B)** A "beadOverlay" image is generated when more than one bead is in the FOV, taking into account the declared user parameters (bead size, ROI size, bead position on *z* stack). **(C)** For each color/channel combination profile view images are composed of three maximum intensity projections, *xy, xz, yz* (side views) are generated. Crosses indicate the respective position of the green channel (first channel declared using the stack order) and the red channel. This is done for all channel combinations. **(D)** Calculation of the reference distance $r_{ref}$. Left: Centers of objects (A and B) are drawn as red and green spheres, respectively. The PSF is schematized in light yellow, while the first Airy volume appears in dark yellow. The former width, height, and depth define the resolution along the three axes. Middle: A and B are not colocalized as $r > r_{ref}$. Right: A and B are colocalized as $r ≤ r_{ref}$. Illustration from Cordelières and Bolte, ImageJ User and Developer Conference Proceedings, 2008, Luxembourg. **(E)** A ratio table is generated for each bead indicating the measured coregistration ratios of all channel combinations, the theoretical resolution for each channel, and the bead position coordinates. **(F)** If ratio values are within specs they are highlighted in green; if not, they are highlighted in red, when more than one bead is in the FOV, a ratio table gives the mean ratio values with their SD values. The number of beads taken into account for each channel is also given.

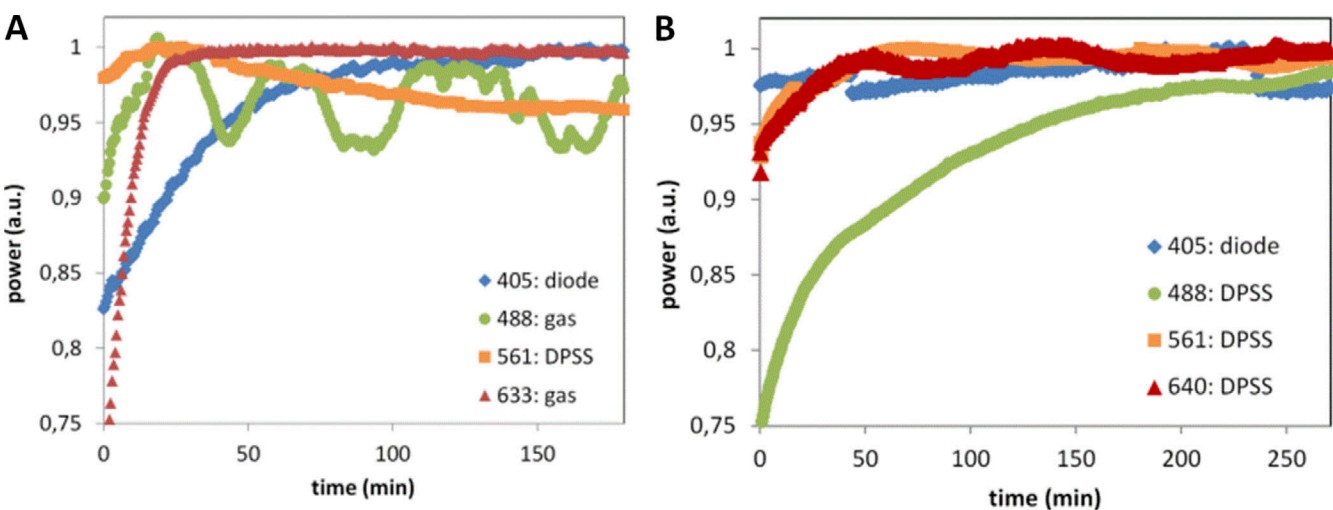

**Figure S9.** **Warming up time of lasers of two different systems. (A and B)** Warm up of four different laser types on a (A) LSCM microscope (SP5 – Leica of Fig. 6 F) equipped with an argon laser (488), HeNe laser (633), a diode laser (405), and a DPSS laser (561), and a (B) SDCM microscope with diode (405) and DPSS lasers. It is to be mentioned that all sources for both systems pass through an AOTF device. The LSCM setup shows (1) some decrease of the DPSS 561 nm laser and (2) high fluctuations of the power of the argon 488 nm line. These observations were a warning for the facility team, who after contacting the service technician of the microscope manufacturer and after a visit on-site, managed to identify and correct the problem. A damaged fiber and aging of the lasers were identified as the main reasons for these poor performances. The SDCM setup in B shows an extremely long warm-up time for the 488 nm laser.

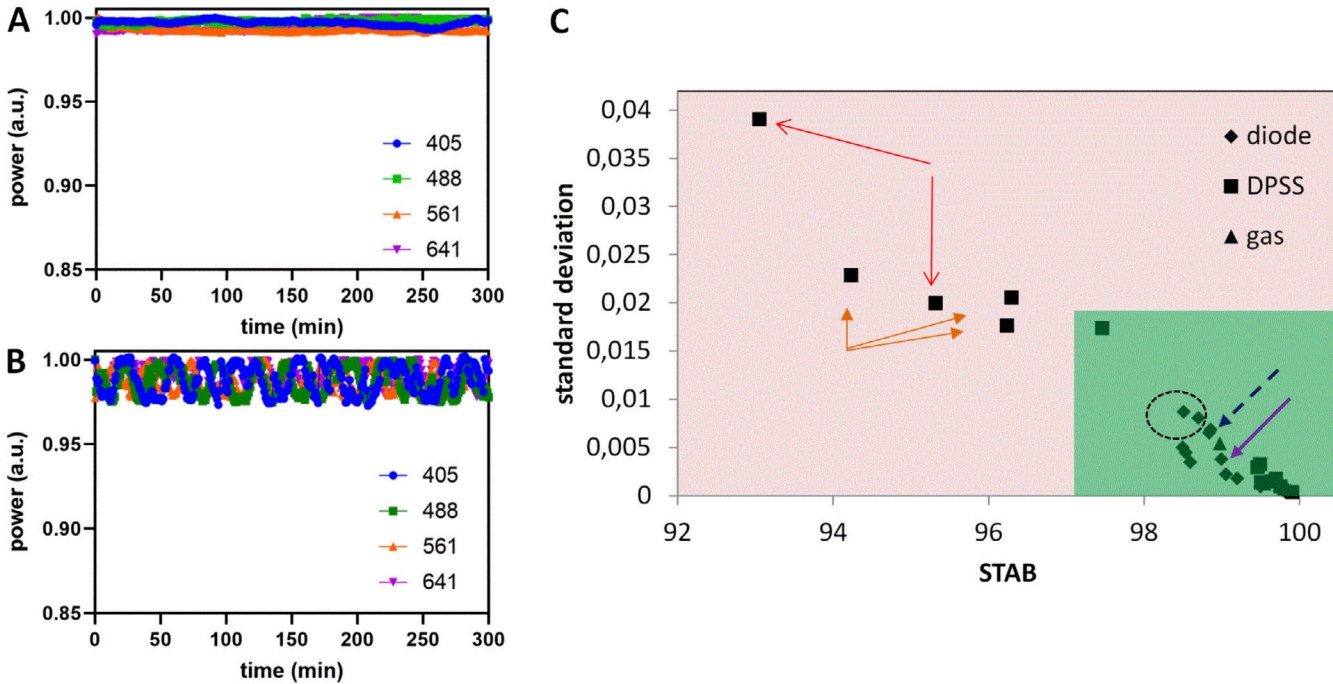

**Figure S10.** **Illumination laser power short-term stability. (A and B)** Short-term stability monitoring for two SDCM equipped with four diode lasers. Although the setup (B) shows higher fluctuations than (A), the calculated STAB metric stays relatively high, in the range of 98 < STAB < 99. **(C)** Laser power stability STAB versus SD is normalized to "1" laser power values over 5 min. The black dashed circle contains the metrics measurements of the four laser lines of B. With low stability, the lasers come from the same microscope. The red open arrows point to two DPSS lasers of the same WF-TIRF microscope pointed out with red arrows in Fig. 6 D. The orange closed arrows point to three DPSS lasers coming from the same SDCM microscope. The blue dashed and magenta closed arrows point to the blue and magenta arrow laser of Fig. 6 D, respectively.

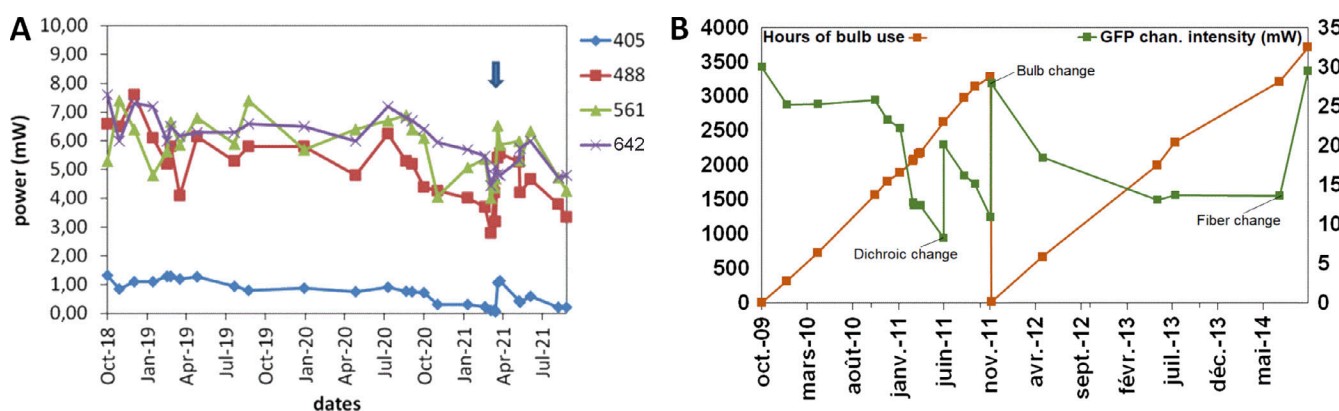

Figure S11. **Illumination source power long-term stability. (A)** Long-term stability of four laser sources on a TIRF microscope (setup of Fig. 6 B). All four lasers are coupled in an optical fiber. The system was out of maintenance contract and the source alignment was regularly done by the facility staff. The blue arrow shows the optical fiber replacement and total realignment of the lasers. **(B)** Long-term stability of X-Cite source. A single replacement of the bulb took place, represented by a drop in the bulb hours (orange line) and a decrease in the emitted signal (green line). As expected other changes in some light-path components (dichroic mirror, fiber) influenced the measured signal as well.

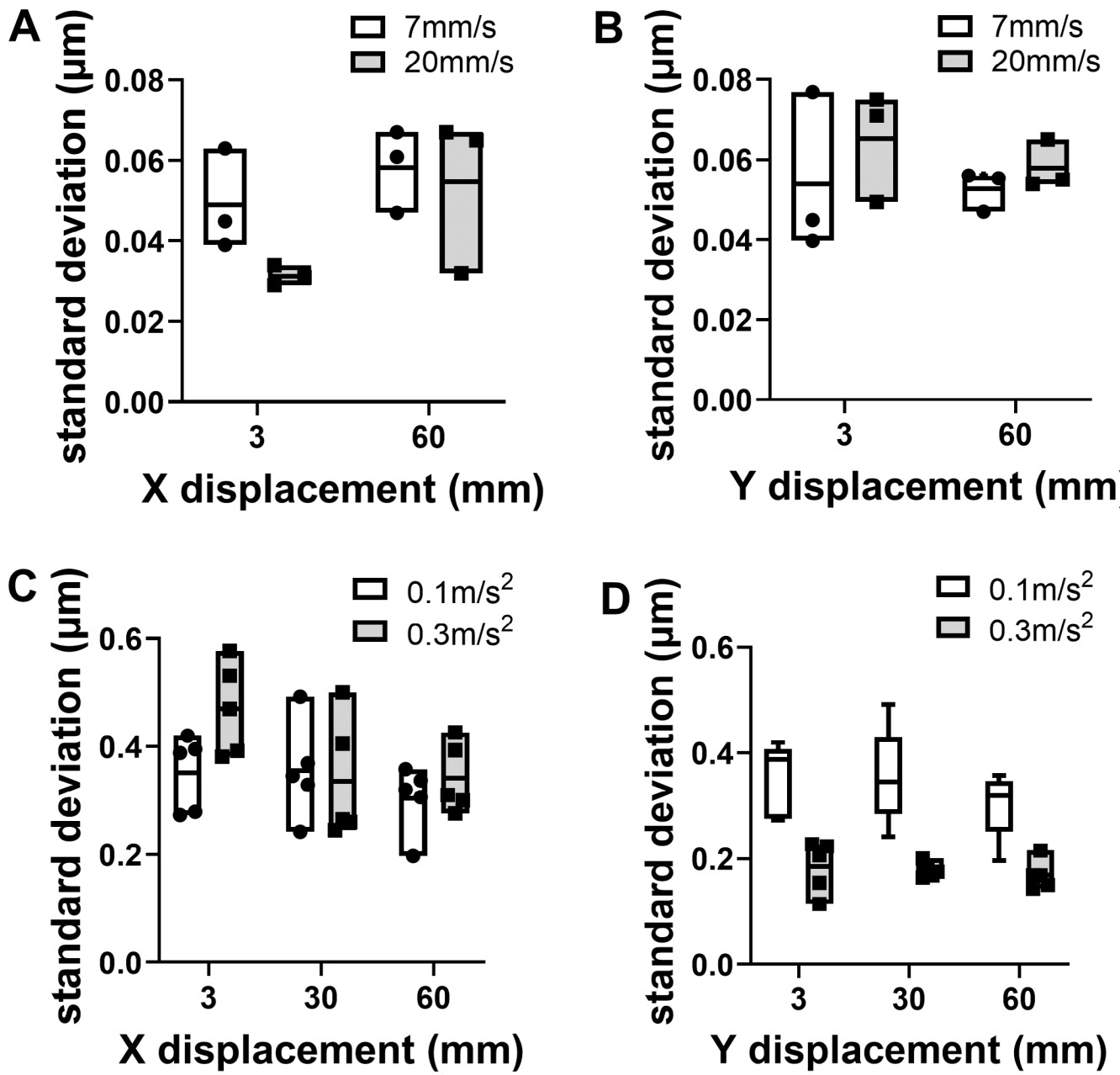

Figure S12. **Stage positioning repeatability dependence on stage speed and acceleration. (A and B)** Bi-directional stage repeatability along the *x* (A) and *y* (B) axis for one stage for two different stage speeds (7 mm/s and 20 mm/s) for two different *xy* displacements (3 and 60 mm). The horizontal line of the floating bar shows the mean value for three consecutive repeatability measurements. The 60 mm *xy* value corresponds to movements from the center to the extremities of a multiwell plate. The studied stage was an IX3-SSU Olympus Ultrasonic Stage with linear encoders. **(C and D)** Bi-directional stage repeatability along the *x* (C) and *y* (D) axis for one stage for two different stage accelerations (0.1 m²/s and 0.3 m²/s), for a speed of 10 mm/s, for three different *xy* displacements (3, 30, and 60 mm). The horizontal line of the floating bar shows the mean value for five consecutive repeatability measurements. The studied stage was a Märzhäuser SCANplus IM 130 × 85 two stepper stage on an inverted WF microscope (Olympus, IX83). We observe small differences between the *x* and *y* axes, but globally the applied stage speeds and accelerations do not influence the *xy* repeatability.

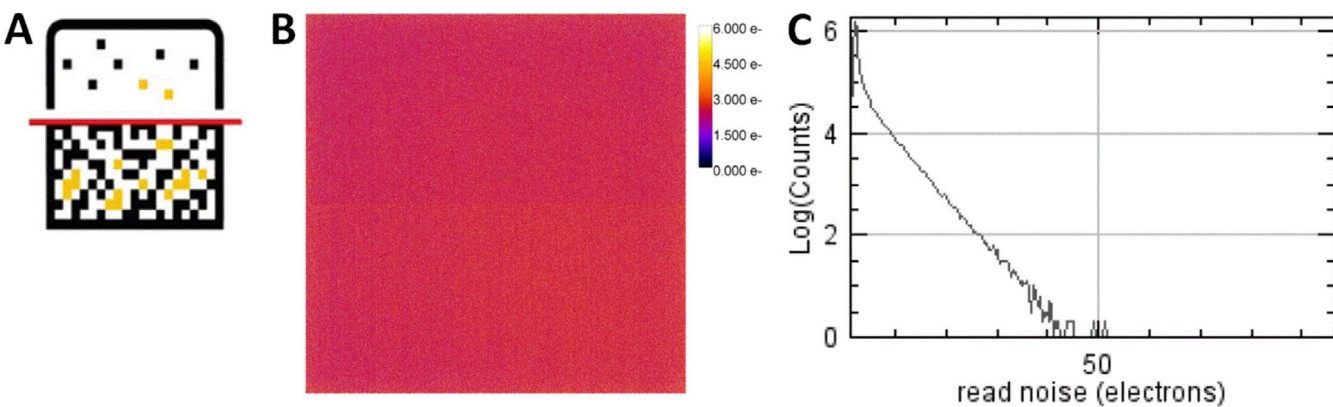

| Field | | Value |
|---|---|---|
| Offset value (ADU) | | 101.1 |
| Noise | rms (e-) | 1.741 |
| | median (e-) | 1.164 |
| | DSNU (e-) | 1.633 |

Figure S13. **Camera noise specifications (read noise, dark offset, DSNU) workflow calculation with the MetroloJ_QC. (A)** Camera tool icon. **(B)** Noise map in electrons calculated for an sCMOS camera after generating an average intensity projection image and an SD of the intensity projection image of a time-lapse of 100 images in the dark. The average intensity projection image is also used to calculate the DSNU and offset in D. **(C)** Noise distribution of B. **(D)** Calculated offset and noise specifications of B.

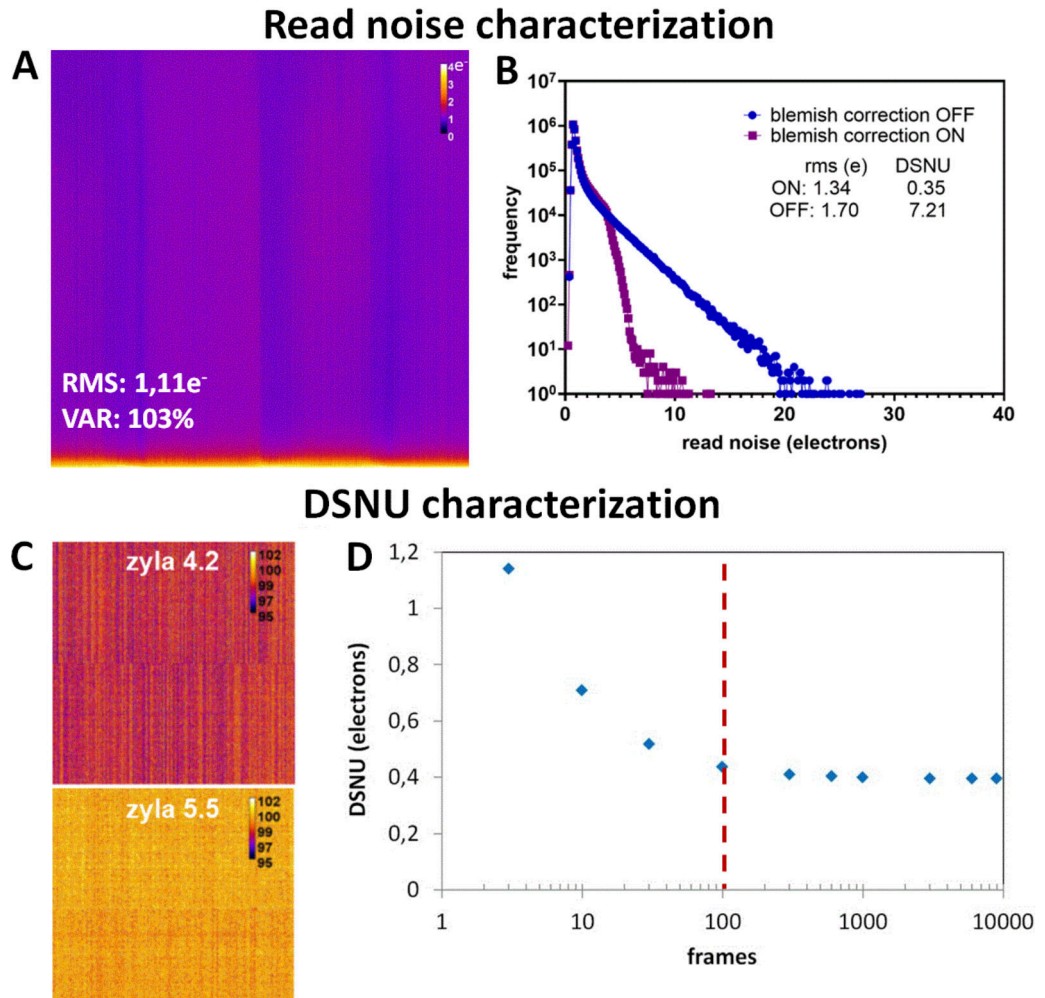

Figure S14.    **Camera read noise and DSNU map and distributions. (A)** Read noise map in electrons for a back-illuminated sCMOS camera (Fusion BT; Hamamatsu) after generating a SD of the intensity projection image of a timelapse of 100 images in the dark. Vertical stripes are apparent (due to pixel offset issue) and a horizontal bright line at the bottom of the image, due to a glow effect for a full chip acquisition. **(B)** Read noise distribution for a Flash4 v3 camera (Hamamatsu) and comparison of blemish correction on read noise for pixel correction ON (magenta) or OFF (blue). Inset table: measured values of read noise (RMS) and DSNU for both cases. **(C)** DSNU map (dark offset of 100 images averaged) for two different sCMOS Andor cameras in the central part (500 × 500 pixels). Different patterns are distinguished, depending on the sensor corrections (column, pixel) and the vertical juxtaposition of the two matrices of this kind of sCMOS. Scale bar in intensity (digital) values. **(D)** Distribution of DSNU values calculated from different dark image dataset sizes for a sCMOS zyla 4.2 camera (Andor). Using a 100 frames dataset (dashed red line), the calculated DSNU is 0.438 e⁻ while for 9,000 frames the DNSU is 0.395 e⁻.

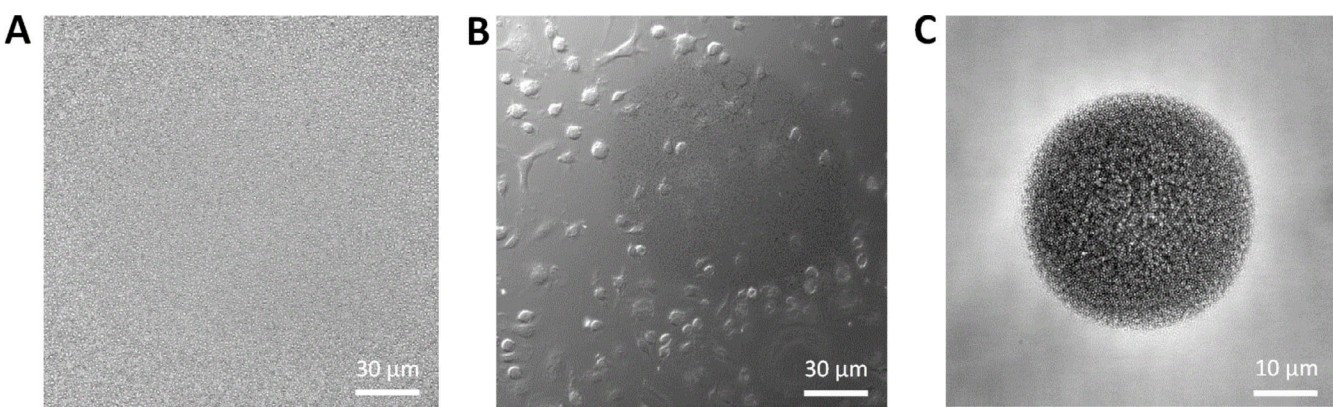

Figure S15. **Condensation effect on the camera chip. (A)** Condensation is present with small crystals that are present in the whole chip surface (sCMOS camera). **(B)** Effect of imaging a biological sample (with DIC transmitted light) with a camera presenting condensation issues (darker round zone in the middle of the image; sCMOS camera). **(C)** Extreme condensation case where imaging is impossible (image acquired with transmitted light with an EMCCD camera).

