## [Peer Review File · The Journal of Cell Biology]

Quality assessment in light microscopy for routine use through simple tools and robust metrics

Orestis Faklaris, Leslie Bancel-Vallee, Aurelien Dauphin, Baptiste Monterroso, Perrine Frere, David Geny, Tudor Manoliu, Sylvain de Rossi, Fabrice Cordelieres, Damien Schapman, Roland Nitschke, Julien Cau, and Thomas Guilbert

Corresponding Author(s): Orestis Faklaris, CNRS

Review Timeline:

Submission Date:	2021-07-15
Editorial Decision:	2021-08-30
Revision Received:	2022-04-04
Editorial Decision:	2022-05-17
Revision Received:	2022-07-20
Editorial Decision:	2022-08-12
Revision Received:	2022-08-24

Monitoring Editor: Joerg Bewersdorf

Scientific Editor: Tim Spencer

Transaction Report:

DOI: <https://doi.org/10.1083/jcb.202107093>

August 30, 2021

Re: JCB manuscript #202107093

Dr. Orestis Faklaris
CNRS
Biocampus
1919 route de Mende
Montpellier 34293
France

Dear Dr. Faklaris,

Thank you for submitting your manuscript entitled "Long-term quality assessment in light microscopy through accessible tools and reliable metrics". The manuscript was assessed by expert reviewers, whose comments are appended to this letter. We invite you to submit a revision if you can address the reviewers' key concerns, as outlined here.

You will see that the reviewers are largely enthusiastic about the study but have raised a large number of issues - most of them relatively minor and/or straightforward - that we feel must be addressed before the paper would be suitable for publication in JCB. One common theme from all three reviewers is the need to reorganize the paper to make it clearer and more accessible to the general cell biologist (as opposed to just core facility experts). We also hope that you will be able to address each of the reviewers' other concerns accordingly.

GENERAL GUIDELINES:

Text limits: Character count for a Tool is < 40,000, not including spaces. Count includes title page, abstract, introduction, results, discussion, acknowledgments, and figure legends. Count does not include materials and methods, references, tables, or supplemental legends.

Figures: Tools may have up to 10 main text figures. Figures must be prepared according to the policies outlined in our Instructions to Authors, under Data Presentation, <https://jcb.rupress.org/site/misc/ifora.xhtml>. All figures in accepted manuscripts will be screened prior to publication.

Supplemental information: There are strict limits on the allowable amount of supplemental data. Tools may usually have up to 5 supplemental figures but, given the nature of this paper, we can allow you to exceed this number. Up to 10 supplemental videos or flash animations are allowed. A summary of all supplemental material should appear at the end of the Materials and methods section.

As you may know, the typical timeframe for revisions is three to four months. However, we at JCB realize that the implementation of social distancing and shelter in place measures that limit spread of COVID-19 also pose challenges to scientific researchers. Lab closures especially are preventing scientists from conducting experiments to further their research. Therefore, JCB has waived the revision time limit. We recommend that you reach out to the editors once your lab has reopened to decide on an appropriate time frame for resubmission. Please note that papers are generally considered through only one revision cycle, so any revised manuscript will likely be either accepted or rejected.

Thank you for this interesting contribution to Journal of Cell Biology. You can contact us at the journal office with any questions, cellbio@rockefeller.edu.

Sincerely,

Joerg Bewersdorf, PhD
Monitoring Editor
Journal of Cell Biology

Tim Spencer, PhD
Executive Editor
Journal of Cell Biology

Reviewer #1 (Comments to the Authors (Required)):

This is an important and timely manuscript on quality control and monitoring of fluorescence microscopes. The basic data and analysis are good and of high value to the cell biology community and to anybody using microscopy in a quantitative manner.

The authors have created a unique opportunity within the French Imaging community that has long been noticed internationally and is now replicated in the US and Canada by BINA with support from CZA (aka Metrology Suitcase). The proposed scope of QC work is driven by this long-term project and by considerations of practical (day-to-day) feasibility and impact. Consequently, the proposed levels of QC fall short of an ideal, all-encompassing solution. However, perfection can be the death of good and I truly believe that the proposed scope can be deemed reasonable and justified by many experts. The ability to survey microscope performance independent of manufacturer support and under real life conditions is essential to establish community standards around reporting on imaging-based experiments and data sets as well as to arrive at reasonable expectations about QC documentation fluorescence microscopes in the research setting. As such this manuscript will impact both the imaging facility community and individual labs in a substantial manner. I mention this background to underscore how timely and needed this work is.

In its current form the manuscript has several major shortcomings. All of them seem redeemable and doing so will increase the accessibility of this work for members of the community. To state this bluntly though: there is work to be done from where the manuscript is today to reach publication grade. I truly hope you find my frank and many pointers below helpful.

General comments

- 1) It is imperative that the data underlying this manuscript is deposited publicly, including images as they were recorded and tables with data on all objectives, stages and cameras with enough information to be used by others for metanalysis. For instance, being able to look specifically at a certain objective to compare against the same one in a readers' facility, or to track data from several objectives and look at trends over time are beyond the scope of a manuscript like this one but are of highest value to the community.
- 2) The manuscript is built around the MetrologJ-QC Fiji plugin, but a supplement that provides user instructions is missing as is a detailed description of the working of the analysis done in the plugin. Even if that information is available elsewhere it needs to be linked and integrated with this manuscript. The reason is that most readers (especially those that will profit the most from this work) will not be able to follow the overwhelming amount of implicit assumptions around analysis in the current form of this manuscript.
- 3) The manuscript is frequently imprecise in wording or using terminology that is unclear. I give specific examples in the "detailed comments" section of this review. The manuscript also reads very different in different sections and needs a deep edit to ensure terms (or vocabulary as the authors refer to) are used in the same way by all authors and that the language is precise in all sections. The manuscript has been edited and Ulrike Boehm is mentioned as "critical reader", but the manuscript needs another, deep edit to be readable by a general audience. I do consider myself a fair expert on the topics but had to make assumptions multiple times per page and had a very hard and frustrating time with the text. I spell this out to explain to the authors that their work is highly appreciated, but not (yet) helpful to most potential readers.
- 4) The manuscript is full of statements that are not backed up by citations (and for many of the experience-based statements that is acceptable) or the cited reference is such that the point made is not easily found. It is imperative that reasoning is added for those cases there no literature exists and that for more detailed arguments the argument of the reference is made here and then cited. Otherwise, many key points will get lost to the reader as simple "opinions without backup" by the authors and diminish the value of this excellent work. In other words - dear authors please keep in mind that most readers will not be experienced facility directors or deep matter experts. Those are actually not your target audience, but you often use language that makes sense to "us" but not to those colleagues without the context of your daily work and interest. This remark also includes the many "qualifiers" used to indicate case options (e.g., "if necessary" without stating in what cases necessary applies).
- 5) From the title through arguments in the manuscript the "long-term" monitoring is a crucial element of what makes this work great. However, there is only really one figure (Fig 5 2012 to 2020 monitoring of SP5 microscope) that is truly long term and two

years' worth of PSF data presented. Stability of illumination profiles, changes in positioning precision, data on stability of different light sources (e.g. LEDs, Bulbs and light guides) or how consistent warm up times are over the life time of the light source would add a lot to the manuscript. The impression from anecdotal conversations about this project at conferences in years past has been that this group has collected data like it. It would really be great to get that in this manuscript and released with the data (see point 1).

6) The figure legends are in general not sufficient, but especially Figures 3, 4, S3, S5, 6 and S7 (makes no sense, align color code with Fig 5e) need attention. S10 is missing but might be S8?

7) Please make sure all limit conditions (green boxes) are motivated and not just dropped on the reader. Why is the box of the size and position you draw it in? See text around Figures 3 and 5 for example of magic appearance of "green box".

8) Units: Please plot figures in electrons, avoid a.u. as much as possible, add a table to the power section with power density values for your measurements, grouped by microscope type. Examples are listed in the detailed comments.

Detailed comments

9) The abstract needs a minor work over. "what biology requires from microscopy" is neither precise, nor exactly what the work is about (minimizing variance from an experimental source that can easily be addressed and is widely neglected) nor a suitable opening. On page 2 you have a great summary: "Our intend here is to provide...". That is indeed a great point for a summary/abstract...

10) Point iii) in the abstract needs to clearly refer to the analysis of QC data, the current wording could just as well imply a general demand (and a good and justified one for that matter, but this is not the place for it).

11) Last sentence of abstract "... first extensive ..." is maybe a bit too much, see your own bibliography, "... of A light microscope..." is too narrow, you are looking at several and general microscope performance; "... monitor THE microscope performance ..." again, you are not limited to a specific single microscope. [These are examples of simple, yet annoying language edits that you have far too many of, please edit the manuscript for easy-to-read English.]

12) Introduction: first two sentences use "quantification" and "quality of the observation". Besides the philosophical joy of how quality becomes quantifiable my best guess is that you are speaking to the need for "known performance that can be quantified to assess reliability of data from an imaging source , aka microscope?"

13) Page 2 "... the microscope should be stable over time ..." I get it, but this is a rather unspecific way to state your point here and the sentence reads straightforward if let out: "To obtain consistent quality deviation from standard values (calibrated or expected performance) should be understood and corrected." My point here, adding "should be stable" or "if necessary" blurs the message and is not actually telling the reader anything helpful as the context is left to the reader. There are likely hundreds of examples like this in the manuscript that make it VERY hard to read.

14) Page 2 "This task ..." "reproducibility among labs" and "over time monitoring" are different scales. Both points are good, but the way they are combined is not helpful nor does it add clarity.

15) Page 2 "ten ... core facilities" this is the moment to boost: how many microscopes, objectives, cameras, stages, filter sets, over what time? Make it impressive and collect all the stats here.

16) This is a general comment for your consideration: Starting with "i) Lateral ..." you start an overview of the 6 plus 1 values/topics/aims of your QC proposal. Shorten each section here and adding details to the main part might be more helpful as the reader doesn't need to go back and forth in the text. Here it would be more helpful to state the topic (e.g. resolution as you do) and motivate why it is of practical importance. Have the detailed discussion as part of the Material, Method and Results section on the topic.

17) Another general comment here is that in some of the sections i) - vii) you state the exact metric you propose (not necessarily develop!) but in some cases you just state that you propose a metric but don't reveal what the metric is.

18) i) while one can argue about the role of the excitation volume in confocal microscopes, the diffraction limit observed on the detector is emission dependent. Excitation is confusing here. Also, the spots are not killed by ("...diffraction-limit die to the ...") but caused by ("...diffraction-limit DUE to the ...") the diffraction limit ;).

19) Still in i) "Due to technical limitations..." ideally provide a reference or add more information on the why. It's a true and very good point...

20) Still in i) "This study..." Validation is a big word here, and the whole sentence doesn't really add anything.

I will stop dealing with language use related examples here... . There are, however, any number of such examples on the next

20 something pages of the manuscript.

21) Within ii): add reference about the different light properties of sources you discuss. Also, this is about the only time you refer to LEDs and lamps, both are frequent light sources in WF systems, but you do not present any data on them. It would be so helpful if that could be added!!!

22) Within iii) more context on why addressing co-registration at the hardware level is preferable would be helpful. Practically co-registration is not linear across the field of view. You will need to discuss the difference between QC for an objective (best case, centered) and calibration for an experiment (FoV) beads as best-case estimator for a more complicated experimental situation. My suggestion would be to put it in the reorganized section later on, but if you decide against that strategy this would likely be the spot... . Also, the "color-pair" argument is the start of a string of imprecision about how exactly the analysis is done. You refer later to "the elegant MetrologJ" approach, but don't spell it out. Step by step is your friend here... .

23) Within iii) what is completely unclear and remains vague is if there is any z-component to the co-registration measurements?

24) Within iv) you imply that "... excitation and emission collection ..." is done. What detector is collecting the excitation? Or is this a language thing? Later in iv) you mention several sources for variance in biological experiments. Rather focus on the role of the microscope as an avoidable source of variance. The mentioning of room temp in this section would best be related to your data. At least refer to the section that contains Fig S6.

25) Within v): discuss external parameters and how they impact drift and positioning. Don't just mention that there is a multitude of impacting causes, also, you are in a unique position to comment which factors you find practically to be more relevant in your facilities! Next, home-made refers more to cakes and food. "Custom made" or "prepared in the lab" might suit you better here.

26) Within vi) what about calibration of point detectors? Is that supported in MetrologJ_QC? Are you proposing the use of a specific prior protocol? It's a big missing point! Next, one needs to read the rest of the manuscript to understand that you really propose to monitor noise levels, not perform a detector calibration. The rationale that is presented later in the discussion is sound, as you say it's too complicated at this time. But it would be worth being upfront about it and to explain how noise monitoring is helpful.

27) Data Analysis: this should be vii) and more detail be provided on the plugin. Also, an example data set would ideally be supplemental material for the manuscript! In this section it's also a good idea to list other plugins that are being used.

28) Bead sample: high density of beads is demanded for instance by PSFJ and within the physics community regarded as essential for calibration microscopes. This is a point that will need more reasoning for the low-density choice and how that possibly relates to the aim of QC over time vs. calibration for a specific experiment.

29) Acquisition protocol: the "400 low pass" is likely a 400 nm long pass dichroic?

30) Acquisition protocol: in the discussion cleaning of the lens is suggested, that should be mentioned here!

31) Acquisition protocol: what are "alternative methods" to remove saturated beads? Are those supported by plugin?

32) Acquisition protocol: you might want to be specific about the FWHM being 2.355 times sigma of the Gaussian fit. This is largely unknown to most microscope users.

33) Page 11 "Metric": it is unclear what image processing is done to "take into account the resolution in that position and angle of highest emission wavelength".

34) Page 12 "Sample": in what cases is it advisable to monitor temperature? Do you assume that the baseline of a room over a 24h cycle is known to the user and part of the decision-making process?

35) Page 12 Acquisition: here you state that you clean the dry lens (see 30 not so for oil lens). You might want to consider making a statement about the frequencies of immersion left on dry objectives here? A more detailed comment on the impact of "blinking" would also add to this section.

36) Page 13 Stage drift: You introduce only one criteria here (15 nm/min) but not all three you propose. Here or in the discussion it would be most helpful to discuss the relation between experiment and stability criteria. For instance, if distances between two signals in two colors is measured and the two signal sources drift together the relevant time domain is the delay between image in color 1 and the corresponding image in color 2.

37) Page 14: "For both stage drift and repeatability the flatness of the stages was checked." HOW? Did you put the stage on a super flat granite surface? Measure curvature?

- 38) Page 15: although discussed later, the topic of how many images is needed is central. It would be a good idea to cite the paper from Lidke and Bewersdorf labs here. It is also not clear what is really measured, or better, the text leaves open the possibility that the conversion factor is determined, although my best guess is that you use the manufacturer specification.
- 39) Page 16: what is "... an external service provider revision."?
- 40) Page 17: the sampling size of pixels for the PSF measurement should depend on the N.A. The text reads as if you assume the same pixel size if needed for a 20 or 25x 0.7 N.A. objective and a 60/63x 1.4 N.A. objective.
- 41) Page 17: "Obviously, for the DAPI ... optics ... of system are set for ... GFP"Please add more information and make sure you really mean "set" and not design, optimization or performance peak.
- 42) Page 17 refer to discussion for details on asymmetry of confocal PSF. Also, in the discussion section (page 26) it needs more detail to explain how a bead with multiple emitters can be impacted by the polarization of the excitation laser. This is well shown for single emitter's dipole moment, but for multiple emitter its less obvious. Also add why this has nothing to do with scan speed and scan direction. Did you check for any orientation correlation between beads?
- 43) Page 17: why do you restrict the work to lasers only?
- 44) Page 18: Stabilo is not a known entity everywhere. You might want to add some information of maker, color, series. While the German Edding 3000 in red does a fair job the American Sharpie does not.
- 45) Page 18 Figure S2, it is unclear if part of these profiles is from afore mentioned out of focus signal?
- 46) Page 18: how is "normalization by diagonal" done? Diagonal to what? Corner to corner, through longest distance of 90% intensity?
- 47) Page 18: How is the limit (green box in Figure 3) defined? How is it rationalized?
- 48) Page 18: What is the reason to not transfer fU completely to supplemental material? I can see value in testing it, but for the summary of your work you find its always the same as U.
- 49) Page 19: for clarity decide for use of either objective or lens. While I am no foreigner to calling my objectives lenses, it seems better to use minimal vocabulary, and you use lenses as "other parts of microscope" elsewhere in text.
- 50) Page 19: The theoretical limit for co-registration seems to be the diffraction limit, but it is unclear if it is used as "plus-minus" one diffraction limit or "plus-minus" 0.5 diffraction limits. Maybe you can add a cartoon to exemplify what your analysis steps are in detail? Also, why not use a "center of mass" or a "fitted center point" to do center-center distances in nm?
- 51) Page 20: The paragraph starting with "Before defining the metrics, ..." Is a case study and as such a good example of the motivation that should be given in the overview of parameters for QC in the introduction.
- 52) Page 20 Fig 5, All plots seem to be in normalized power. Can you give an idea of the actual power densities for the different systems? To compare between labs this value is essential... Maybe add a table with measured power and power density per microscope modality?
- 53) Page 20: Fig 7, it seems worthy to discuss the 405 nm laser in more detail. It's a striking example, do you have more such data to compile another supplemental figure? These are good examples to lead with and connect to failed biological experiments.
- 54) Page 22: active/de-active focus control, again it makes sense to discuss why no focus control is preferred for QC, but not necessarily for calibration for experiment.
- 55) Page 22: clarification is needed for the 4D projection. These are 3D projections and the white dot is the time dependent variable, so indicates the movement, correct?
- 56) Page 22: Label x and y axis in Figure 6a.
- 57) Page 22: A table with all stage models as supplement would be great.
- 58) Page 22: "... (<45min) are observed, both rates are ..." Are the rates you mean the drift speed before stabilization and after?
- 59) Page 22: "Finally, sometimes no stabilization ..." Can you replace "sometimes" with a number? How often is this happening, is it always for the same stage, or sporadic for many or all stages?

- 60) Page 22: "... year, drift of a system" Do you have examples you can show? Is "system" here equal to "stage" or is it relevant that a specific (or any) microscope was attached to the stage?
- 61) Page 23: "Any other drift parameter ..." can you give an example of the sort of serious error you predict? Possibly have an example with image data.
- 62) Page 23: Figure 7 axis label for B is missing and not even in legend...
- 63) Page 23: The example of the ASI stage with linear encoder and its replacement seems more of an example case. Possibly use in intro?
- 64) Page 24: Did you test stages repeatedly? Can you add data on stage performance over long term or stage lifetime?
- 65) Page 24: "constructor" might use "manufacturer"?
- 66) Page 24: Figure 8 (b) please plot all figures in electrons.
- 67) Page 24: Describe what "blemish correction" does here and not in discussion section. Also mention alternative names and software settings used to do this. Possibly discuss the impact of this sort of correction on Likelihood based analysis methods, as the replaced pixel is a dependent value on surrounding pixels and Poisson statistics are violated.
- 68) Page 24: Please spell out VAR. The definition in discussion section is quite good.
- 69) Page 24: figure 8, consider rescaling 8d to 75% to 150% and extend in length, 8F has lots of empty information, rescale and show 95-100% in shorter graph will provide space.
- 70) Page 25: Consider discussing the separation of Gaussian noise component here?
- 71) Page 25: "... (one should convolve the two PSFs...)" not clear what you mean. And by the way, why not use Qdots for PSFs?
- 72) Page 26: Did any of the laser modules use square fibers? Can you add data for lamps on U and C? Can you add the number of laser lines and laser combiners you measured?
- 73) Page 27: "Widefield microscopes performed apparently better as compared to confocal ... ". This finding is quite a surprise, as textbook knowledge (e.g. Stelzer in Handbook of confocal microscopy) and practical data from 4Pi configurations (Peters lab) both imply that due to the pinhole alignment the inherent colocalization is better with confocal microscopes. This is not contesting your finding but underscoring how important QC on day-to-day used microscopes really is. It would be worth reworking the text a little to clarify for which color range your finding applies, possibly compile another table here.
- 74) Page 27: Illumination power. Did you try using a beam profiler instead of a power meter to acquire power and profile in the same run? You have likely chosen the power meter for the S170C slide power meter head, but your arguments pro and contra different equipment will still be interesting to the reader. Similarly, you should spend a few words on why only laser systems are used (page 28).
- 75) Page 28: The discussion about measuring power out of the laser head vs. out of objective is very good. In many ways, though, you are making a strong case for the need to access monitoring data from within the laser combiner, laser head or microscope body, which ever might be available.
- 76) Page 28: Stage drift - "... parameters such as mechanical ... stability". Do you include optical table and damping systems here? Do you have any additional data on what optical table microscopes were on that you did QC measurements on? Could optical table data be correlated with your other QC results to see if better performing microscopes used a specific subgroup of optical tables, or tables not older than, or ... ?
- 77) Page 29: "if these change (...), the drift measurements are no longer valid." Do you have data to demonstrate?
- 78) Page 29: "To preserve intrinsic heat ...". Are you referring to heated stage insets or to the heat capacity of the stage as is or possible gradients of heat induced by the motors in the case of frequent moving?
- 79) Page 29: "... in accordance with the manufacturers' protocols ... ". Can you give an example for this and explain why you might expect a stage on a test stand to perform better than mounted on a microscope stand?
- 80) Page 29: Camera noise. As mentioned above point detector noise monitoring is also needed. You refer to two publications, but is point detector analysis supported in your plugin?

81) Page 29: the VAR metric as reverence to manufacturing specs is clever. Is the 10% tolerance based on some principal consideration, data driven or just a "reasonable" cut off?

82) Page 30: Conclusion. The argument that a limited set of test samples that are affordable and robust is key. A power meter with the slide sensor is not affordable though and demands upkeep with calibration of the power meter. As most authors seem to be members of QuaRep, can you comment on the discrepancies found in WG 1 between power meters and if you used the same power meter for all power measurements? As mentioned above, Relative power per system as a QC value is likely okay, but for cross lab comparison the power density would be needed. Please discuss this and give some data for your experiments. This might also be relevant to your discussion of PSF acquisition settings as a common power density would further equalize acquisition conditions.

83) Page 31: "... metrics complement THE ISO norm ..." Please add ISO norm number in addition to citation.

Reviewer #2 (Comments to the Authors (Required)):

General remarks:

Microscope maintenance and quality control (QC) tests can be used to ensure optimal performance, and enhance the reproducibility of imaging data. The authors aim to present straightforward tests that can be done to assess and report the performance of fluorescence microscopes. They developed automated protocols to assess microscope performance and collected an impressive set of data from several fluorescence microscopy imaging modalities from 10 different core facilities over many years. They provide protocols for assessing the following metrics: axial/lateral resolution via the PSF, uniformity of field illumination, registration shift, illumination intensity stability over time, stage drift/fidelity, and detector noise. Each of these metrics can limit performance and may have a negative impact on the fidelity of data collected with suboptimal instruments.

QC in light microscopy data is a topic that has received a lot of (much-deserved) discussion lately. Many of the QC tests used here should be standard for core facilities that house instruments used for a wide range of qualitative and quantitative applications. Many researchers vastly underestimate the need for imaging system QC and articles like this are helpful because they elucidate the issue and provide straightforward and accessible strategies and resources for addressing them. Overall, the protocols presented here are sound, and accompanied by useful tips for interpretation. Moreover, they made an admirable effort to design protocols using techniques, software, and materials that are cost-effective, open-source and not too time-consuming.

However, I found it difficult to understand the target audience for the manuscript as written. The authors often assume knowledge that is common for core facility directors and managers, but far less so among cell biologists. While the data collected is an impressive effort, I think it would be better served if the manuscript was either targeted towards core staff with clearly-stated assumed expertise, or towards cell biologists with a microscope in their lab. In addition, the manuscript is poorly referenced throughout; I provide several examples below.

The authors developed an ImageJ plug-in to automate much of the analysis, but this is not emphasized enough in the manuscript to say that this is the focus of the manuscript. There are many previous publications that provide protocols for collection of QC data from imaging systems, using very similar if not identical protocols used in this work; the most novel aspects of the analysis are therefore the plug-in and the large dataset they collected. I also found that they missed opportunities to pull potentially interesting findings from their data, and sometimes instead emphasized less interesting aspects.

The authors often overstate the importance of these types of QC measurements. While I agree that they are critical in a core facility where instruments are routinely used for a wide range of applications and specimens, they are not necessarily critical for all types of experiments and specimens. In an effort not to 'scare off' cell biologists, I think the wording should be softened and specific examples of when a test is critical - and when it is not - should be provided. Alternatively, the authors could emphasize that their tests are meant to ensure optimal performance of the microscope, and not necessarily that they are critical for all experiments.

Specific Comments

Page 2

paragraph (para) 3: Add reference Petrak & Waters, 2014, which includes protocols for several of the tests performed in this work.

"Therefore, the image quality and the subsequent quantification analysis are affected". This is an example of an overstatement. While the assessment of the PSF is critical for some experiments, many common quantitative analyses are likely unaffected by a less than ideal PSF - for example, counting nuclei or co-localization of diffuse proteins in the cytoplasm.

Page 3:

"all pixels should have the same value". I know what the authors mean here, but it is not strictly correct due to detector and

poisson noise. This type of statement should also be referenced.

"Unfortunately, illumination light source alignment and optical aberrations (from the objective lens and additionally optics included in the light path) can affect the homogeneity of the field illumination. Whereas, laser and LED light sources are coupled to the microscope with optical fibers and relay optics that help homogenize the beam's Gaussian profile, not perfectly calibrated scanners can influence the laser illumination uniformity as well as the detection uniformity of laser scanning confocal microscopes. Bulb-type sources like metal-halide, mercury or xenon arc lamps are not very homogeneous to begin with and additional optical elements like diffusers are often necessary. Furthermore, the dichroic mirror and the filter positioning in the filter cubes can impact the observed field illumination for every color channel." This is an example of insufficient references; each sentence here should include 1 or more references.

"Accurate image data quantification requires characterizing the field illumination pattern and correcting for heterogeneities if necessary." This is not the case for every type of experiment. In this case the authors do list several examples in the following sentence, but this is an overstatement even for some of the examples given (eg, tracking and segmentation may be made easier by flatfield correction, but it is not strictly required to produce accurate data).

"Previous studies investigate the field illumination measurements by defining theoretical metrics and measurement protocols" MA Model has published excellent protocols for flatfield correction that I have found to be both easy to implement and far more reproducible than plastic slides.

"a wrong alignment of the excitation sources (if several lasers or LEDs are used)". I'm not sure how the alignment excitation source would lead to registration shifts in x, y or z. Please reference and/or explain.

"Accurate correction should be performed at the subpixel level". In many cases, a simple pixel shift is more than adequate to correct for registration shifts.

"and tackling the co-registration is more efficient at the hardware level than using postprocessing." I wouldn't say this is always true. A shift coming from a wedge in the dichroics, for example, is easy to fix in software and difficult (or expensive) to correct in the microscope.

Page 4:

"Compared to past studies 11,27, we define..." Add reference 6.

"Axial drift is a typical problem when using high NA objectives with a shallow depth of field." Some knowledge is assumed here and a reference is not provided. The statement could be interpreted to mean that axial drift is more prevalent or caused by high NA objectives, when it is only that smaller drifts are more detectable due to the shallow depth of field.

"For multi-position imaging the positioning repeatability of the stage should be high enough to image the same fluorescent points in the field of view (FOV)". This seems like an arbitrary parameter that is not necessary for every experiment.

Page 5:

In general, the detector noise section often pools CCD and sCMOS cameras together, rather than explaining and referencing their differences. This is important because of the difference in noise distribution (ie, read noise measurements per frame vs per pixel). The authors do state this elsewhere and demonstrate it in Fig 8, but since it greatly influences how the data should be collected and analysed it should be emphasized.

"This detector parameter is always provided in the manufacturer's datasheet." While this is true for the research grade cameras from the major manufacturers most often used, it is not true for all CMOS and CCD detectors marketed towards imaging research. The authors should also explain what a datasheet is and how to find it.

"Quantitative image data analysis, especially for maximum likelihood estimation methods, requires individual pixel characteristics, such as variation of the read noise". This is rarely true when poisson noise, not read noise, is limiting.

"In that same protocol, the dark offset signal value is measured (a value arbitrarily adjusted by the manufacturer to allow access of the complete read noise), together with the dark signal nonuniformity (DSNU), which is a spatial noise. The DSNU refers to the non-uniformity between pixels, is measured in electrons and is the standard deviation of all pixels dark offset" These are examples that require expertise to understand and should be referenced. The authors should also emphasize the difference between sCMOS and CCD with regard to 'spatial noise'.

"We used sub-resolution beads for measuring the PSF of the high numerical aperture (NA) lenses". Reference Hiraoka, Sedat, Agard 1990; book chapters by P Goodwin.

Page 6

"diluted solution". Diluted how much and in what?

Page 7

"induced by the potential refractive index mismatch between mounting medium" Reference Hell et al 2011.

"Concerning the lateral Z sampling, the Shannon-Nyquist criterion was also considered which gave for instance 0.15 and 0.2 μm step size for the highest NA objectives (1.4) for confocal and wide-field microscopy respectively." Provide reference for how these values are calculated.

"1 Airy unit (AU)": Define and reference

"a weird PSF": Please be more specific and provide references for aberrations.

Page 8

"Alternative options can be used to discard any saturated beads from image." Like what?

"to avoid aberrations". Please be more specific and provide explanation/references for increased aberrations at the periphery of the FOV.

Page 9:

"Aberrant "PSF" (non-bead features that are considered as beads using the algorithms)". I found this confusing. Do you mean that the algorithm may detect spots that do not originate from the bead, and they can be removed computationally?

"We used a highly concentrated (1mM) dye solution (rhodamine) between a glass slide and coverslip for this specimen." Reference Model.

Page 10

"the maximum and minimum intensity". Does this mean the highest and lowest value pixel? Does noise or the presence of a hot pixel ever result in an inaccurate analysis?

I was surprised the authors didn't suggest averaging a set of images to reduce noise, as is the standard procedure for computational correction of inhomogeneous illumination. Did the authors find this was not necessary for their analysis? If the max & min pixel intensities were used, averaged images should be more accurate.

"The user can choose to discard or not a saturated image" There must be some limits to this - what if all the pixels are saturated?

"Whenever saturation occurs in a few isolated pixels, noise may be removed using a Gaussian blur of sigma = 2". Another example of a statement that should include a reference.

Page 11

Sample section: How does one determine the appropriate sized bead for the imaging system?

"Alternatively, some characterization of evenness of the chromatic aberration can be performed, using different beads locations within the field of view." Provide a reference.

"Images with more than one bead can be analyzed". Is there an upper limit to the number of beads the plugin can handle?

"Different strategies for the evaluation of illumination power stability exist." Provide references.

"measures the power at the sample plane" should be "near the sample plane"

Page 12

"The laser should be warmed up and switched on at least one hour before the measurements." This seems arbitrary. How was the time period determined?

"eliminate laser noise" How was this done?

"The blanking function" What is this? Provide explanation or reference.

"short, mid, and long time scale". Provide some suggestions as to how these time frames should be chosen.

Page 13

"We preferred using a 60/63x high NA oil immersion objective, to be close to the experimental conditions when overnight experiments are started and measured precise XYZ variations." I wasn't sure what this means? Please clarify.

"if all three axis displacement speeds are less than 15 nm/min for each axis, a drift that can be neglected in diffraction limited experiments." This may not be negligible in longer term acquisitions.

Page 14

"10x or 20x dry objectives" Explain how the objective lens affects optimal bead size.

"the flatness of the stages was checked" How?

Page 15

"When possible, camera shutter was closed to avoid any unwanted light contributions and thus unwanted shot and read noise". Most modern cameras do not have physical shutters. Does this mean the camera cap should be screwed on? Or light sent to another port? Also, read noise will be present regardless of whether or not light reaches the detector - read noise is an inherent property of the readout amplifier on the camera, and, unlike shot noise, it is independent of signal. Based on the rest of the section, it's clear the authors know this and so this was probably just phrased in a confusing way, but it could be misleading to readers who aren't very familiar with the different types of noise in a digital image and where they come from.

"The detector exposure time was set to short values (e.g. 10 ms) to avoid Dark Signal Non Uniformity (DSNU) contribution" Provide a reference.

"(dark noise is considered equal to zero for this short acquisition time)" Provide a reference. What was the range of dark noise for the cameras used in this study?

Page 16

"After many tests, repositioning, and tilt adjustment of the camera, the symmetry was recovered (measurement of December 2019 and later on)." The reader will likely be as interested in how to correct for identified problems as they are in methods of revealing them. The authors give several anecdotal observations like this one, but little explanation as to why the method worked. These should either be left out or explained further. Eg, what does "repositioning" mean?

"(<https://svi.nl/NyquistRate>)". Please provide peer-reviewed publications as references, rather than (or in addition to) manufacturers' websites.

"The brightest beads should be used, and this is the reason we recommend using single labeled beads" I also use Invitrogen beads, and have found that single labeled fluospheres are brighter than multi-color beads. But is this universally true for all bead manufacturers'?

"The beads diameter should be below the resolution limit." Or perhaps the resolution required for the experiment?

Page 17

"We observed that if the SBR is low, then the precision on the FWHM calculation is low (Figure S1)". This is of course expected - please provide a reference.

"as the objectives and the optics of the whole system are set for the GFP channel" Were the microscopes used set up in this way? Or is this in reference to typical manufacturers' specifications? If the latter, provide a reference.

"Attention was paid to the SBR in LSCM images." Why state this specifically for LSCM? Isn't the same true for all modalities? "due to spherical aberrations" How was this determined? Provide a reference.

"Figure 2c shows the Lateral Asymmetry Ratio (LAR), defined as the ratio between the smallest and highest of both x and y experimental FWHM." Of an individual bead? Or an averaged population?

Page 18

"The thin layer of a fluorescence dye gives a more precise illumination pattern and can be more convenient for shading corrections for tile scan acquisitions" Why? Provide a reference.

"For spinning disk, the pinhole crosstalk" Provide a reference

"We performed the flat illumination measurements for 130 objective lenses for a wide range of magnifications (5x-100x)". On how many microscopes, using how many different types of light sources? I think it equally (if not more) informative to parse this data out by type of light source, as I would guess they would have the largest effect.

"Wide-field setups show a high dispersion of C and U values. Finally, the analyzed spinning disk setups had less dispersed values closer to the ideal 100%." Why? Is this due to the type of light source? Or the imaging modality? Were different types of objective lenses (eg, correction for aberration) used?

"Most systems designs involve the 405 laser line passing through a different fiber/lightpath than "visible" laser lines. The alignment is different and explains these differences." Provide a reference.

Page 19

"Our co-registration study collected data from setups of the three microscopy techniques using over 70 different lenses" Many other components can affect co-registration. How many different filters, dichroics, multi camera systems were included? Were there any trends associated with specific components?

"We believe it is harder to achieve a high SBR, as needed to further correctly identify beads and estimate their center coordinates, using these tiny beads." This is surprising, given how commonly diffraction-limited beads are used to measure and correct for registration shifts. This data should be included.

"while the combinations with the Blue are more often associated with poor co-registration" Is this consistent with the manufacturers tolerance for the lenses used?

"However, spinning-disk confocal systems are more often associated with insufficient chromatic correction with 63% of the measured ratios below 1, compared to LSCM or wide-field workstations with 70% and 84%, respectively." Is this difference based solely on imaging modality? Are there any other components common between the spinning disks but not the others? If I'm not mistaken, Neofluar refers to fluorite, not achromat, lenses.

"63x 1.4 NA Leica Plan-APO objective" & "Hence, this objective is not recommended, and better-corrected lenses (lambda blue for Leica with better correction for the near UV light)..." I would be more conservative about recommending against the use of a particular make & model of lens without more rigorous study.

Page 20

"the filter set cube was not correctly aligned" What does this mean and how was it identified?

"A similar cube equivalent case was associated with 0.19 and 0.23 B-G and G-R ratios, confirming that the issue came from the misalignment of the cube components (dichroic mirror and filters) and was not related to some type of poor objective correction." Isn't this most often due to a wedge in a dichroic or emission filter, rather than an alignment issue?

"far from the theory". I think you mean far from the ideal.

"upstream/downstream of the light path" Please give examples of upstream and downstream components that can cause chromatic shifts.

"The G-FR couple is slightly worse than the equivalent G-R that is expected as the wavelength difference is higher" Expected based on theory or manufacturers tolerance? Please reference.

Illumination power stability section - reference Swedlow et al 2002

Page 21

"Some other lasers have lower STAB (<97%) and higher standard deviation (0.2), such as the Argon 488nm laser line of Figure 8b." I believe this should be Figure 5b.

"As highlighted in Figure 5e". I believe this should be Figure 5f.

"Figure 5d shows that there is a correlation between the power and temperature fluctuations" should be changed to "that, in some cases, there is a correlation"

Page 22

"i) Stage drift". This section covers XY & Z drift. Z drift can result from sources unrelated to the stage.

"but brightfield observation of various features is possible as well" What type of brightfield features work? Do they need to be diffraction limited?

Page 23

"When the velocity before and after stabilization is identical and really small (lower than 15 nm per minute, which is the theoretical resolution of a high NA objective lens after 10+ min), the drift can be considered negligible (high precision experiments, e.g. super-resolution purposes)." This amount of drift may be problematic for localization microscopy, which can take hours to acquire.

"Any other drift parameter combination should be rejected or taken into account during the image analysis step because it can lead to serious misinterpretation." Again, this is not true for every type of experiment.

"When x-y linear encoders are mounted on the x-y plates of the stage, then the repeatability is further improved." Provide a reference.

"below the usual maximal 1 um manufacturer's specifications". This depends on the make & model, and would likely be better for linear encoded stages.

"It is not easy to define limit values for repositioning, as it highly depends on the stage type. For instance, for the two ASI stages having linear encoders, although they have the same characteristics and were mounted on the same microscope, they present different repositioning values." Should this be "individual stage", given the next sentence?

Page 24

"Other parameters can influence the repeatability here, like the proper fixing of the sample, the waiting time (if it exists) for each position, the acceleration of the stage and the temperature variations." These are indeed important parameters. Were any of them kept constant in this study, in some or all of the data collection?

"for 100 frames" Why was this number chosen? It seems like more than enough for CCD, but insufficient for CMOS.

The difference in read noise between CCD and CMOS is particularly important here. This should be explained in more detail and references provided.

Page 25

"The observed wide-field higher dispersion of the measured/theoretical axial resolution ratio, as compared to both confocal types, could be associated with some spherical aberration issues" This is only one possible explanation. Other aberrations and vibration are other common sources.

"In our case we use 0.175 nm beads that give slightly larger PSF values compared to point source PSF (one should convolve the two PSFs for the theoretical values)." This should be 175 nm.

Page 26

"considering optimal conditions such as shot-noise free confocal imaging" should be "considering optimal, though unobtainable, conditions such as shot-noise free confocal imaging"

"However, we propose, as defined using experimental data from Figure 3a distributions, that Centering above 40% and Uniformity above 50% can be considered acceptable" Again, this is very dependent on the experiment.

Page 27

"This clearly shows that, when observed, shifts are likely to be associated with imaging of near UV excited dyes." Were violet corrected lenses included in the study, and if so did they perform better?

"while going through the spinning-disk unit and the microlens array of the disks" Please explain or reference how the microlenses could cause chromatic shifts.

"showing that the microscopes with all their components are most adapted for imaging at that range of the visible spectrum" Should be "the microscopes we tested"

"gives hints on how to correct data to achieve some qualitative, rough comparison" I'm not sure what is meant here, but it sounds dangerous. Image corrections should be made with the same rigor as any other data manipulation.

Page 29

"we considered this value as our experimental tolerance value" Please explain the rationale behind choosing this value.

"Warm pixels (i.e., pixels showing significantly higher signal than the average pixels) are more frequent with sCMOS cameras" Provide a reference.

"Here we define a hot pixel as a pixel that does not respond to light." This may be confusing, as 'hot pixel' (in my experience) is used to describe a consistently bright pixel.

Figures

1b. This data is for one objective only. Was similar data collected for other objectives?

1c. Was the acquisition adjusted to maintain SNR at different pixel size?

Fig 2. While the experimental/theoretical ratio should help normalize for differences in lens NA, it would be interesting to know how correction for aberration affects these data.

Fig 3. It would be interesting to see how the type of light source (including how the light source is delivered into the light path) affects these data. I would also like to see how the sensor size and FOV affect uniformity.

Fig 4. Some lenses are labeled Plan-Fluor. What are the others?

Fig 6. This data does not differentiate between stabilization of stage drift and stabilization of the sample on the stage. This should be made clear.

Fig 8b. Is blemish correction a feature of the camera, acquisition software or performed post-acquisition?

Fig 8c legend. For clarity, use 'intensity value' rather than digital number, since intensity value is used elsewhere in the manuscript.

Fig 9. I had a hard time interpreting this figure. Please provide a more detailed explanation of the purpose/message.

Fig S1. Signal-to-noise ratio as well as SBR would be useful in interpreting these data.

Fig S2. An explanation and reference should be provided for what appears to be a disk speed/camera sync artifact in these images.

Fig S3. Again, what types of light sources are represented here?

Minor comments:

Page 2, last paragraph: change 'die' to 'due'

Page 3, para 3: change 'additionally' to 'additional'

Throughout the manuscript the authors refer to point scanning confocals as "confocal" and spinning disk confocals as "spinning disk". Since spinning disks are confocals too, perhaps stick to acronyms, such as LSCM and SDCM.

Reviewer #3 (Comments to the Authors (Required)):

This manuscript, "Long-term quality assessment in light microscopy through accessible tools and reliable metrics", details six areas for quality control (QC) measurements of light microscope imaging systems, provides protocols for making the necessary measurements for each and gives specific values for "acceptable" performance in each QC area. This work builds on that of the Confocal ISO from 2019 and QUAREP-LiMi project. Having clearly defined equipment standards is important for reproducibility and robustness in light microscopy experiments.

Large scale feedback

This topic is very important and the authors have clearly put a lot of careful thought and diligent work into this manuscript. The amount of data and the range of sources is amazing. The information provided in this manuscript is a great framework around which to organize the evaluation and monitoring of instrument capability, which is in turn a great foundation for reproducible and robust microscopy experiments. I very much appreciate the summary table at the end and the attempt at pulling all the information together with the heuristic map.

However, I had a hard time getting through the manuscript for two main reasons: The overall organization and that it has a large number of words (two challenges I am usually guilty of).

Having the intro, methods, results and discussion for each QC item separated into those headers made it really hard for me to keep track of the important details for each QC item. I had to keep flipping back and forth to keep continuity and answer questions and still felt uneasy about my understanding. I also feel there is a lot of duplication because of this physical separation of the same topic. Perhaps the manuscript would be easier to read with an organization more like this (though doesn't have to be this exactly):

-Short aggregate intro (already there).

- Maybe include info on Shannon-Nyquist here since it comes up in a lot of places? If not, have an explanation and/or references of what it is in the first mention

-Headers = QC items (Lateral and Axial Resolution, Field Illumination, etc)

- Sub-headers = Intro/Methods/Results/Discussion

- Short aggregate discussion pulling together evidence for impact of QC that are presented in each results section.

- Conclusion (already there)

For the dense text, there are a few broad things that I think will shorten it and/or make it read more easily. First is the reorganization outlined above--I suspect replicated information will get deleted. Two others are evaluation of passive voice versus active voice (sometimes one is shorter than the other; instead of "We conclude X is...", does "X was..." work?) and significantly reducing (but not eliminating!) transitional phrases (such as "of note," etc). Lastly, using fewer qualifying words (see next item) can help streamline arguments.

I found several instances where qualifying words (such as "enough", "negligible", "often", etc.) were used with no follow up on what they relate to quantitatively (just like here where I don't give a number for "several"). Two examples of where specification is missing are detailed here, but keep an eye out for it throughout the manuscript:

- Page 16 very top when talking about setting up the PFS metrics. You state "robust enough" and "kept negligible" then say "1 to 5%" and "4 to 6%" is small. Can you address the question Why are those good values? This might be something to address

more broadly in the introduction.

- Page 18, middle. Results section about figure 3. "Centering value was often close to 100% using LSM". Estimating from the graph, it isn't clear that the distribution along Centering is different between the modalities. It would be helpful if "often" had a number.

It is not always clear to me what criteria are used to define the "acceptable" values. For several of the QC items it seems like "what most of our measurements fit into". While it is not bad for that to be a component of the definition, having discussion and/or guidance about the quantitative impact of being within or outside of the acceptable range would be helpful.

Medium scale feedback

- Illustrations of the parameters being measured in each QC test would be really helpful for readers to understand what is being measured and compared. I can envision this either all in one figure or as a first panel in the separate figures. This might be more complicated than it's worth for some measurements, but try for as many as possible.

-The methods for sample preparation need more detail on implementation. Specifically, it would be helpful for the bead sample preparation to include dilution factor or final percent solids in the final dilution used to achieve the bead density specified. This will help readers have a place to start and maximize likelihood of success.

- For the Field Illumination QC, it is unclear to me why a high uniformity but low centration (upper left of the graph) is considered less optimal than moderate uniformity and moderate centration (lower left corner of the green square), exp after looking at the example images in Figure S3. Very low intensity differences will give more robust quantification than highly centered but more variable illumination. Discussion of the impact on quantification, perhaps with an example, would be helpful.

- Page 16, 2nd paragraph regarding the system where one PFS axis changed after a service visit. State only the information about the data ("10% decrease in X-axis FWHM but no change for y-axis FWHM after an external service provider revision"). Save the recommendation about checking before and after service work for the discussion since that applies to multiple measurements.

- Page 17, lower half of the page, shorten the sentence about difficulty in getting Nyquist with WF and SDC. "...meeting Shannon-Nyquist criterion is limited by the fixed physical pixel size of the detector array."

- Figure S2

- panel C. Based on the images in this panel, the Stabilo looks like it has substantially worse Uniformity than the Chroma slide. Based on panel A, I'm assuming the chroma slide and rhodamine film would be similar. The text says the Stabilo is sufficiently equivalent to the Chroma slide, but does it pass your own Uniformity standards?

- In the figure legend, Stabilo autocorrected to "stabile" a couple times. I did not check the main text to see if it happened there, too.

- Figure S3.

- The gradient values for the example images are not legible. Do they all use the same gradient LUT so a color scale key could be used? If possible, remove the text from the images-it is too small to read and obscures the pattern a bit. It is unclear where the Center is.

- The Heuristic map is not a familiar schematic for me and was not immediately intuitive (and sadly Google did not easily provide a definition to help). 1-2 sentences for the figure legend about its goal/purpose and/or how to read it would be helpful.

- Table 2. Would it get too bust to add a column of recommend measurement interval? Or maybe add a separate table for that?

Odds and Ends

- Mention the summary table in the introduction

- Describe the goal of Shannon-Nyquist sampling at the first mention. Not everyone who will read this will be familiar.

- "automatize" sounds unusual to my US "ear"; I suspect you are using it where I would use "automate". I leave it to the authors to determine their preference.

- Figure 8, panel c images. Is the color scale the same for both? If so, only need one. If not, please make them the same.

- Figure 8, panel f. Should there be scale bars here?

- To help link this with other great articles about robust quantification, instrument characterization and experimental reproducibility, it might be helpful to readers to cite the following papers in the intro or discussion. I leave it up to the authors to determine if this is within the scope of their manuscript.

- John M. Heddleston, Jesse S. Aaron, Satya Khuon, Teng-Leong Chew; A guide to accurate reporting in digital image acquisition - can anyone replicate your microscopy data?. J Cell Sci 15 March 2021; 134 (6): jcs254144. doi: <https://doi.org/10.1242/jcs.254144>

- Montero Llopis P, Senft RA, Ross-Elliott TJ, Stephansky R, Keeley DP, Koshar P, Marqués G, Gao YS, Carlson BR, Pengo T, Sanders MA, Cameron LA, Itano MS. Best practices and tools for reporting reproducible fluorescence microscopy methods. Nat Methods. 2021 Jun 7. doi: 10.1038/s41592-021-01156-w. Epub ahead of print. PMID: 34099930.

This manuscript covers an important topic for biological research, that of measuring the light microscope system with which imaging data is collected and evaluating the performance thereof. It represents an immense effort to collect empirical data and

evaluate appropriate criteria. I believe the indicated adjustment will significantly improve impact and usability. Great work and I hope to see a revised version!

--Wendy Salmon

Whitehead Institute (until Sept 2)

University of North Carolina at Chapel Hill School of Medicine starting Sept 13

MRI – Biocampus
1919 route de Mende
34293, Montpellier
France

March 31, 2022

Dear JCB Editorial Board,

We would like to submit a manuscript to Journal of Cell Biology on the Quality Control for instruments in light microscopy.

We would like to warmly thank the three reviewers for their enthusiasm in reviewing this work. After reading the numerous comments we decided to work on the manuscript and make a significant number of modifications.

Following the proposition of the reviewers, and in agreement with the editors, we modified the structure of the paper by dedicating one chapter (with Introduction, Methods, Results, and Discussion) per examined quality control parameter. We think it makes it clearer, easier to read with fewer repetitions. We limit the number of characters at ~59,000, a number that exceeds the 40,000 recommended for a Tool paper, but due to the nature of our paper and all the fields that it treats, we did our best to limit the characters and describe and comment sufficiently the results.

We tried to make the paper more accessible to a larger community, made up mainly of biologists who use light microscopes, and not just core facility staff. Some metrics and limit values may be slightly different in this new version. Our aim was to bring them closer to the needs of biologists, and particularly to adapt the limit values to the level of requirement of each experiment.

Please find below the answers to the reviewers' comments point by point. The reviewers' comments are in black color, our responses in blue.

The authors declare no competing financial interests.

Thank you for your consideration on this manuscript and we are looking forward to hearing from you.

Sincerely,

Orestis Faklaris

Responses to Reviewers

Reviewer 1

Reviewer #1 (Comments to the Authors (Required)):

This is an important and timely manuscript on quality control and monitoring of fluorescence microscopes. The basic data and analysis are good and of high value to the cell biology community and to anybody using microscopy in a quantitative manner.

The authors have created a unique opportunity within the French Imaging community that has long been noticed internationally and is now replicated in the US and Canada by BINA with support from CZA (aka Metrology Suitcase). The proposed scope of QC work is driven by this long-term project and by considerations of practical (day-to-day) feasibility and impact. Consequently, the proposed levels of QC fall short of an ideal, all-encompassing solution. However, perfection can be the death of good and I truly believe that the proposed scope can be deemed reasonable and justified by many experts. The ability to survey microscope performance independent of manufacturer support and under real life conditions is essential to establish community standards around reporting on imaging-based experiments and data sets as well as to arrive at reasonable expectations about QC documentation fluorescence microscopes in the research setting. As such this manuscript will impact both the imaging facility community and individual labs in a substantial manner. I mention this background to underscore how timely and needed this work is.

In its current form the manuscript has several major shortcomings. All of them seem redeemable and doing so will increase the accessibility of this work for members of the community. To state this bluntly though: there is work to be done from where the manuscript is today to reach publication grade. I truly hope you find my frank and many pointers below helpful.

First of all, we would like to deeply thank Reviewer 1. The critical proofreading work done here is really impressive. We realized immediately that the message of the submitted article was perfectly understood and we hope that our modifications for this new submission meet the majority of the sometimes demanding but each time relevant raised remarks.

General comments

1) It is imperative that the data underlying this manuscript is deposited publicly, including images as they were recorded and tables with data on all objectives, stages and cameras with enough information to be used by others for meta-analysis. For instance, being able to look specifically at a certain objective to compare against the same one in a readers' facility, or to track data from several objectives and look at trends over time are beyond the scope of a manuscript like this one but are of highest value to the community.

The point raised here by R1 is indeed of major importance. The public repository availability of the entire dataset is a subject that the authors have discussed on numerous occasions. There are several reasons that prevent us from doing this work. Having all data available requires that the classification and categorization (by core-facility, by type of microscope, by manufacturer brand, by type of device, by date...) is carefully done. We would also like to be rigorous and check metadata to ensure the relevance of a possible meta-analysis. Obviously, some of our data is several years old, from a time when metadata integrity verification still seemed a long way off. Finally, if the verification of the completeness of the metadata and the classification of the data seem feasible, the realization of this work would require a considerable amount of time that the authors do not have at their disposal. Typically, software like the Caterina Strambio-De-Castillia's team Micro-Meta App could help prevent this kind of impediment in the near future.

Subsequently, the authors propose to make available a public repository under a download link with reference images for each described protocol. Readers of this work will thus be able to test the MetroloJ_QC plugin and the defined metrics and tolerance values.

2) The manuscript is built around the MetrologJ-QC FIJI plugin, but a supplement that provides user instructions is missing as is a detailed description of the working of the analysis done in the plugin. Even if that information is available elsewhere it needs to be linked and integrated with this manuscript. The reason is that most readers (especially those that will profit the most from this work) will not be able to follow the overwhelming amount of implicit assumptions around analysis in the current form of this manuscript.

We provided more information on the plugin. We added a general main figure (Figure 2) and supplementary figures that describe the plugin workflow for each of the seven treated topics. We also added a link to a github page in the text, where both the plugin and the detailed user manual are available.

3) The manuscript is frequently imprecise in wording or using terminology that is unclear. I give specific examples in the "detailed comments" section of this review. The manuscript also reads very different in different sections and needs a deep edit to ensure terms (or vocabulary as the authors refer to) are used in the same way by all authors and that the language is precise in all sections. The manuscript has been edited and Ulrike Boehm is mentioned as "critical reader", but the manuscript needs another, deep edit to be readable by a general audience. I do consider myself a fair expert on the topics but had to make assumptions multiple times per page and had a very hard and frustrating time with the text. I spell this out to explain to the authors that their work is highly appreciated, but not (yet) helpful to most potential readers.

The manuscript is extensively modified in order to be easier to read, both in its general structure and in the vocabulary used, attempting to avoid implicit assumptions as much as possible. We also had help from native language proofreaders to improve the overall consistency and readability of this work.

4) The manuscript is full of statements that are not backed up by citations (and for many of the experience-based statements that is acceptable) or the cited reference is such that the point made is not easily found. It is imperative that reasoning is added for those cases there no literature exists and that for more detailed arguments the argument of the reference is made here and then cited. Otherwise, many key points will get lost to the reader as simple "opinions without backup" by the authors and diminish the value of this excellent work. In other words - dear authors please keep in mind that most readers will not be experienced facility directors or deep matter experts. Those are actually not your target audience, but you often use language that makes sense to "us" but not to those colleagues without the context of your daily work and interest. This remark also includes the many "qualifiers" used to indicate case options (e.g., "if necessary" without stating in what cases necessary applies).

We added more citations for the treated statements. For the cases where no literature exists, we added reasoning or supplementary results, as is the case of supplementary figure S1, where some additional data is shown to give reasoning to the statement of the PSF position dependence on the FOV aberrations. We often modified our language and tried to make it easier to read for our target audience composed mainly of biologists.

5) From the title through arguments in the manuscript the "long-term" monitoring is a crucial element of what makes this work great. However, there is only really one figure (Fig 5 2012 to 2020 monitoring of SP5 microscope) that is truly long term and two years' worth of PSF data presented. Stability of illumination profiles, changes in positioning precision, data on stability of different light sources (e.g. LEDs, Bulbs and light guides) or how consistent warm up times are over the life time of the light source would add a lot to the manuscript. The impression from anecdotal conversations about this project at conferences in years past has been that this group has collected data like it. It would really be great to get that in this manuscript and released with the data (see point 1).

R1 brings up a very important point here. The notion of long-term monitoring is indeed difficult to sustain if the curves presented over several years cannot be provided for all the measurements.

Thus, if we provide them for the measurement of PSF and for laser stability, other metrics are more difficult to present in this form, mainly for practical reasons. For field homogeneity, for example, two joint metrics form the tolerance value, which makes it difficult to trace over time. We added bulb light sources stability over time (Figure S11) and camera read noise stability over years (Figure 9D).

6) The figure legends are in general not sufficient, but especially Figures 3, 4, S3, S5, 6 and S7 (makes no sense, align color code with Fig 5e) need attention. S10 is missing but might be S8?

We showed special attention and reformulated all figure legends.

7) Please make sure all limit conditions (green boxes) are motivated and not just dropped on the reader. Why is the box of the size and position you draw it in? See text around Figures 3 and 5 for example of magic appearance of "green box".

Once again, this point is of great importance. The authors deeply think that the microscopy community, as the QUAREP-LiMi consortium for instance, has to start working with tolerance values defined experimentally over time, through a big volume of data. The common goal is to redefine these values, probably with more subtle ranges, dependent on the type of imaging experiments.

We explained more where these green boxes come from and added information on the influence of the proposed values on biological imaging experiments.

8) Units: Please plot figures in electrons, avoid a.u. as much as possible, add a table to the power section with power density values for your measurements, grouped by microscope type. Examples are listed in the detailed comments.

Figures are now in electrons.

All plots, indeed, are normalized (A.U.) to ensure an understandable visualization of the data. The goal in this work is to follow fluctuations of power intensity of sources over time. We were not very interested in the absolute values. We understand that the absolute power value detected through an objective at fixed nominal power is an interesting data to compare the performance of different systems, but that is not our purpose here.

Detailed comments

9) The abstract needs a minor work over. "what biology requires from microscopy" is neither precise, nor exactly what the work is about (minimizing variance from an experimental source that can easily be addressed and is widely neglected) nor a suitable opening. On page 2 you have a great summary: "Our intend here is to provide...". That is indeed a great point for a summary/abstract...

As mentioned previously, we revised the whole text. The abstract introduction is now the following : "Reproducibility in light microscopy represents a goal to achieve to improve research results, but standardized quality guidelines monitoring microscope stability performance over time are still lacking. Here, we provide affordable hardware and open source software tools, rigorous protocols, and define reference values to assess quality control metrics for the most common fluorescence light microscopy modalities." taking into account your comment.

10) Point iii) in the abstract needs to clearly refer to the analysis of QC data, the current wording could just as well imply a general demand (and a good and justified one for that matter, but this is not the place for it).

We reformulated the whole abstract. The analysis of QC data is described by the phrase: "We designed the MetroJ_QC ImageJ/Fiji java plugin to incorporate the metrics and automate analysis."

11) Last sentence of abstract "... first extensive ..." is maybe a bit too much, see your own bibliography, "... of A light microscope..." is too narrow, you are looking at several and general microscope performance; "... monitor THE microscope performance ..." again, you are not limited to a specific single microscope. [These are examples of simple, yet annoying language edits that you have far too many of, please edit the manuscript for easy-to-read English.]

All along the manuscript, we tried to either simplify or be more specific where needed. For example, the last sentence of the abstract is now: "Measurements collected from ten imaging facilities allow us to propose an extensive characterization..." We paid attention to all these language details and we hope to have improved the reading of the manuscript.

12) Introduction: first two sentences use "quantification" and "quality of the observation". Besides the philosophical joy of how quality becomes quantifiable my best guess is that you are speaking to the need for "known performance that can be quantified to assess reliability of data from an imaging source , aka microscope?

In the new version of the manuscript these phrases do not exist.

13) Page 2 "... the microscope should be stable over time ..." I get it, but this is a rather unspecific way to state your point here and the sentence reads straightforward if let out: "To obtain consistent quality deviation from standard values (calibrated or expected performance) should be understood and corrected." My point here, adding "should be stable" or "if necessary" blurs the message and is not actually telling the reader anything helpful as the context is left to the reader. There are likely hundreds of examples like this in the manuscript that make it VERY hard to read.

Once again, thank you for pointing out these examples. We tried to improve the manuscript in that way.

14) Page 2 "This task ..." "reproducibility among labs" and "over time monitoring" are different scales. Both points are good, but the way they are combined is not helpful nor does it add clarity.

We deleted the "over time monitoring" of that phrase and focused here on the "reproducibility among labs". The phrase becomes: "It is however essential in research to ensure quantifiable results and reproducibility among laboratories"

15) Page 2 "ten ... core facilities" this is the moment to boost: how many microscopes, objectives, cameras, stages, filter sets, over what time? Make it impressive and collect all the stats here.

We took into consideration the comment and added the requested infos. The phrase becomes: "we collected experimental data from over ten light microscopy core facilities, checking 101 objectives, 37 laser sources, 22 stages, 32 cameras, and automated the acquisition and analysis procedures".

16) This is a general comment for your consideration: Starting with "i) Lateral ..." you start an overview of the 6 plus 1 values/topics/aims of your QC proposal. Shorten each section here and adding details to the main part might be more helpful as the reader doesn't need to go

back and forth in the text. Here it would be more helpful to state the topic (e.g. resolution as you do) and motivate why it is of practical importance. Have the detailed discussion as part of the Material, Method and Results section on the topic.

As we strongly modified the paper, the Introduction part is now more general. The six topics are described in their own section.

17) Another general comment here is that in some of the sections i) - vii) you state the exact metric you propose (not necessarily develop!) but in some cases you just state that you propose a metric but don't reveal what the metric is.

In the new version of the manuscript we tried to make the introduction of each section uniform, by naming the metrics but not developing them further.

18) i) while one can argue about the role of the excitation volume in confocal microscopes, the diffraction limit observed on the detector is emission dependent. Excitation is confusing here. Also, the spots are not killed by ("...diffraction-limit die to the ...") but caused by ("...diffraction-limit DUE to the ...") the diffraction limit ;).

This is now corrected :)

19) Still in i) "Due to technical limitations..." ideally provide a reference or add more information on the why. It's a true and very good point...

We added more information at the Discussion part.

20) Still in i) "This study..." Validation is a big word here, and the whole sentence doesn't really add anything.

You are right, this is now rephrased in the general introduction.

I will stop dealing with language use related examples here... . There are, however, any number of such examples on the next 20 something pages of the manuscript.

21) Within ii): add reference about the different light properties of sources you discuss. Also, this is about the only time you refer to LEDs and lamps, both are frequent light sources in WF systems, but you do not present any data on them. It would be so helpful if that could be added!!!

We added the reference and we made a new figure (figure 4C) that takes into account the type of the source illumination, including LEDs and lamps.

22) Within iii) more context on why addressing co-registration at the hardware level is preferable would be helpful. Practically co-registration is not linear across the field of view. You will need to discuss the difference between QC for an objective (best case, centered) and calibration for an experiment (FoV) beads as best-case estimator for a more complicated experimental situation. My suggestion would be to put it in the reorganized section later on, but if you decide against that strategy this would likely be the spot... . Also, the "color-pair" argument is the start of a string of imprecision about how exactly the analysis is done. You refer later to "the elegant MetroloJ" approach, but don't spell it out. Step by step is your friend here... .

Thank you for this suggestion. We took it into account your comments, step by step, in particular by reformulating the introduction of the co-registration.

23) Within iii) what is completely unclear and remains vague is if there is any z-component to the co-registration measurements?

We cleared up this situation: "To evaluate the co-registration in x-y and z directions we used multi-labeled microspheres and defined a co-registration ratio as our metric."

24) Within iv) you imply that "... excitation and emission collection ..." is done. What detector is collecting the excitation? Or is this a language thing? Later in iv) you mention several sources for variance in biological experiments. Rather focus on the role of the microscope as an avoidable source of variance. The mentioning of room temp in this section would best be related to your data. At least refer to the section that contains Fig S6.

We changed this sentence which was unclear and we now talk about excitation stability and the "intrinsic power of the light source used under conventional excitation".

We also replaced the second phrase with the following one: "Besides sample preparation variability, fluorochrome performance or environmental parameters such as room temperature, and considering that the source stability is characterized, then the microscope can be considered as an avoidable source of variance. This illustrates the importance of source power monitoring, especially for researchers who want to compare images from samples acquired weeks apart."

25) Within v): discuss external parameters and how they impact drift and positioning. Don't just mention that there is a multitude of impacting causes, also, you are in a unique position to comment which factors you find practically to be more relevant in your facilities! Next, home-made refers more to cakes and food. "Custom made" or "prepared in the lab" might suit you better here.

In this part we have tried to make the external influences as explicit as possible. "Home-made" has been replaced by "custom-made".

26) Within vi) what about calibration of point detectors? Is that supported in MetrologJ_QC? Are you proposing the use of a specific prior protocol? It's a big missing point! Next, one needs to read the rest of the manuscript to understand that you really propose to monitor noise levels, not perform a detector calibration. The rationale that is presented later in the discussion is sound, as you say it's too complicated at this time. But it would be worth being upfront about it and to explain how noise monitoring is helpful.

Indeed, we rephrased the introduction of this part to be more upfront about what has been done.

27) Data Analysis: this should be vii) and more detail be provided on the plugin. Also, an example data set would ideally be supplemental material for the manuscript! In this section it's also a good idea to list other plugins that are being used.

In the new version of the paper, we present Data Analysis at the end of the general introduction. Then for every of the six sections we give the detailed workflow that we follow for the data analysis with our plugin, most often supported by a supplementary figure. We give the link to the example data set.

28) Bead sample: high density of beads is demanded for instance by PSFJ and within the physics community regarded as essential for calibration microscopes. This is a point that will need more reasoning for the low-density choice and how that possibly relates to the aim of QC over time vs. calibration for a specific experiment.

This commentary written by R1 allows us to clarify an important point. The work done here is intended for the characterization of a light microscope. We never considered that the collected image data could be used for any other purpose than to measure the performance of a system.

Even if it is tempting to see further, such as using certain data for the calibration of a microscope (shading correction, correction of spherical and chromatic aberrations over the entire field of view, define different level of correction for specific applications, etc.), this is not the subject of our work here.

29) Acquisition protocol: the "400 low pass" is likely a 400 nm long pass dichroic?

Exactly. This is corrected.

30) Acquisition protocol: in the discussion cleaning of the lens is suggested, that should be mentioned here!

It is added to the acquisition protocol session.

31) Acquisition protocol: what are "alternative methods" to remove saturated beads? Are those supported by plugin?

We add : "An alternative option can be used to automatically discard any saturated beads from the image using the MetroloJ_QC plugin."

32) Acquisition protocol: you might want to be specific about the FWHM being 2.355 times sigma of the Gaussian fit. This is largely unknown to most microscope users.

The FWHM calculation is indicated in the MetroloJ_QC manual, found on the github page (link given in the paper).

33) Page 11 "Metric": it is unclear what image processing is done to "take into account the resolution in that position and angle of highest emission wavelength".

We give the reference to the initial MetroloJ plugin where this processing is well explained.

34) Page 12 "Sample": in what cases is it advisable to monitor temperature? Do you assume that the baseline of a room over a 24h cycle is known to the user and part of the decision-making process?

We added a phrase in the text to explain more in what cases we need to monitor temperature : "When we observed instabilities of the source power over time, we monitored the room temperature with the TSP01 temperature probe (Thorlabs) and recorded it with the TSP01 Application software". When the baseline of a room over a 24h cycle is not known to the user, then he should measure the temperature by himself.

35) Page 12 Acquisition: here you state that you clean the dry lens (see 30 not so for oil lens). You might want to consider making a statement about the frequencies of immersion left on dry objectives here? A more detailed comment on the impact of "blanking" would also add to this section.

We actually added a sentence here: " From our experience on imaging facilities, we often observe dry lenses that require thorough cleaning due to the presence of misplaced immersion liquid". About the blanking effect, we added: "We advise to use the bleach point mode, if applicable, to avoid AOTF modulation / blanking of laser beams, which can occur between the end and the beginning of a new scan line to avoid unnecessary specimen exposure. When available, the blanking function was switched off to avoid measurement disturbance. "

36) Page 13 Stage drift: You introduce only one criteria here (15 nm/min) but not all three you propose. Here or in the discussion it would be most helpful to discuss the relation between experiment and stability criteria. For instance, if distances between two signals in two colors is measured and the two signal sources drift together the relevant time domain is the delay between image in color 1 and the corresponding image in color 2.

True. This sentence is now in the Results "tolerance value" section. The second part of this comment is now explained in the Discussion section.

37) Page 14: "For both stage drift and repeatability the flatness of the stages was checked." HOW? Did you put the stage on a super flat granite surface? Measure curvature?

We did the simplest way we know. "For both stage drift and repeatability acquisitions, the microscopes were set up on active optical tables, as for all other measurements in this work. We checked the flatness of the stages by using spirit-levels for WF and SDCM, and by imaging a mirror in reflection mode for SLCM (protocol exists to achieve values on the stage horizontality [S. You *et al.*, Microscope calibration protocol for single-molecule microscopy, OptExpress 2021])."

38) Page 15: although discussed later, the topic of how many images is needed is central. It would be a good idea to cite the paper from Lidke and Bewersdorf labs here. It is also not clear what is really measured, or better, the text leaves open the possibility that the conversion factor is determined, although my best guess is that you use the manufacturer specification.

We hope that the new formulation is clearer: « In our protocol, the read noise is measured by the fluctuation of the value of a pixel along a temporal acquisition of 100 images, without any light influence. Read noise is then considered as a temporal noise. While the quantity of 100 frames may seem over-estimated for CCD read noise measurement, it is a fair compromise for sCMOS read noise (obviously not to characterize every pixels' attributes⁸⁵), and DSNU measurements, as explained below ».

39) Page 16: what is "... an external service provider revision."?

This has been rephrased to « five days after a planned manufacturer's maintenance visit »

40) Page 17: the sampling size of pixels for the PSF measurement should depend on the N.A. The text reads as if you assume the same pixel size if needed for a 20 or 25x 0.7 N.A. objective and a 60/63x 1.4 N.A. objective.

We did not succeed to find this particular point at page 17 of the old paper version. We hope that the confusion you pointed out is now cleared up.

41) Page 17: "Obviously, for the DAPI ... optics ... of system are set for ... GFP"Please add more information and make sure you really mean "set" and not design, optimization or performance peak.

This is now rephrased to "Most conventional microscopes and commercially available optics are designed for performing better at visible wavelength " and the following reference was added "Blinton et al., Optical Considerations at Ultraviolet Wavelengths in Confocal Microscopy, Springer 1995"

42) Page 17 refer to discussion for details on asymmetry of confocal PSF. Also, in the discussion section (page 26) it needs more detail to explain how a bead with multiple emitters can be impacted by the polarization of the excitation laser. This is well shown for single emitter's dipole moment, but for multiple emitter its less obvious. Also add why this has nothing to do with scan speed and scan direction. Did you check for any orientation correlation between beads?

Thank you for this relevant remark. To avoid adding confusion to our biologist readers, the authors do not wish to go into the details of such a phenomenon. We discussed with Leica, Nikon and Zeiss to better investigate this effect and the three companies confirmed to us that the origin is the linear polarization of the source under high NA focus. We believe that references provided in the text are sufficient to explain this effect.

43) Page 17: why do you restrict the work to lasers only?

We present laser sources because they are the ones used for two of the three microscopy techniques that we study here. For LED and lamps we do not have as much data as for

lasers, that is why we do not present them. We added though our long-term observations for X-Cite sources, as a supplementary figure (Figure S11) and gave some explanations in the text.

44) Page 18: Stabilo is not a known entity everywhere. You might want to add some information of maker, color, series. While the German Edding 3000 in red does a fair job the American Sharpie does not.

To simplify the message and try to be more straightforward, the authors decided to take out the Stabilo part.

45) Page 18 Figure S2, it is unclear if part of these profiles is from aforementioned out of focus signal?

We present in Figure S4C (former figure S2) the intensity distribution with the depth.

46) Page 18: how is "normalization by diagonal" done? Diagonal to what? Corner to corner, through longest distance of 90% intensity?

It is the corner to corner diagonal. The normalization is done by dividing by the width and length of the image. We changed the text to: "The size of the field of view does not influence the metric value (normalization by the geometrical center position of the image)".

47) Page 18: How is the limit (green box in Figure 3) defined? How is it rationalized?

This is the tricky part. Figure 4A shows that most cases are located inside this green box. The boundaries have been defined empirically and some limit cases are also shown in the same figure. The influence of these limit values to biological imaging are illustrated in Figure 4D,E. These are new figures that we added in the new version of the manuscript to better illustrate the influence of QC and the limit values to biological imaging.

48) Page 18: What is the reason to not transfer fU completely to supplemental material? I can see value in testing it, but for the summary of your work you find its always the same as U.

After discussions with the author team of the paper, we decided to completely remove this part that was redundant and not useful for the global comprehension.

49) Page 19: for clarity decide for use of either objective or lens. While I am no foreigner to calling my objectives lenses, it seems better to use minimal vocabulary, and you use lenses as "other parts of microscope" elsewhere in text.

Thank you for this suggestion that helps a lot in the clarity of the global manuscript. We have chosen to use the term objective throughout this work.

50) Page 19: The theoretical limit for co-registration seems to be the diffraction limit, but it is unclear if it is used as "plus-minus" one diffraction limit or "plus-minus" 0.5 diffraction limits. Maybe you can add a cartoon to exemplify what your analysis steps are in detail? Also, why not use a "center of mass" or a "fitted center point" to do center-center distances in nm?

We added Figure S8, that describes the co-registration workflow using MetroloJ_QC. In that figure we give an illustration (Fig. S8d) demonstrating the calculation of the reference (i.e., theoretical or diffraction limit distance) and the 'real' distances.

51) Page 20: The paragraph starting with "Before defining the metrics, ..." Is a case study and as such a good example of the motivation that should be given in the overview of parameters for QC in the introduction.

Thank you for this suggestion, we adapted this sentence and added it to the general introduction.

52) Page 20 Fig 5, All plots seem to be in normalized power. Can you give an idea of the actual power densities for the different systems? To compare between labs this value is essential... Maybe add a table with measured power and power density per microscope modality?

Please look at the reply of comment number 8.

53) Page 20: Fig 7, it seems worthy to discuss the 405 nm laser in more detail. It's a striking example, do you have more such data to compile another supplemental figure? These are good examples to lead with and connect to failed biological experiments.

We do not find a 405 nm laser representation in Figure 7. We think that you make reference to Figure S6, showing a long warming up time of the 405 nm diode laser. We added a warming up measurement for another microscope with four lasers where there is a laser showing similar long warming up time (updated Figure S9).

54) Page 22: active/de-active focus control, again it makes sense to discuss why no focus control is preferred for QC, but not necessarily for calibration for experiment.

We completed this sentence this way: "If present, it can be very useful to activate hardware and software focus corrections for the calibration of experiments. However, to properly monitor the xy behavior of stages and the z stability, the z correction has to be deactivated to allow raw drift evaluation and thus maintain a sensitivity to the environmental parameters which must be taken into account."

55) Page 22: clarification is needed for the 4D projection. These are 3D projections and the white dot is the time dependent variable, so indicates the movement, correct?

The sentence has been modified to : "We characterized first qualitatively the drift by observing the time projection of the maximum intensity 3D projection of a typical bead track (Fig. 7A)."

56) Page 22: Label x and y axis in Figure 6a.

Thanks for the comment, we added the x,y labels.

57) Page 22: A table with all stage models as supplement would be great.

Unfortunately, we only provide an inventory of the number and the type of stages. We tried during this work to stigmatize the manufacturers as little as possible.

58) Page 22: "... (<45min) are observed, both rates are ..." Are the rates you mean the drift speed before stabilization and after?

Absolutely. To be clearer, we rephrased to: "When short stabilization time (<45min) is observed, V_b and V_a values are close."

59) Page 22: "Finally, sometimes no stabilization ..." Can you replace "sometimes" with a number? How often is this happening, is it always for the same stage, or sporadic for many or all stages?

We rephrased it to: "Finally, for one of the 18 tested setups, we found no stabilization during the whole 15h time-lapse period." We explain more about this case in the discussion section.

60) Page 22: "... year, drift of a system" Do you have examples you can show? Is "system" here equal to "stage" or is it relevant that a specific (or any) microscope was attached to the stage?

We do not show data of this in this work. We believe that it is of little relevance to show two completely different drift behaviors of the same system on a single figure. The "system" corresponds to the whole stage + microscope ensemble.

61) Page 23: "Any other drift parameter ..." can you give an example of the sort of serious error you predict? Possibly have an example with image data.

Here again, it seems difficult to us to simply represent a loss of focus or a lateral drift. We added "especially when the focus is lost."

62) Page 23: Figure 7 axis label for B is missing and not even in legend...

You are absolutely right. We added the axis label.

63) Page 23: The example of the ASI stage with linear encoder and its replacement seems more of an example case. Possibly use in intro?

We indeed added it in the intro: "We compared the acquired experimental values to the stage specifications found in the manufacturer's datasheets."

64) Page 24: Did you test stages repeatedly? Can you add data on stage performance over long term or stage lifetime?

Yes, the Methods part indicates : "We then examined the reproducibility of the measurements by performing acquisitions for five consecutive times on five stepper motors stages (Fig. 8C). We observed that the variability of the results was stage dependent (stages 2 and 4 showed lower variability than the rest of the stages)".

65) Page 24: "constructor" might use "manufacturer"?

You are right, "constructor" has been replaced by "manufacturer" in the whole document.

66) Page 24: Figure 8 (b) please plot all figures in electrons.

It is done.

67) Page 24: Describe what "blemish correction" does here and not in discussion section. Also mention alternative names and software settings used to do this. Possibly discuss the impact of this sort of correction on Likelihood based analysis methods, as the replaced pixel is a dependent value on surrounding pixels and Poisson statistics are violated.

Blemish correction has been defined in the Methods section.

Thank you for this suggestion about the impact of this kind of corrections, it is now in the discussion section.

68) Page 24: Please spell out VAR. The definition in discussion section is quite good.

It is done.

69) Page 24: figure 8, consider rescaling 8d to 75% to 150% and extend in length, 8F has lots of empty information, rescale and show 95-100% in shorter graph will provide space.

It is done and it is clearer now (updated Figure 9).

70) Page 25: Consider discussing the separation of Gaussian noise component here?

We think we have now clearly explained the differences between sCMOS and CCD noise components. As our target public is biologists, we do not wish to enter too much into details here, to let the text be read easily. We discuss correction algorithms in the discussion part.

71) Page 25: "... (one should convolve the two PSFs...)" not clear what you mean. And by the way, why not use Qdots for PSFs?

These brackets have been removed. Although Qdots can be good point sources, here we only considered the most common material available in laboratories.

72) Page 26: Did any of the laser modules use square fibers? Can you add data for lamps on U and C? Can you add the number of laser lines and laser combiners you measured?

We added Figure 4C which now classifies U and C calculations according to the light sources. No square fibers or diffuser were used.

73) Page 27: "Widefield microscopes performed apparently better as compared to confocal ...". This finding is quite a surprise, as textbook knowledge (e.g. Stelzer in Handbook of confocal microscopy) and practical data from 4Pi configurations (Peters lab) both imply that due to the pinhole alignment the inherent colocalization is better with confocal microscopes. This is not contesting your finding but underscoring how important QC on day-to-day used microscopes really is. It would be worth reworking the text a little to clarify for which color range your finding applies, possibly compile another table here.

We thank the reviewer for this comment. We went through all the results and performed a closer statistical analysis, with the dispersion and mean values for the ratio distribution for all modalities (insets of figure 5A, B). We observe that when comparing the DAPI channel with the other visible ones, the wide field performs better. We think it comes from the separate optical paths for the 405nm beam for the confocal techniques. But when comparing the visible channels, then the LSCM shows better coregistration results. We comment on that in the Discussion section.

74) Page 27: Illumination power. Did you try using a beam profiler instead of a power meter to acquire power and profile in the same run? You have likely chosen the power meter for the S170C slide power meter head, but your arguments pro and contra different equipment will still be interesting to the reader. Similarly, you should spend a few words on why only laser systems are used (page 28).

We compiled data from 10 core-facilities, which implies that we all used the same equipment. S170C slide power meter was the best value for money power-meter, although a beam profiler could have given terribly interesting data/results. We added a long-term stability measurement graph on LED source (Figure S11) and we modified the text accordingly.

75) Page 28: The discussion about measuring power out of the laser head vs. out of objective is very good. In many ways, though, you are making a strong case for the need to access monitoring data from within the laser combiner, laser head or microscope body, whichever might be available.

Thank you.

76) Page 28: Stage drift - "... parameters such as mechanical ... stability". Do you include optical table and damping systems here? Do you have any additional data on what optical table microscopes were on that you did QC measurements on? Could optical table data be correlated with your other QC results to see if better performing microscopes used a specific subgroup of optical tables, or tables not older than, or ... ?

This is a very good point. We actually didn't investigate in that direction, but it makes sense to evaluate the influence of optical tables (active and passive) on QC measurements. Maybe a project for QUAREP-LiMi WG6 ?

77) Page 29: "if these change (...), the drift measurements are no longer valid." Do you have data to demonstrate?

No we do not. We changed to: " Drift measurements only make sense if environmental conditions (e.g., temperature, humidity, air flow, warming time of the stage) are kept stable. As external changes (e.g., period of the year) influence the drift results, time monitoring of the drift is necessary."

78) Page 29: " To preserve intrinsic heat ...". Are you referring to heated stage insets or to the heat capacity of the stage as is or possible gradients of heat induced by the motors in the case of frequent moving?

Indeed, we are referring to the heat capacity of the stage.

79) Page 29: "... in accordance with the manufacturers' protocols ... ". Can you give an example for this and explain why you might expect a stage on a test stand to perform better than mounted on a microscope stand?

We changed this part to: "Last but not least, our measurements were performed considering the stage as part of the microscope and under "real" conditions, which is not always in accordance with the manufacturers' protocols and datasheet values where the stage is considered as an isolated item. In the latter case, the stage is not subjected to the conditions of temperature change in the context of live cell imaging or drifts."

80) Page 29: Camera noise. As mentioned above point detector noise monitoring is also needed. You refer to two publications, but is point detector analysis supported in your plugin?

Not yet. Since the plugin has been developed for our purpose, point detector analysis is not supported yet. We hope to obtain good results with QUAREP WG2 to be able to implement it.

81) Page 29: the VAR metric as reverence to manufacturing specs is clever. Is the 10% tolerance based on some principal consideration, data driven or just a "reasonable" cut off?

It is a cut off that we suggested from our data collection. It is quite difficult to obtain bad values since cameras we tested are still efficient despite their age.

82) Page 30: Conclusion. The argument that a limited set of test samples that are affordable and robust is key. A power meter with the slide sensor is not affordable though and demands upkeep with calibration of the power meter. As most authors seem to be members of QuaRep, can you comment on the discrepancies found in WG 1 between power meters and if you used the same power meter for all power measurements? As mentioned above, Relative power per system as a QC value is likely okay, but for cross lab comparison the power density would be needed. Please discuss this and give some data for your experiments. This might also be relevant to your discussion of PSF acquisition settings as a common power density would further equalize acquisition conditions.

Although R#1 is perfectly right on these statements, our work here does not treat all these aspects.

Actually, all these suggestions seem to fit perfectly the context of Quarep-Limi and the corresponding WG1 and the authors deeply hope that they will be addressed soon. As mentioned before in these comments, cross lab comparison is not our goal here. Our goal is more the democratization of long term quality data assessment. We hope to contribute modestly to this objective through this work.

83) Page 31: "... metrics complement THE ISO norm ..." Please add ISO norm number in addition to citation.

It's done.

Reviewer #2 (Comments to the Authors (Required)):

General remarks:

Microscope maintenance and quality control (QC) tests can be used to ensure optimal performance, and enhance the reproducibility of imaging data. The authors aim to present straightforward tests that can be done to assess and report the performance of fluorescence microscopes. They developed automated protocols to assess microscope performance and

collected an impressive set of data from several fluorescence microscopy imaging modalities from 10 different core facilities over many years. They provide protocols for assessing the following metrics: axial/lateral resolution via the PSF, uniformity of field illumination, registration shift, illumination intensity stability over time, stage drift/fidelity, and detector noise. Each of these metrics can limit performance and may have a negative impact on the fidelity of data collected with suboptimal instruments.

QC in light microscopy data is a topic that has received a lot of (much-deserved) discussion lately. Many of the QC tests used here should be standard for core facilities that house instruments used for a wide range of qualitative and quantitative applications. Many researchers vastly underestimate the need for imaging system QC and articles like this are helpful because they elucidate the issue and provide straightforward and accessible strategies and resources for addressing them. Overall, the protocols presented here are sound, and accompanied by useful tips for interpretation. Moreover, they made an admirable effort to design protocols using techniques, software, and materials that are cost-effective, open-source and not too time-consuming.

However, I found it difficult to understand the target audience for the manuscript as written. The authors often assume knowledge that is common for core facility directors and managers, but far less so among cell biologists. While the data collected is an impressive effort, I think it would be better served if the manuscript was either targeted towards core staff with clearly-stated assumed expertise, or towards cell biologists with a microscope in their lab. In addition, the manuscript is poorly referenced throughout; I provide several examples below.

The authors developed an ImageJ plug-in to automate much of the analysis, but this is not emphasized enough in the manuscript to say that this is the focus of the manuscript. There are many previous publications that provide protocols for collection of QC data from imaging systems, using very similar if not identical protocols used in this work; the most novel aspects of the analysis are therefore the plug-in and the large dataset they collected. I also found that they missed opportunities to pull potentially interesting findings from their data, and sometimes instead emphasized less interesting aspects.

The authors often overstate the importance of these types of QC measurements. While I agree that they are critical in a core facility where instruments are routinely used for a wide range of applications and specimens, they are not necessarily critical for all types of experiments and specimens. In an effort not to 'scare off' cell biologists, I think the wording should be softened and specific examples of when a test is critical - and when it is not - should be provided. Alternatively, the authors could emphasize that their tests are meant to ensure optimal performance of the microscope, and not necessarily that they are critical for all experiments.

The authors would like to warmly thank Reviewer 2 for his very pertinent remarks throughout this work.

A big effort was made in the rewriting of this paper and we hope that it now appears in a more clear way that we address ourselves to microscope users who are biologists and not only to facility staff.

We gave more emphasis to the ImageJ plugin that we developed, by adding more information in the main text, we made figure 2 presenting the plugin and gave supplementary figures with explanations for each measured QC field when using the plugin.

We also tried to remove and modify as much as possible the phrases of the text that focused on cases when the quality control measurements are outside the defined limit values. We tried to link the results and the limit values with biologic imaging experiments that make sense for every microscope user with a biological background.

Specific Comments

Page 2

paragraph (para) 3: Add reference Petrak & Waters, 2014, which includes protocols for several of the tests performed in this work.

This reference is now added, thanks.

"Therefore, the image quality and the subsequent quantification analysis are affected". This is an example of an overstatement. While the assessment of the PSF is critical for some experiments, many common quantitative analyses are likely unaffected by a less than ideal PSF - for example, counting nuclei or co-localization of diffuse proteins in the cytoplasm.

This specific point was pointed out by the other reviewers. We made substantial modifications on this manuscript. We hope that they are in the right direction. We added many similar examples to soften the general tone which could be considered "alarmist".

Page 3:

"all pixels should have the same value". I know what the authors mean here, but it is not strictly correct due to detector and Poisson noise. This type of statement should also be referenced.

You are absolutely right. The sentence is now changed to : "Ideally, all pixels should have values very close to each other, considering the intrinsic Poisson noise of sensors."

"Unfortunately, illumination light source alignment and optical aberrations (from the objective lens and additionally optics included in the light path) can affect the homogeneity of the field illumination. Whereas, laser and LED light sources are coupled to the microscope with optical fibers and relay optics that help homogenize the beam's Gaussian profile, not perfectly calibrated scanners can influence the laser illumination uniformity as well as the detection uniformity of laser scanning confocal microscopes. Bulb-type sources like metal-halide, mercury or xenon arc lamps are not very homogeneous to begin with and additional optical elements like diffusers are often necessary. Furthermore, the dichroic mirror and the filter positioning in the filter cubes can impact the observed field illumination for every color channel." This is an example of insufficient references; each sentence here should include 1 or more references.

All these statements seem to be *a priori* knowledge to the authors but you are absolutely right, each sentence needs to be referenced. So we add:

For LSCM, lasers are coupled to the microscope with optical fibers and relay optics and spatial filtering can be used to homogenize the beam Gaussian profile. [Laser Sources for Confocal Microscopy, DOI: 10.1007/978-1-4757-5348-6_5].

Scanners that are not perfectly calibrated can influence the laser illumination uniformity as well as the detection uniformity of LSCMs. [The Intermediate Optical System of Laser-Scanning Confocal Microscopes, DOI: 10.1007/978-0-387-45524-2_9]

For WF techniques, bulb-type sources like metal-halide, mercury or xenon arc lamps are not very homogeneous and additional optical elements like diffusers are often necessary.

[Super-resolution imaging of multiple cells by optimized flat-field epi-illumination, <https://doi.org/10.1038/nphoton.2016.200>]

Furthermore, the dichroic mirror and the filter positioning in the filter cubes can have an impact on the observed field illumination for each channel. [The Intermediate Optical System of Laser-Scanning Confocal Microscopes, DOI: 10.1007/978-0-387-45524-2_9]

"Accurate image data quantification requires characterizing the field illumination pattern and correcting for heterogeneities if necessary." This is not the case for every type of experiment. In this case the authors do list several examples in the following sentence, but this is an overstatement even for some of the examples given (eg, tracking and segmentation may be made easier by flatfield correction, but it is not strictly required to produce accurate data).

In order to be more specific, this sentence has been changed to "Accurate image intensity quantification over the entire field of view requires characterizing the field illumination pattern and correcting for heterogeneities if necessary".

"Previous studies investigate the field illumination measurements by defining theoretical metrics and measurement protocols" MA Model has published excellent protocols for flatfield correction that I have found to be both easy to implement and far more reproducible than plastic slides.

We added the MA Model reference, thank you for the suggestions. However, MA Model et al. apply shading correction. It is not our purpose here since we don't aim to calibrate a flatfield correction but only characterize it.

"a wrong alignment of the excitation sources (if several lasers or LEDs are used)". I'm not sure how the alignment excitation source would lead to registration shifts in x, y or z. Please reference and/or explain.

We added a reference here: [R.M. Zucker, 2006]

"Accurate correction should be performed at the subpixel level". In many cases, a simple pixel shift is more than adequate to correct for registration shifts.

This sentence is removed.

"and tackling the co-registration is more efficient at the hardware level than using postprocessing." I wouldn't say this is always true. A shift coming from a wedge in the dichroics, for example, is easy to fix in software and difficult (or expensive) to correct in the microscope.

Again, the authors rephrased this sentence in the Discussion : "co-registration shifts of a microscope should be understood by the user, who should be aware of the type of the objective correction type and execute experiments accordingly. Co-registration corrections should be performed preferentially at the hardware level, avoiding as much as possible image post-processing."

Page 4:

"Compared to past studies 11,27, we define..." Add reference 6.

We added this reference.

"Axial drift is a typical problem when using high NA objectives with a shallow depth of field." Some knowledge is assumed here and a reference is not provided. The statement could be interpreted to mean that axial drift is more prevalent or caused by high NA objectives, when it is only that smaller drifts are more detectable due to the shallow depth of field.

The sentence has been changed to "using high NA objectives, small axial drifts are likely to be detected due to the shallow depth of field." and we added ref [S. Inoué, in "Foundations of Confocal Scanned Imaging in Light Microscopy", Handbook of Biological Confocal Microscopy, 2nd ed. Pawley].

"For multi-position imaging the positioning repeatability of the stage should be high enough to image the same fluorescent points in the field of view (FOV)". This seems like an arbitrary parameter that is not necessary for every experiment.

This is perfectly true. We changed this sentence into: "For multi-position experiments, a key parameter is the stage repeatability, i.e., the system's ability to reposition to the same point."

Page 5:

In general, the detector noise section often pools CCD and sCMOS cameras together, rather than explaining and referencing their differences. This is important because of the difference

in noise distribution (ie, read noise measurements per frame vs per pixel). The authors do state this elsewhere and demonstrate it in Fig 8, but since it greatly influences how the data should be collected and analysed it should be emphasized.

We think that after the modifications we now clearly explain the differences between sCMOS and CCD noise components in the Methods part. We agree that it was important to do so.

"This detector parameter is always provided in the manufacturer's datasheet." While this is true for the research grade cameras from the major manufacturers most often used, it is not true for all CMOS and CCD detectors marketed towards imaging research. The authors should also explain what a datasheet is and how to find it.

Thank you for this suggestion. We now explain this in the Methods section of the stage part where we first use the datasheet term: "This value usually appears in the datasheet of the stage which is a document containing all parameter values which are measured before the purchase for that item ideally or for this item category. This information is useful to properly characterize the device at the installation and over time."

"Quantitative image data analysis, especially for maximum likelihood estimation methods, requires individual pixel characteristics, such as variation of the read noise". This is rarely true when poisson noise, not read noise, is limiting.

We hope that this sentence is now in a better context in the Methods section: "This difference in architecture of the reading method results in lower average photodiode read noise for sCMOS compared to CCD cameras, while for sCMOS, read noise is no longer constant across the chip⁸⁰⁻⁸³. This must be taken into consideration when using image analysis algorithms (e.g. single-molecule localization) on images acquired with sCMOS cameras^{84,85}. Also, some quantitative image data analysis, like maximum likelihood estimation methods, requires individual pixel characteristics, such as variation of the read noise⁸⁵"

"In that same protocol, the dark offset signal value is measured (a value arbitrarily adjusted by the manufacturer to allow access of the complete read noise), together with the dark signal nonuniformity (DSNU), which is a spatial noise. The DSNU refers to the non-uniformity between pixels, is measured in electrons and is the standard deviation of all pixels dark offset" These are examples that require expertise to understand and should be referenced. The authors should also emphasize the difference between sCMOS and CCD with regard to 'spatial noise'.

We believe that the new structure of this chapter now emphasizes the CCD / sCMOS differences for spatial and temporal noises.

"We used sub-resolution beads for measuring the PSF of the high numerical aperture (NA) lenses". Reference Hiraoka, Sedat, Agard 1990; book chapters by P Goodwin.

It's done.

Page 6

"diluted solution". Diluted how much and in what?

We added the necessary information. Now the phrase is as follows: "The beads were vortexed to avoid aggregates and diluted in distilled water to achieve a density of 10^6 beads/ml".

Page 7

"induced by the potential refractive index mismatch between mounting medium" Reference Hell et al 2011.

We added the reference.

"Concerning the lateral Z sampling, the Shannon-Nyquist criterion was also considered which gave for instance 0.15 and 0.2 μm step size for the highest NA objectives (1.4) for confocal

and wide-field microscopy respectively." Provide reference for how these values are calculated.

We added reference: The Pixilated Image - R.H. Webb & C.K. Dorey - in confocal handbook (ed. Pawley, J.B.).

"1 Airy unit (AU)": Define and reference

We added the following text : "For LSCM imaging the pinhole was set to 1 Airy unit (AU), a setting at which the pinhole size matches the diameter of the central Airy disk of the diffraction pattern to assess the standard performance of a confocal microscope, keeping a good compromise between signal intensity collection and resolution." The reference Cox et al. 2004 was added.

"a weird PSF": Please be more specific and provide references for aberrations.

We now changed "a weird PSF" to "a degraded PSF" and we illustrate typical examples in fig. 3.

Page 8

"Alternative options can be used to discard any saturated beads from image." Like what?

To be more specific, we rephrase as : "An alternative option can be used to discard automatically any saturated beads from the image using the MetroloJ_QC plugin."

"to avoid aberrations". Please be more specific and provide explanation/references for increased aberrations at the periphery of the FOV.

We added ref [S.W. Hell & E.H.K. Stelzer - Lens aberrations in Confocal fluorescence microscopy - in Handbook of confocal microscopy, ed. Pawley, J.B.] and we illustrate that effect in figure S1.

Page 9:

"Aberrant "PSF" (non-bead features that are considered as beads using the algorithms)". I found this confusing. Do you mean that the algorithm may detect spots that do not originate from the bead, and they can be removed computationally?

Exactly. We rephrased the phrase into : "If the algorithm detects spots that do not originate from the beads, they can be removed computationally using the Gaussian fitting parameter R^2 for each FWHM measurement."

"We used a highly concentrated (1mM) dye solution (rhodamine) between a glass slide and coverslip for this specimen." Reference Model.

We give now more precisions, the phrase is as follows: "We used a highly concentrated (1mM) dye solution (rhodamine G6 – Sigma-Aldrich) between a glass slide and coverslip for this specimen."

Page 10

"the maximum and minimum intensity". Does this mean the highest and lowest value pixel?

Does noise or the presence of a hot pixel ever result in an inaccurate analysis?

I was surprised the authors didn't suggest averaging a set of images to reduce noise, as is the standard procedure for computational correction of inhomogeneous illumination. Did the authors find this was not necessary for their analysis? If the max & min pixel intensities were used, averaged images should be more accurate.

To be more specific on this point, we added this text : "Only one frame is required. A supplementary step of averaging the intensities of several images can be used for calibrating the flat field correction and creating a shading correction image, necessary for tile scan

imaging^{53,54}. In our case we focus on characterizing the field illumination and a single image is sufficient.”

"The user can choose to discard or not a saturated image" There must be some limits to this - what if all the pixels are saturated?

In the acquisition protocol, the authors precise that “The acquisition parameters were set to take full advantage of the detector dynamic range and avoid saturation.” We believe that this statement is clear enough.

"Whenever saturation occurs in a few isolated pixels, noise may be removed using a Gaussian blur of sigma = 2". Another example of a statement that should include a reference.

Once again, this is perfectly true. We give the reference: P Bankhead - ImageJ, 2014 - microscopist.co.uk

Page 11

Sample section: How does one determine the appropriate sized bead for the imaging system?

We now give the explanation in the “Acquisition protocol” section. We added data to compare different bead sizes for co-registration (Figure S7).

"Alternatively, some characterization of evenness of the chromatic aberration can be performed, using different beads locations within the field of view." Provide a reference.

Here we modified the text in the intro: “In this study, we proposed a protocol to evaluate co-registration for an objective (best case centered) and do not seek calibration for the entire field of view (FOV). Calibration is often useful as co-registration is not linear across the FOV⁵⁹. A spatially resolved analysis of co-registration might help to identify the region of interest of the FOV and correct the shifts (e.g. for colocalization studies, ratiometric or FRET imaging).”

We added the reference “Kozubek et al., An efficient algorithm for measurement and correction of chromatic aberrations in fluorescence microscopy, Journal of Microscopy, 2000”.

"Images with more than one bead can be analyzed". Is there an upper limit to the number of beads the plugin can handle?

Not to our knowledge. However, the plugin allows taking into account a distance constraint between the beads. If the confluence is too high so the beads are too close, they will not be processed.

"Different strategies for the evaluation of illumination power stability exist." Provide references.

Here, the meaning was not clear enough. We changed this sentence to: “A large set of tools for the evaluation of illumination power stability exist.”

"measures the power at the sample plane" should be "near the sample plane"

We corrected the text, thank you.

Page 12

"The laser should be warmed up and switched on at least one hour before the measurements." This seems arbitrary. How was the time period determined?

To gain consistency with the rest of the text, we added a "warm-up" part. We added some more data on the warm-up measurements, shown in Figure S9.

"eliminate laser noise" How was this done?

Here we actually meant that the detection is noisier at low intensity emission signals. Using 100% source power, according to the sensitivity range of the sensor, is a way to reduce source fluctuations.

Accordingly, the sentence is changed to "We set the source power at 100% or at 20% for cases where the sensors may be damaged". We added the relative reference [Laser Sources for Confocal Microscopy, DOI: 10.1007/978-1-4757-5348-6_5].

"The blanking function" What is this? Provide explanation or reference.

We added the following as reviewer R1 also asked for this point:

"We advise to use the bleach point mode, if applicable, to avoid AOTF modulation / blanking of laser beams, which can occur between the end and the beginning of a new scan line to avoid unnecessary specimen exposure. When available, the blanking function was switched off to avoid measurement disturbance"

"short, mid, and long time scale". Provide some suggestions as to how these time frames should be chosen.

We added examples like "time-lapse experiments" for mid-term or "z-stack acquisitions" for short-term.

Page 13

"We preferred using a 60/63x high NA oil immersion objective, to be close to the experimental conditions when overnight experiments are started and measured precise XYZ variations." I wasn't sure what this means? Please clarify.

We clarified this sentence: "We used a 60/63x high NA oil immersion objective to be closer to experimental conditions where a strong numerical aperture makes it possible to finely detect drifts in the three dimensions."

"if all three axis displacement speeds are less than 15 nm/min for each axis, a drift that can be neglected in diffraction limited experiments." This may not be negligible in longer term acquisitions.

In the Results part, we stated: "When V_x and V_y values are under 15nm/min and close to each other, the drift is negligible (high precision experiments, e.g. super-resolution purposes). Accumulated over time, this amount of drift may be problematic for localization microscopy (acquisition time of minutes or hours)."

Page 14

"10x or 20x dry objectives" Explain how the objective lens affects optimal bead size.

In this work, we did not study the influence of the bead sizes on the repositioning precision. This could be a project for QUAREP-LiMi WG6.

"the flatness of the stages was checked" How?

As we replied to a similar question of R#1 (com 37), we did the simplest way we know. "For both stage drift and repeatability acquisitions, the microscopes were set up on active optical tables, as for all other measurements in this work. We checked the flatness of the stages by using spirit-levels for WF and SDCM, and by imaging a mirror in reflection mode for SLCM (protocol exists to achieve values on the stage horizontality [S. You *et al.*, Microscope calibration protocol for single-molecule microscopy, OptExpress 2021])."

Page 15

"When possible, camera shutter was closed to avoid any unwanted light contributions and thus unwanted shot and read noise". Most modern cameras do not have physical shutters. Does this mean the camera cap should be screwed on? Or light sent to another port? Also, read noise will be present regardless of whether or not light reaches the detector - read noise is an inherent property of the readout amplifier on the camera, and, unlike shot noise, it is

independent of signal. Based on the rest of the section, it's clear the authors know this and so this was probably just phrased in a confusing way, but it could be misleading to readers who aren't very familiar with the different types of noise in a digital image and where they come from.

R#2 is right. We rephrased this paragraph: "Read noise is an inherent property of camera sensors, regardless of whether or not light reaches the detector. Thus, the measurements are carried out without excitation light and without sample. When possible, the camera shutter was closed to avoid any unwanted light contributions."

"The detector exposure time was set to short values (e.g. 10 ms) to avoid Dark Signal Non Uniformity (DSNU) contribution" Provide a reference.

We added reference [T. Orzanowski and P.W. Wachulak "Investigation of CCD camera photo-response nonuniformity with a laser-plasma soft x-ray source", Proc. SPIE 10974, Laser Technology 2018: Progress and Applications of Lasers, 1097410].

(dark noise is considered equal to zero for this short acquisition time)" Provide a reference. What was the range of dark noise for the cameras used in this study?

We were not interested in dark noise within the framework of this study. We recommended using short exposure time (10ms) to assess read out noise and DSNU. At this exposure time dark noise is considered as negligible. We added the reference "Lambert et al., Assessing camera performance for quantitative microscopy, Methods in Cell Biology, 2014". To our knowledge, Dark Noise = $\sqrt{\text{Dark current} * \text{acquisition time}}$. For the cameras we tested, dark currents are given in the datasheets and are in the range of [0.06 - 0.55] electrons/pixel/sec.

Page 16

"After many tests, repositioning, and tilt adjustment of the camera, the symmetry was recovered (measurement of December 2019 and later on)." The reader will likely be as interested in how to correct for identified problems as they are in methods of revealing them. The authors give several anecdotal observations like this one, but little explanation as to why the method worked. These should either be left out or explained further. Eg, what does "repositioning" mean?

We made this phrase more precise: "We recovered the symmetry after taking out and reinstalling the camera, paying special attention to have no tilt between the camera, the camera adaptor and the microscope body (measurement of December 2019 and later on)."

"(<https://svi.nl/NyquistRate>)". Please provide peer-reviewed publications as references, rather than (or in addition to) manufacturers' websites.

We added the reference: Piston, D. W. Choosing Objective Lenses: The Importance of Numerical Aperture and Magnification in Digital Optical Microscopy. The Biological Bulletin 195, 1–4 (1998).

"The brightest beads should be used, and this is the reason we recommend using single labeled beads" I also use Invitrogen beads, and have found that single labeled fluospheres are brighter than multi-color beads. But is this universally true for all bead manufacturers'?

This is an overstatement, we did not test all manufacturer's beads, but for the main two bead manufacturers that we tested (Invitrogen and Spherotech) we came to the same conclusion.

"The beads diameter should be below the resolution limit." Or perhaps the resolution required for the experiment?

Thank you for this suggestion. Our intention here is to measure the performance of the lens itself. Accordingly, the user of the microscope can conclude and decide on the experiments that he can carry out.

Page 17

"We observed that if the SBR is low, then the precision on the FWHM calculation is low (Figure S1)". This is of course expected - please provide a reference.

Once again, thank you. We added reference Stelzer 1998. As suggested by another reviewer, we also added SNR influence on PSF information (figure S2).

"as the objectives and the optics of the whole system are set for the GFP channel" Were the microscopes used set up in this way? Or is this in reference to typical manufacturers' specifications? If the latter, provide a reference.

It is in reference to typical manufacturers' specifications. We changed the phrase as follows: "Most conventional microscopes and commercially available optics are designed for performing better at visible wavelengths" and gave the reference "Blinton et al., Optical Considerations at Ultraviolet Wavelengths in Confocal Microscopy, Springer 1995".

"Attention was paid to the SBR in LSCM images." Why state this specifically for LSCM? Isn't the same true for all modalities?

This sentence was removed.

"due to spherical aberrations" How was this determined? Provide a reference.

We added two references for a theoretical explanation of spherical aberrations "Hell et al. Aberrations in confocal fluorescence microscopy induced by mismatches in refractive index, J. of Microscopy, 1993" and an experimental approach and observations of "Cole et al. Measuring and interpreting point spread functions to determine confocal microscope resolution and ensure quality control. Nat Protoc 6, 1929–1941 (2011)."

"Figure 2c shows the Lateral Asymmetry Ratio (LAR), defined as the ratio between the smallest and highest of both x and y experimental FWHM." Of an individual bead? Or an averaged population?

This is rephrased into: "We studied the symmetry of the PSF by calculating the mean values of the Lateral Asymmetry Ratio (LAR), defined as the ratio between the minimum and maximum x and y experimental FWHM of individual beads (Fig. 3D)."

Page 18

"The thin layer of a fluorescence dye gives a more precise illumination pattern and can be more convenient for shading corrections for tile scan acquisitions" Why? Provide a reference.

For the thin layer the collected out of focus light is much lower than for a plastic slide. We added the reference "Model, M. A. & Burkhardt, J. K. A standard for calibration and shading correction of a fluorescence microscope. Cytometry 44, 309–316 (2001)."

"For spinning disk, the pinhole crosstalk" Provide a reference

We added the reference "Toomre D, Pawley JB (2006) Disk-scanning confocal microscopy. Handbook of Biological Confocal Microscopy, ed Pawley JB (Springer, NY), pp 221–238."

"We performed the flat illumination measurements for 130 objective lenses for a wide range of magnifications (5x-100x)". On how many microscopes, using how many different types of light sources? I think it equally (if not more) informative to parse this data out by type of light source, as I would guess they would have the largest effect.

We added this information at the Results part ("35 microscopes") and we studied the dependence of the U and C metrics values according to the source type (Figure 4C).

"Wide-field setups show a high dispersion of C and U values. Finally, the analyzed spinning disk setups had less dispersed values closer to the ideal 100%." Why? Is this due to the type of light source? Or the imaging modality? Were different types of objective lenses (eg, correction for aberration) used?

As we explain in the Discussion part, we believe that this is due to the different illumination mode. For WF case the auto-alignment of the bulb sources was not always accurate compared to SDCM where the laser alignment is accurate and more constant in time. "Results also showed that lamp sources result in the least favorable U-C values, most often due to less accurate auto-alignment procedure. Laser sources are more accurately aligned (high C values) and present a better Gaussian profile, even though for SDCM and WF techniques their coupling to large area detectors results often in low U-C values."

In addition, most of the studied WF microscopes were equipped with sCMOS cameras (i.e., worse U values), whereas for SDCM 60% of the microscopes were CSU X1 models and 40% W1 models. For CSU X1 the smaller FOV contributes to giving better U values.

We indeed mention in the Results part: "To note that 60% of the SDCM microscopes were CSU X1 model and 40% W1 model. The former shows better U-C values, due to the smaller FOV"

And in Figure S6: "We indeed observed that low U/high C accuracy combinations were mostly associated with larger sensor sizes (magenta circle in A)."

The type of objective lenses used were similar for the WF and SDCM cases.

"Most systems designs involve the 405 laser line passing through a different fiber/lightpath than "visible" laser lines. The alignment is different and explains these differences." Provide a reference.

We added reference "James Pawley, Appendix 2: Light paths of current commercial confocal light microscopes for biology, Handbook of biological confocal microscopy, 2nd edition, 1995, Plenum, NY".

Page 19

"Our co-registration study collected data from setups of the three microscopy techniques using over 70 different lenses" Many other components can affect co-registration. How many different filters, dichroics, multi camera systems were included? Were there any trends associated with specific components?

In the new version of the manuscript, we mention different specific components in the results section (filter cube for instance) and we give possible suggestions for the identification of the co-registration issue, in the Discussion section. We added some measurements and now we are at 84 tested objectives for the co-registration.

"We believe it is harder to achieve a high SBR, as needed to further correctly identify beads and estimate their center coordinates, using these tiny beads." This is surprising, given how commonly diffraction-limited beads are used to measure and correct for registration shifts. This data should be included.

We added Figure S7 that shows the influence of the beads size to the co-registration accuracy calculated by our plugin. It is important to mention that our principal aim here is not to correct registration shifts but to characterize them (a similar response is given to R#1 comment 28).

"while the combinations with the Blue are more often associated with poor co-registration" Is this consistent with the manufacturers tolerance for the lenses used?

Yes it is. We added the reference "A.C. Bliton and J.D. Lechleiter, "Optical considerations at ultraviolet wavelengths in confocal microscopy, Handbook of biological confocal microscopy, second edition".

"However, spinning-disk confocal systems are more often associated with insufficient chromatic correction with 63% of the measured ratios below 1, compared to LSCM or wide-field workstations with 70% and 84%, respectively." Is this difference based solely on imaging modality? Are there any other components common between the spinning disks but not the others?

Indeed, microlenses add chromatic aberrations.

We thus added reference "T. Azuma and T. Kei, Super-resolution spinning-disk confocal microscopy using optical photon reassignment, 2015"

If I'm not mistaken, Neofluar refers to fluorite, not achromat, lenses.

You are right. We modified the text and we do not mention 'achromat' lens.

"63x 1.4 NA Leica Plan-APO objective" & "Hence, this objective is not recommended, and better-corrected lenses (lambda blue for Leica with better correction for the near UV light)..." I would be more conservative about recommending against the use of a particular make & model of lens without more rigorous study.

This is perfectly correct. We modified the text, now the comments are in the Discussion part and do not target the Leica company. More specifically we now indicate: "co-registration shifts of a microscope should be understood by the user, who should be aware of the type of the objective correction type and execute experiments accordingly."

Page 20

"the filter set cube was not correctly aligned" What does this mean and how was it identified?

We gave more precisions, now the phrase is as follows: "After testing the same objective on a different microscope and testing different equivalent filter cubes on that setup, we found out that one source of error came from the alignment of the filter in the cube."

"A similar cube equivalent case was associated with 0.19 and 0.23 B-G and G-R ratios, confirming that the issue came from the misalignment of the cube components (dichroic mirror and filters) and was not related to some type of poor objective correction." Isn't this most often due to a wedge in a dichroic or emission filter, rather than an alignment issue?

We considered here that a wedge in a dichroic or emission filter is a misalignment issue.

"far from the theory". I think you mean far from the ideal.

We have corrected that.

"upstream/downstream of the light path" Please give examples of upstream and downstream components that can cause chromatic shifts.

In the new version of the manuscript we deleted this phrase. We give concrete examples at the coregistration section, like the microlenses issue, the cube filters or the dichroics.

"The G-FR couple is slightly worse than the equivalent G-R that is expected as the wavelength difference is higher" Expected based on theory or manufacturers tolerance? Please reference.

Based on theory. We added the reference "A.C. Bliton and J.D. Lechleiter, "Optical considerations at ultraviolet wavelengths in confocal microscopy, Handbook of biological confocal microscopy, second edition".

Illumination power stability section - reference Swedlow et al 2002

Thank you, we did it.

Page 21

"Some other lasers have lower STAB (<97%) and higher standard deviation (0.2), such as the Argon 488nm laser line of Figure 8b." I believe this should be Figure 5b.

Thank you, we corrected it.

"As highlighted in Figure 5e". I believe this should be Figure 5f.

Thank you, we corrected it.

"Figure 5d shows that there is a correlation between the power and temperature fluctuations" should be changed to "that, in some cases, there is a correlation"

The phrase is now modified as you suggest.

Page 22

"i) Stage drift". This section covers XY & Z drift. Z drift can result from sources unrelated to the stage.

In the discussion section, we added : "A z drift can also arise due to a bad tightening of its drive axis, which is possible on aging microscopes."

"but brightfield observation of various features is possible as well" What type of brightfield features work? Do they need to be diffraction limited?

Here we were thinking about dust elements. From our experience and the positioning repeatability calculated values, they don't need to be diffraction limited.

Page 23

"When the velocity before and after stabilization is identical and really small (lower than 15 nm per minute, which is the theoretical resolution of a high NA objective lens after 10+ min), the drift can be considered negligible (high precision experiments, e.g. super-resolution purposes)." This amount of drift may be problematic for localization microscopy, which can take hours to acquire.

In the Results part, we now state: "When V_x and V_y values are under 15nm/min and close to each other, the drift is negligible (high precision experiments, e.g. super-resolution purposes). Accumulated over time, this amount of drift may be problematic for localization microscopy (acquisition time of minutes or hours)."

"Any other drift parameter combination should be rejected or taken into account during the image analysis step because it can lead to serious misinterpretation." Again, this is not true for every type of experiment.

We changed this sentence to: "Any other drift parameter combination should be earnestly examined because it can lead to failed experiments, especially when the focus is lost."

"When x-y linear encoders are mounted on the x-y plates of the stage, then the repeatability is further improved." Provide a reference.

We added reference: "Thorn, K., & Stuurman, N. (2014). Digital Microscopy. In *Handbook of Digital Imaging*. Location: Wiley. UCSF. Retrieved from <https://escholarship.org/uc/item/7s21820j>"

"below the usual maximal 1 um manufacturer's specifications". This depends on the make & model, and would likely be better for linear encoded stages.

This is right. To be more specific, we added values: "This value is within the specifications of the manufacturers, which are usually inferior to 1 μm for stepper motors and 0.7 μm for linear encoders."

"It is not easy to define limit values for repositioning, as it highly depends on the stage type. For instance, for the two ASI stages having linear encoders, although they have the same

characteristics and were mounted on the same microscope, they present different repositioning values." Should this be "individual stage", given the next sentence?

This is right. We restructured the paragraph to avoid confusion.

Page 24

"Other parameters can influence the repeatability here, like the proper fixing of the sample, the waiting time (if it exists) for each position, the acceleration of the stage and the temperature variations." These are indeed important parameters. Were any of them kept constant in this study, in some or all of the data collection?

We agree with the reviewer that these parameters are indeed important and we thus present now some supplementary data (Figure S12) that study the influence of the stage speed and acceleration on the repeatability. We give some comments in the Results section of this chapter. It was difficult, almost impossible, to keep some of these parameters constant in that study for the 22 different stages that we studied, due to the stage and acquisition software variability. We tried to work in similar ranges when this was possible, eg. room temperature ranging 20-23°C or waiting times of 10-300ms.

"for 100 frames" Why was this number chosen? It seems like more than enough for CCD, but insufficient for CMOS.

The choice of this value is now better explained: " In our protocol, the read noise is measured by the fluctuation of the value of a pixel along a temporal acquisition of 100 images, without any light influence. Read noise is then considered as a temporal noise. While the quantity of 100 frames may seem over-estimated for CCD read noise measurement, it is a fair compromise for sCMOS read noise (obviously not to characterize every pixels' attributes⁸⁵), and DSNU measurements, as explained below." [see DSNU section].

The difference in read noise between CCD and CMOS is particularly important here. This should be explained in more detail and references provided.

As mentioned above, we now think that we emphasize the differences better and give more literature.

Page 25

"The observed wide-field higher dispersion of the measured/theoretical axial resolution ratio, as compared to both confocal types, could be associated with some spherical aberration issues" This is only one possible explanation. Other aberrations and vibration are other common sources.

After adding some PSF measurements of some other objectives in the paper and looking closer at the results (new Figure 3C), we conclude that the observed dispersion for the WF case is not higher than the dispersion for the two other techniques, so we did erase the above phrase. The modality that gives the highest axial FWHM dispersion is the LSCM, and we gave the explanations in the main text (spherical aberrations, water objectives, pinhole, 405nm alignment, etc).

"In our case we use 0.175 nm beads that give slightly larger PSF values compared to point source PSF (one should convolve the two PSFs for the theoretical values)." This should be 175 nm.

We did that correction.

Page 26

"considering optimal conditions such as shot-noise free confocal imaging" should be "considering optimal, though unobtainable, conditions such as shot-noise free confocal imaging"

We did that correction.

"However, we propose, as defined using experimental data from Figure 3a distributions, that Centering above 40% and Uniformity above 50% can be considered acceptable" Again, this is very dependent on the experiment.

This conclusion is now added to the discussion : "If the tolerance values are not met, an after-sales service visit is not necessarily essential. One should keep in mind that depending on the application, low values can have serious (quantification of the intensity of objects located over the entire field of view) or manageable consequences (segmentation, tracking)."

Page 27

"This clearly shows that, when observed, shifts are likely to be associated with imaging of near UV excited dyes." Were violet corrected lenses included in the study, and if so did they perform better?

Most of the lenses were Plan Apo lenses, apart from some cases that are noted as Plan-Fluor or Neofluar lenses. However, even for the Plan Apo lenses, the UV corrections are not perfect, it is really lens "generation" and constructor dependent.

"while going through the spinning-disk unit and the microlens array of the disks" Please explain or reference how the microlenses could cause chromatic shifts.

We added reference "T. Azuma and T. Kei, Super-resolution spinning-disk confocal microscopy using optical photon reassignment, 2015".

"showing that the microscopes with all their components are most adapted for imaging at that range of the visible spectrum" Should be "the microscopes we tested"

Thank you for that correction.

"gives hints on how to correct data to achieve some qualitative, rough comparison" I'm not sure what is meant here, but it sounds dangerous. Image corrections should be made with the same rigor as any other data manipulation.

We removed this phrase.

Page 29

"we considered this value as our experimental tolerance value" Please explain the rationale behind choosing this value.

As mentioned several times in this new version, mostly experimental limit values are defined. That was particularly tricky for noise measurements since results we obtained were extremely reliable.

"Warm pixels (i.e., pixels showing significantly higher signal than the average pixels) are more frequent with sCMOS cameras" Provide a reference.

We actually simplified this part and removed this sentence.

"Here we define a hot pixel as a pixel that does not respond to light." This may be confusing, as 'hot pixel' (in my experience) is used to describe a consistently bright pixel.

We now defined hot and cold pixels in that way : "When available, we also acquired data with blemish corrections to replace cold and hot pixels (defective pixels for which incoming light has no influence on, with constant zero value and saturated value of the sensor dynamic, respectively) with the average intensity of surrounding pixels⁹⁰."

Figures

1b. This data is for one objective only. Was similar data collected for other objectives?

Yes, similar data was collected for some other objectives. We show here only one example, as we think it is representative.

1c. Was the acquisition adjusted to maintain SNR at different pixel size?

Yes, we always tried to keep a similar SNR.

Fig 2. While the experimental/theoretical ratio should help normalize for differences in lens NA, it would be interesting to know how correction for aberration affects these data. In the Results section we give some examples for spherical and coma aberrations. We also added a phrase for the lenses not corrected in the UV range "It should be mentioned that we avoided measuring PSF for wavelength ranges with objectives that are not meant to be corrected. More specifically, this is the case for Plan Fluor or NeoFluar objectives for the DAPI channel. For instance, a 40x, 1.3 NA NeoFluar objective gave a ratio of 3.3 and 2.1 for xy and z respectively. These values were not added in Figure 3B, as we considered that these objectives are not meant to give acceptable performances in that wavelength ranges".

Fig 3. It would be interesting to see how the type of light source (including how the light source is delivered into the light path) affects these data. I would also like to see how the sensor size and FOV affect uniformity.

We added Figure 4C showing the U-C values according to the light source type. We show some examples of limit values for the sensor size and FOV in Figure 4A, illustrating how these parameters affect uniformity.

Fig 4. Some lenses are labeled Plan-Fluar. What are the others?

PLAN-APO

Fig 6. This data does not differentiate between stabilization of stage drift and stabilization of the sample on the stage. This should be made clear.

We consider that once the sample is placed on the stage, then it becomes part of it, so the stabilization time is common. A phrase at the Discussion part makes that clear: "Our results showed that a stabilization time is mandatory before starting the acquisition. Furthermore, it is very important to consider that once the sample is fixed on the stage, it becomes an integral part of this device."

Fig 8b. Is blemish correction a feature of the camera, acquisition software or performed post-acquisition?

It is a feature of the camera that can be activated or not through the software. It is by default possible to do it when using the camera constructor's software, but when using another software sometimes users do not have the choice (sometimes the correction is not proposed and sometimes the option of not using that correction is not proposed). It is activated before the acquisition.

Fig 8c legend. For clarity, use 'intensity value' rather than digital number, since intensity value is used elsewhere in the manuscript.

Figure 8c is now Figure S14c. We did as you suggested.

Fig 9. I had a hard time interpreting this figure. Please provide a more detailed explanation of the purpose/message.

We now provide a more detailed explanation in the legend of the figure and at the beginning of the Conclusion part, that we believe helps in better interpreting that figure (Figure 10 now).

Fig S1. Signal-to-noise ratio as well as SBR would be useful in interpreting these data.

We now provide both SNR and the SBR of the same data (Figure S2a and Figure S2b).

Fig S2. An explanation and reference should be provided for what appears to be a disk speed/camera sync artifact in these images.

We added the phrases : "Artifacts shown on these images come from a synchronization artifact between Nipkow disk rotation and camera acquisition. The image projected on the sensor chip results from the integration of multiple individual scans by the Nipkow disk. A short exposure time can result in these artifacts". We added the reference: "Chong et al., Optimization of spinning disk confocal microscopy: synchronization with the ultra-sensitive EMCCD, Proceedings Biomedical optics 2004"

Fig S3. Again, what types of light sources are represented here?

We show this data in Figure 4C now and we included the type of the light sources as well as the type of the detector Figure S6.

Minor comments:

Page 2, last paragraph: change 'die' to 'due'

Page 3, para 3: change 'additionally' to 'additional'

Thanks.

Throughout the manuscript the authors refer to point scanning confocals as "confocal" and spinning disk confocals as "spinning disk". Since spinning disks are confocals too, perhaps stick to acronyms, such as LSCM and SDCM.

Thank you for this suggestion. We used the proposed acronyms in the revised version.

Reviewer #3 (Comments to the Authors (Required)):

This manuscript, "Long-term quality assessment in light microscopy through accessible tools and reliable metrics", details six areas for quality control (QC) measurements of light microscope imaging systems, provides protocols for making the necessary measurements for each and gives specific values for "acceptable" performance in each QC area. This work builds on that of the Confocal ISO from 2019 and QUAREP-LiMi project. Having clearly defined equipment standards is important for reproducibility and robustness in light microscopy experiments.

Large scale feedback

This topic is very important and the authors have clearly put a lot of careful thought and diligent work into this manuscript. The amount of data and the range of sources is amazing. The information provided in this manuscript is a great framework around which to organize the evaluation and monitoring of instrument capability, which is in turn a great foundation for reproducible and robust microscopy experiments. I very much appreciate the summary table at the end and the attempt at pulling all the information together with the heuristic map.

However, I had a hard time getting through the manuscript for two main reasons: The overall organization and that it has a large number of words (two challenges I am usually guilty of).

Having the intro, methods, results and discussion for each QC item separated into those headers made it really hard for me to keep track of the important details for each QC item. I had to keep flipping back and forth to keep continuity and answer questions and still felt uneasy about my understanding. I also feel there is a lot of duplication because of this physical separation of the same topic. Perhaps the manuscript would be easier to read with an organization more like this (though doesn't have to be this exactly):

-Short aggregate intro (already there).

- Maybe include info on Shannon-Nyquist here since it comes up in a lot of places? If not, have an explanation and/or references of what it is in the first mention

-Headers = QC items (Lateral and Axial Resolution, Field Illumination, etc)

- Sub-headers = Intro/Methods/Results/Discussion

- Short aggregate discussion pulling together evidence for impact of QC that are presented in each results section.

- Conclusion (already there)

Thanks for this proposition. We modified the structure of the manuscript towards that direction.

For the dense text, there are a few broad things that I think will shorten it and/or make it read more easily. First is the reorganization outlined above--I suspect replicated information will get deleted. Two others are evaluation of passive voice versus active voice (sometimes one is shorter than the other; instead of "We conclude X is...", does "X was..." work?) and significantly reducing (but not eliminating!) transitional phrases (such as "of note," etc). Lastly, using fewer qualifying words (see next item) can help streamline arguments.

I found several instances where qualifying words (such as "enough", "negligible", "often", etc.) were used with no follow up on what they relate to quantitatively (just like here where I don't give a number for "several"). Two examples of where specification is missing are detailed here, but keep an eye out for it throughout the manuscript:

- Page 16 very top when talking about setting up the PFS metrics. You state "robust enough" and "kept negligible" then say "1 to 5%" and "4 to 6%" is small. Can you address the question: Why are those good values? This might be something to address more broadly in the introduction.

Thank you for this relevant remark which allows us to improve the manuscript quality. We modified the main text, and more specifically in the PSF section we replaced the above text with the following:

"We first investigated if the proposed procedure was robust enough, by examining: i) the image processing: for one z-stack of a single bead five repetitions of the image processing with our plugin gave strictly identical results, ii) the image acquisition: for five acquisitions (five different time points) of the same bead we studied the variability of the measured FWHM for a single bead (Fig. S1). We found for the WF case $x=0.256 \pm 0.002 \mu\text{m}$; $y=0.26 \pm 0.002 \mu\text{m}$; $z=0.673 \pm 0.02 \mu\text{m}$, iii) the field of view containing several beads (Fig. S1B). For the beads that are in the central area of the image (the central area is the area that we defined in our protocol that contains the beads to be taken into account for the FWHM calculations) we found $x=0.261 \pm 0.004 \mu\text{m}$; $y=0.266 \pm 0.003 \mu\text{m}$; $z=0.685 \pm 0.02 \mu\text{m}$. For the entire field of view we found $x=0.267 \pm 0.016 \mu\text{m}$; $y=0.274 \pm 0.02 \mu\text{m}$; $z=0.676 \pm 0.032 \mu\text{m}$. From the above results we conclude that the image processing is repeatable and robust. The repeatability accuracy of the different acquisitions showed a variability of less than 1% in xy and less than 3% in the z direction. When processing several beads from the central part of the field of view we found a variability of less than 2% in xy and less than 3% in the z axis. For the whole field of view the variability increased to 8% in xy and 5% in z axis. These results show that first a small variability exists among the acquisitions and it is higher for the z-axis but it stays low. When the whole field of view was taken into account, spatial aberrations that are highly linked to the objective type and quality, caused a higher variability."

- Page 18, middle. Results section about figure 3. "Centering value was often close to 100% using LSM". Estimating from the graph, it isn't clear that the distribution along Centering is different between the modalities. It would be helpful if "often" had a number.

With the aim of quantifying more in detail our results and using less frequently the word "often" we performed a statistical analysis of the U-C distribution and we present the results in Figure 4B.

We realize that the LSCM modality shows the highest dispersion for both U- metric values. The Centering metric was significantly higher for WF and SDCM techniques with a lower dispersion than for LSCM. We give possible explanations in the Discussion section. We did an analysis of U-C depending on the fluorescence channel (DAPI channel gives less good results). The U-C distribution with the detectors is now a supplementary figure (Figure S6). We did a statistical analysis of the U-C distribution with the excitation source (Figure 4C).

It is not always clear to me what criteria are used to define the "acceptable" values. For several of the QC items it seems like "what most of our measurements fit into". While it is not bad for that to be a component of the definition, having discussion and/or guidance about the quantitative impact of being within or outside of the acceptable range would be helpful.

Thank you, you're absolutely right. The other reviewers also point this out. All along the revised manuscript, we tried to achieve clarity in the justification of the limit values (which are based on experimental data) and give their effect on biological applications.

Medium scale feedback

- Illustrations of the parameters being measured in each QC test would be really helpful for readers to understand what is being measured and compared. I can envision this either all in one figure or as a first panel in the separate figures. This might be more complicated than it's worth for some measurements, but try for as many as possible.

Thank you for that comment. We added Figure 1, which gives a schematic overview of the seven proposed QC guideline fields.

-The methods for sample preparation need more detail on implementation. Specifically, it would be helpful for the bead sample preparation to include dilution factor or final percent solids in the final dilution used to achieve the bead density specified. This will help readers have a place to start and maximize likelihood of success.

In the revised version we took into account your comment and added more details on the sample preparation, and specifically for the bead slide preparation. For example, for the PSF slide preparation we now indicate: "We used 175 nm diameter blue and green microspheres for the PSF measurements, for the DAPI and GFP channel respectively (beads PS-Speck, #7220 Thermo Fisher Scientific). Beads were vortexed to avoid aggregates and diluted in distilled water to achieve a density of 10^6 beads/ml.

- For the Field Illumination QC, it is unclear to me why a high uniformity but low concentration (upper left of the graph) is considered less optimal than moderate uniformity and moderate concentration (lower left corner of the green square), exp after looking at the example images in Figure S3. Very low intensity differences will give more robust quantification than highly centered but more variable illumination. Discussion of the impact on quantification, perhaps with an example, would be helpful.

Indeed, you're right. We modified the phrase in the Discussion part as follows "For biological imaging we defined the experimental limit values of $U > 50\%$ and $C > 20\%$. If the tolerance values are not met, an after-sales service visit is not necessarily essential. One should keep in mind that depending on the application, low values can have serious (quantification of the intensity of objects located over the entire field of view) or manageable consequences (segmentation, tracking)". We also changed the limit values, based on their influence to biology (added Figure 4D-F).

- Page 16, 2nd paragraph regarding the system where one PFS axis changed after a service visit. State only the information about the data ("10% decrease in X-axis FWHM but no change for y-axis FWHM after an external service provider revision"). Save the recommendation about checking before and after service work for the discussion since that applies to multiple measurements.

We did as you recommend in the revised version (Discussion part: "We recommend acquiring PSFs and calculating the associated metrics before and after any scheduled revision of a system, to compare QC data").

- Page 17, lower half of the page, shorten the sentence about difficulty in getting Nyquist with WF and SDC. "...meeting Shannon-Nyquist criterion is limited by the fixed physical pixel size of the detector array."

In the revised version, this sentence no longer exists. In the Results part we explain as follows: " The WF and SDCM FWHM measurements show similar behavior one to the other. The measured FWHM values are not as good along the x-y axis than the z-axis, compared to the theoretical ones. Of note, as the sampling criterion was met in z, the observed ratios for the axial FWHM are closer to the theoretical value". And in the Discussion part: "Variability was associated with suboptimal sampling rate (e.g., inevitable for WF or SDCM for the lateral values for low magnification objectives and big pixel size cameras)"

- Figure S2

- panel C. Based on the images in this panel, the Stabilo looks like it has substantially worse Uniformity than the Chroma slide. Based on panel A, I'm assuming the chroma slide and rhodamine film would be similar. The text says the Stabilo is sufficiently equivalent to the Chroma slide, but does it pass your own Uniformity standards?

The authors decided for simplicity reasons and after some comments of other reviewers to remove the Stabilo part. For the Chroma and Rhodamine slide results, we give the Uniformity results in the table of Figure S4B, that show that Uniformity is slightly better for Rhodamine slide.

- In the figure legend, Stabilo autocorrected to "stabile" a couple times. I did not check the main text to see if it happened there, too.

Same answer as for previous comment here, Stabilo part is removed.

- Figure S3. The gradient values for the example images are not legible. Do they all use the same gradient LUT so a color scale key could be used? If possible, remove the text from the images-it is too small to read and obscures the pattern a bit. It is unclear where the Center is.

Thank you for mentioning it. We restructured this figure which is now Figure 4 in the main text. Even if these gradient images are bigger, it is still not possible to read the values (the values and images come from our ImageJ MetroJ_QC plugin). Colors are predefined in function of the % value : hot ones for high signal, cold ones for low signal. For clarity reasons we now give the gradient LUT in Figure S5.

- The Heuristic map is not a familiar schematic for me and was not immediately intuitive (and sadly Google did not easily provide a definition to help). 1-2 sentences for the figure legend about its goal/purpose and/or how to read it would be helpful.

We indeed added more explanations in the legend and in the main text. We believe this kind of representation is useful to figure out the magnitude of the task.

- Table 2. Would it get too bust to add a column of recommend measurement interval? Or maybe add a separate table for that?

You are absolutely right. We added in the Intro the sentence : « We established QC guidelines and proposed tolerance values and recommended monitoring frequencies that provide valuable indicators of the setup "health" state and experimental reproducibility (Table 1). The recommended frequency depends on multiple parameters like the per year occupation time of the microscope, the number of its different users (major difference if the system is in a core facility with dozens of users per year or in a research team, used by the same people the whole year), the human resources that take care of the microscope, and the kind of biological experiments carried out .»

Previous Table 2 now became Table 1.

We also added the recommended measurement frequency in Table 1 for each Monitoring QC field. These values are discussed in every parameter section now. Thank you for this suggestion.

Odds and Ends

- Mention the summary table in the introduction
- Describe the goal of Shannon-Nyquist sampling at the first mention. Not everyone who will read this will be familiar.

We explain the Shannon-Nyquist criterion in the revised version the first time it the term appears and give a reference (Shannon et al., Communication in the Presence of Noise, Proceeding of the IRE, 1949) , as follows: "The criterion requires a sampling interval equal to twice the highest spatial frequency of the specimen to accurately preserve the spatial resolution in the resulting digital image".

- "automatize" sounds unusual to my US "ear"; I suspect you are using it where I would use "automate". I leave it to the authors to determine their preference.

In the revised version we use the US version and replaced "automatize" by "automate".

- Figure 8, panel c images. Is the color scale the same for both? If so, only need one. If not, please make them the same.

This panel is now in Figure S14c. Yes, the color scale is the same. We prefer keeping both images to show the different patterns on the two sensors.

- Figure 8, panel f. Should there be scale bars here?

We do not think so. We represent the stability metric over time for different cameras.

- To help link this with other great articles about robust quantification, instrument characterization and experimental reproducibility, it might be helpful to readers to cite the following papers in the intro or discussion. I leave it up to the authors to determine if this is within the scope of their manuscript.

- John M. Heddleston, Jesse S. Aaron, Satya Khuon, Teng-Leong Chew; A guide to accurate reporting in digital image acquisition - can anyone replicate your microscopy data?. J Cell Sci 15 March 2021; 134 (6): jcs254144. doi: <https://doi.org/10.1242/jcs.254144>

- Montero Llopis P, Senft RA, Ross-Elliott TJ, Stephansky R, Keeley DP, Koshar P, Marqués G, Gao YS, Carlson BR, Pengo T, Sanders MA, Cameron LA, Itano MS. Best practices and tools for reporting reproducible fluorescence microscopy methods. Nat Methods. 2021 Jun 7. doi: 10.1038/s41592-021-01156-w. Epub ahead of print. PMID: 34099930.

Thank you for this proposition. We added these two very recent references. We are also very happy to mention the excellent work of Paula Montero Llopis.

This manuscript covers an important topic for biological research, that of measuring the light microscope system with which imaging data is collected and evaluating the performance thereof. It represents an immense effort to collect empirical data and evaluate appropriate criteria. I believe the indicated adjustment will significantly improve impact and usability. Great work and I hope to see a revised version!

We truly hope that this revised version meets your expectations.

--Wendy Salmon

Whitehead Institute (until Sept 2)

University of North Carolina at Chapel Hill School of Medicine starting Sept 13

May 17, 2022

Re: JCB manuscript #202107093R

Dr. Orestis Faklaris
CNRS
Biocampus
1919 route de Mende
Montpellier 34293
France

Dear Dr. Faklaris,

Thank you for submitting your revised manuscript entitled "Quality assessment in light microscopy for routine use through simple tools and robust metrics". The manuscript has been seen again by the original reviewers whose full comments are appended below. While the reviewers all feel that the manuscript has made significant progress, important issues remain to be addressed.

As you will see, reviewer #3 has only one minor comment and now recommended publication. However, reviewers #1 and 2 continue to feel that the clarity of the manuscript needs to be improved, particularly with respect to the idea of making these sometimes complicated and technical guidelines more accessible to the general JCB reader. One possible approach here is to ask one or more colleagues who utilize these microscopy techniques but are not themselves "experts" to take a thorough look through the manuscript and highlight the areas that they find confusing or insufficiently clear. Reviewers #1 and 2 have also highlighted some statements that they feel are either unclear or inaccurate and have indicated some areas that have not been sufficiently referenced - we hope that you will be able to address these concerns as well.

As you will also see, reviewer #1 feels that the raw data should be made available to the reader and we would agree that this is a good idea, if practical. We would suggest that perhaps you deposit such raw data in a general repository such as Figshare or Zenodo (or similar) which will provide you a DOI which you can then cite/link to in the manuscript. Finally, reviewer #1 also suggests that you add some statements to the Discussion wherein you explicitly state (and discuss) the limits of these guidelines, particularly in the area of comparability of performance.

As you may know, our general policy is that papers are considered through only one revision cycle; however, given that the suggested changes are relatively minor we are open to one additional short round of revision.

Please aim to submit the final revision within one to two months (though, if you need more time than this, please let us know and we can work something out), along with a cover letter that includes a point-by-point response to the remaining reviewer comments.

Thank you for this interesting contribution to Journal of Cell Biology. You can contact me or the scientific editor listed below at the journal office with any questions, cellbio@rockefeller.edu.

Sincerely,

Joerg Bewersdorf, PhD
Monitoring Editor
Journal of Cell Biology

Tim Spencer, PhD
Executive Editor
Journal of Cell Biology

Reviewer #1 (Comments to the Authors (Required)):

This is a re-review of this manuscript and everything positive stated about this work still remains. The revision done was substantial and the manuscript is very much improved. The responses to the reviewers comments (and I acknowledge that there were a lot of comments) were reasonable and the changes made sufficient, but for two points: 1) access to the raw data and 2) comparability of performances.

Regarding 1) access to raw data: The limits of hosting the amount of data is acknowledged, but given that QC data should be

metadata to research data and are crucial for a correct analysis this manuscript will be setting the tone for the community and it is important to create open access and transparency. For this reason I urge the authors to reconsider their position and curate their data to make the accessible. The likely strongest argument against making data open from the authors is that they feel they open "Pandora's box" as there will be limits in direct comparability between microscopes. The authors argue in their response that these technical limits made them focus on providing QC metrics over comparing performance. While that is a valid point, it is not one that should stand in the way of open data access compatible with FAIR norms. Especially since the authors are contradicting themselves on the issue, see point 2).

Regarding 2) comparability of performance: As laid out in 1) the authors have some concerns about comparative performances, but in the camera section change their position and explicitly state the comparability improvement by using photons over grey values. A short section at the end of the manuscript in the summary discussion would be helpful to outline the limits within the data sets as acquired and add recommendations on using QC data acquired with the presented protocols for microscope (performance) comparison.

Besides that there are still issues with clarity of the text. While overall there is a massive improvement of the text there are still significant challenges for a non-expert reader. To give some examples:

- Abstract page 1: "Reproducibility in light microscopy represents a goal to achieve to improve research results" likely means "Increasing rigor and reproducibility in light microscopy aims to improve research results..."
- Out of several examples from the introduction, take a look at the second paragraph, the redundancy and lack of clarity. The argument is good, but it needs to be written in a straight forward and clear manner.
- Or page 3&4: "the microscope objective will work on its manufactured design"
- The "Shannon-Nyquist criterion" is evoked, but no reasoning is given to the why it applies.
- On page 9: "..., considering the intrinsic Poisson noise ..." From the flow of the argument in that section it seems that similar pixel values are to be expected for obvious reasons, the mentioning of the Poisson distribution for light detection would need to be explained. For the reader it is important that the authors say what they mean exactly and not add half-sentences that imply the need for more details without providing them.
- Page 10: "Previous studies investigated the field illumination ..." This is a lead argument for the beginning of the section! State what has been done and why you chose a distinct subset of metrics!

Examples like this can be found throughout the text and are truly editorial in nature, but important for the audience of readers.

Within the 7 subsections I would suggest to replace "Introduction" with "Background". There is a good introduction to the whole manuscript already.

For power measurement of light sources the authors might want to mention the recently released protocol from QuaRep working group 1. It adds to this work.

The literature overview provided by the authors is impressive and by itself a helpful resource for the community. However, the foundational work by Ted Young, Lucas van Vlies and Raimund Ober on camera calibration is missing, as is later work by Fang Huang, Bewersdorf and Lidke as well as the work by Rieger and Stallinga and work by the Ober lab on sampling requirements is missing and should be added.

Reviewer #2 (Comments to the Authors (Required)):

In the revised version, the authors provided a major reorganization of the manuscript and added additional data. Unfortunately, it is my opinion that the work remains inappropriate for publication in JCB for several reasons:

The manuscript assumes an understanding of microscopy that, in my experience, is beyond the majority of cell biologists. The manuscript contains inaccurate statements and poor references. The writing is generally hard to read and the manuscript is very long with many figures.

The effort to characterize so many microscopes is impressive, but I'm not convinced that the data is useful to a broad audience, and the measurements and methods they suggest are not novel. They do use a novel ImageJ plugin to analyze the data, but introduction and validation of the tool is not the focus of the manuscript.

While I appreciate the extensive work the authors put into the revision, I do not think they have adequately addressed the issues above. I recommend that the authors send this work to a specialized journal, such as Journal of Microscopy. To effectively reach a more general audience, the authors would need to dramatically increase their effort to define, explain and reference the background knowledge that is necessary to appreciate and implement their suggestions.

I also remain unconvinced that the suggested QC time intervals and acceptable limits are applicable to the wide range of imaging experiments performed by cell biologists. I think this audience would benefit more from a thorough treatment of the problems that may be lurking in their microscope and how they should be addressed, in addition to how to test for them. As in the previous version, the authors offer time intervals and acceptable limits that are based on the data they collected from their

microscopes and their general knowledge as core facility staff, but the rationale for the time intervals and acceptable limits is not discussed. They do say that "The recommended frequency depends on multiple parameters like the per year occupation time of the microscope, the number of its different users (major difference if the system is in a core facility with dozens of users per year or in a research team, used by the same people the whole year), the human resources that take care of the microscope, and the kind of biological experiments carried out." In my opinion, this statement argues against attempts to define general time intervals and acceptable limits for QC. When some rationale is given, it is most often inadequate (e.g., "From the experimental results shown in Figure 3B, we propose to set the tolerance value at a ratio of 1.5 between measured and theoretical FWHM. 69% of the lateral FWHM and 82% of the axial measurements are located inside the 1.5 ratio green square.").

I also think that it is critical to this type of work to not only provide a method of assessing a problem, but also - whenever possible - solutions. The authors list troubleshooting suggestions in Table 1, but many of them will not be helpful to novice microscopists (e.g., "beam size at the back focal plane of the objective" and "cosmic rays") and there are no references provided. They also make statements such as "The user of the microscope can nevertheless continue monitoring the PSF quality of the objective while keeping in mind that imaging of small structures or applying super-resolution techniques should be avoided (Fig 3F, G, H)." Vague phrases that are open to interpretation (such as "small structures") are not helpful to most biologists, and depending on the extent of the aberration(s) may be risky advice.

Below I give a few specific examples to support my general assessments. While this is not a complete list, I hope these examples will clarify my general opinion and aid the authors in future attempts to publish this work.

Examples:

"One has to collect images with a sampling density high enough in the x-y and z dimension to get the most accurate PSF Gaussian fitting." - Most biologists 1) usually do not think in terms of sampling when collecting images, and may not understand what is meant by "collecting images with a sampling density high enough" and 2) don't know that the PSF central maxima can be approximated by a Gaussian, making it unclear why Gaussian fitting is used.

"We showed that when using detector arrays, the sensor size influenced the observed U values." - 'Detector arrays' is not a common phrase understood by biologists, and the inconsistent use of detector vs sensor is potentially confusing. Here the authors are using 'detector' to refer to the photodiode and 'sensor' to refer to the chip. Elsewhere, detector is used to refer to the chip. The authors use different terms for the same concept/part throughout the review (e.g., Poisson noise and shot noise). Switching between terms can be particularly confusing to non-experts, especially when the terms are not defined.

The authors use the phrasing "short, mid, and long time scale" to define lengths of experiments, an important parameter in determining whether a detected problem may cause problems for a particular type of experiment (e.g., stage drift). In my first review I asked the authors to define these time scales. In the revised versions they "added examples like "time-lapse experiments" for mid-term or "z-stack acquisitions" for short-term." These examples are not useful. One may need to collect a seconds-minutes timescale timelapse using camera streaming to image fast dynamics, or a very large z-stack to image a cleared organ. The authors should explain how to determine if the error may be a problem for a given experiment, so that the researchers can decide for themselves if the error may affect their conclusions.

"Ideally, all pixels should have values very close to each other, considering the intrinsic Poisson noise of sensors." - Poisson noise is inherent to the stochastic and quantum nature of light, it is not an intrinsic property of sensors.

In their letter, the authors state (in response to my comment that they should recommend collecting multiple images) that "In our case we focus on characterizing the field illumination and a single image is sufficient." - I would argue that making any measurement once is not good practice, regardless of whether the measurement is made for QC purposes or for image processing/corrections.

"Coma aberrations can also be present, although their influence on the measured x-y FWHM stays low." - Coma causes elongation of the lateral PSF. Other aberrations, such as field curvature and astigmatism, are not mentioned.

"Hence, whenever homogeneity illumination is affected in the case of a wide sensor chip, besides choosing better corrected objectives, no corrective action can be undertaken for the microscope input, apart from cropping the image to a central area of the sensor or using digital shading correction." - I understand what the authors are getting at here, but in the case of a widefield system adjusting the collector lens or liquid light guide position can make a big difference.

"It is important to note that camera manufacturers apply protocols to correct homogeneity of the chip through the DSNU (column or pixel correction for sCMOS), average hot pixels or adjust the linearization of the sensor." The new reference provided is from 1995. Camera technology has dramatically changed since then, and sCMOS detectors were introduced to the biological imaging community much later.

With regards to measuring stage drift: "A fluorescent bead is convenient, but bright field observation of various features, like

dust, is possible as well" - This is not true if the dust is on the camera faceplate.

"WF techniques, bulb-type sources like metal-halide, mercury or xenon arc lamps are not very homogeneous and additional optical elements like diffusers are often necessary" - The new reference provided is a paper from the Manley lab in which they build a unique system (FIFI) for decreasing illumination inhomogeneity in a custom-built localization microscopy system using laser illumination. I'm not sure how that citation is appropriate here.

"The results showed not unexpectedly that most of the objectives are better corrected for the G-R combinations, while the combinations with the blue channel (DAPI) are more often associated with poor co-registration. This ensures that the tested microscopes were most adapted for imaging at that range of the visible spectrum, often used for live-cell imaging." The blue light forming the DAPI image is visible. DAPI can be excited with UV light below the visible spectrum, but the excitation wavelength does not effect co-registration (i.e., you would see the same misregistration when imaging a fluorescent bead and a luminescent bead of the same wavelengths).

"Spatial and time-resolved quantification of a labelled cell component involves fluorescence quantification. Such emission quantification relies on the fluorochrome photophysical properties and is linearly related to the dye concentration and the intrinsic power of the light source used under conventional excitation." - Until ground state depletion is reached, which is quite easy with the high intensity laser of a point scanning confocal.

Finally, a somewhat off-topic comment with regards to the statement "The authors would like to warmly thank Reviewer 2 for his very pertinent remarks throughout this work." I would like to kindly suggest that the authors do not make assumptions with regards to the gender of the reviewers of their manuscripts.

Reviewer #3 (Comments to the Authors (Required)):

WOW! I am so incredibly impressed with the improvements to the manuscript and, most importantly, the manuscript as it stands.

This manuscript presents context, guidelines and protocols for quality control measurements of six categories of light microscope systems. It is based on an incredible amount of data collection and synthesis of the extensive literature. It is a timely and important contribution to help increase rigor and reproducibility in light microscopy experiments. Importantly, it is presented in a manner that will be approachable to a wide variety of microscope users.

The authors adequately addressed all my concerns and exceeded expectations.

One minor comment on Page 7: "comma aberrations" is in the text instead of "coma aberrations".

Well done!

Wendy Salmon
University of North Carolina at Chapel Hill

RESPONSES TO REVIEWERS

REVIEWER #1

This is a re-review of this manuscript and everything positive stated about this work still remains. The revision done was substantial and the manuscript is very much improved. The responses to the reviewers comments (and I acknowledge that there were a lot of comments) were reasonable and the changes made sufficient, but for two points: 1) access to the raw data and 2) comparability of performances.

Regarding 1) access to raw data: The limits of hosting the amount of data is acknowledged, but given that QC data should be metadata to research data and are crucial for a correct analysis this manuscript will be setting the tone for the community and it is important to create open access and transparency. For this reason I urge the authors to reconsider their position and curate their data to make the accessible. The likely strongest argument against making data open from the authors is that they feel they open "Pandora's box" as there will be limits in direct comparability between microscopes. The authors argue in their response that these technical limits made them focus on providing QC metrics over comparing performance. While that is a valid point, it is not one that should stand in the way of open data access compatible with FAIR norms. Especially since the authors are contradicting themselves on the issue, see point 2).

Regarding 2) comparability of performance: As laid out in 1) the authors have some concerns about comparative performances, but in the camera section change their position and explicitly state the comparability improvement by using photons over grey values. A short section at the end of the manuscript in the summary discussion would be helpful to outline the limits within the data sets as acquired and add recommendations on using QC data acquired with the presented protocols for microscope (performance) comparison.

The authors would like to thank Reviewer #1 for her/his comments on this revised version. The two points raised by Reviewer #1 are perfectly relevant and feasible. Here are our answers:

1) Access to raw data: The authors are convinced that the FAIR principle should be the norm for open data in Science. Also, it is of primary importance that researchers use QC data as image metadata. Thus, we decided to classify and make available most of the data used in that work (some of the data is some years old and lack of metadata, this kind of data was not taken into account). The actual size of data is around 360 GB and is on a database through a DOI public link: Faklaris, O., & Guilbert, T. (2022). *MetroloJ_QC_Public_CID_OpenData* [Data set]. DataCite. https://doi.org/10.57889/OD_METROLOJ_QC

2) Comparability of performance: We added a short section at the end of the manuscript.

“It should be noted that the datasets of that work were not acquired to compare the performance of different systems. This kind of comparison may be influenced by parameters difficult to standardize, like the microscope environment or the association of the different microscope components which are by nature different, even if they can have similar datasheets, and thus lead to different results. Our datasets and the resulting guidelines aim in characterizing the microscope's health state over time, using relative and not absolute values.”

Besides that there are still issues with clarity of the text. While overall there is a massive improvement of the text there are still significant challenges for a non-expert reader. To give some examples:

- Abstract page 1: "Reproducibility in light microscopy represents a goal to achieve to improve research results" likely means "Increasing rigor and reproducibility in light microscopy aims to improve research results..."

We modified the initial phrase of the abstract : "Although there is a need to demonstrate reproducibility in light microscopy acquisitions, the lack of standardized guidelines monitoring microscope health status over time has so far impaired the widespread use of quality control (QC) measurements."

- Out of several examples from the introduction, take a look at the second paragraph, the redundancy and lack of clarity. The argument is good, but it needs to be written in a straight forward and clear manner.

We tried to improve the clarity of the entire manuscript, avoid redundancies and make it more comprehensible to biologists. The introduction ("background" in the new version) and conclusion of each subsection are revised. We added examples of the impact of QC measurements to biological imaging at the subsections and the figures of stage drift (Fig. 7E), stage positioning repeatability (Fig. 8E) and camera noise characterization (Fig. 9F).

- Or page 3&4: "the microscope objective will work on its manufactured design"

We tried to remove these non straight-forward sentences.

- The "Shannon-Nyquist criterion" is evoked, but no reasoning is given to the why it applies.

The first time we present the criterion we give its definition and role in biological imaging. We add a sentence that makes it simpler for the biologist to understand the meaning.

"The spatial sampling rate is crucial for accurate PSF measurements. At least the Shannon-Nyquist criterion should be fulfilled to ensure that the PSF image is not deprecated by the lack of spatial sampling. This criterion defines that the pixel size should be equal to at least one half of the resolution of the optical system [33-35]."

- On page 9: "..., considering the intrinsic Poisson noise ..." From the flow of the argument in that section it seems that similar pixel values are to be expected for obvious reasons, the mentioning of the Poisson distribution for light detection would need to be explained. For the reader it is important that the authors say what they mean exactly and not add half-sentences that imply the need for more details without providing them.

We had the choice between defining these notions properly (Poisson noise, shot noise, etc.) or focusing on the notions necessary for the definition of the metrics. We chose to focus on the notions necessary for the QC metrics definition and in the revised version we now do not mention some notions, like the Poisson noise here.

- Page 10: "Previous studies investigated the field illumination ..." This is a lead argument for the beginning of the section! State what has been done and why you chose a distinct subset of metrics! Examples like this can be found throughout the text and are truly editorial in nature, but important for the audience of readers.

The authors preferred to give a short introduction - background at the beginning of each section, showing the importance of the studied parameter for biological imaging. After that short introduction, we focus on what has been studied so far, and give the novelty of our work. This is why we think the above phrase fits better in that part of the section and not at the very beginning.

Within the 7 subsections I would suggest to replace "Introduction" with "Background". There is a good introduction to the whole manuscript already.

We followed your proposition in the new revised version.

For power measurement of light sources the authors might want to mention the recently released protocol form QuaRep working group 1. It adds to this work.

We added the reference of the protocol published by the WG1 of QUAREP.

The literature overview provided by the authors is impressive and by itself a helpful resource for the community. However, the foundational work by Ted Young, Lucas van Vlies and Raimund Ober on camera calibration is missing, as is later work by Fang Huang, Bewersdorf and Lidke as well as the work by Rieger and Stallinga and work by the Ober lab on sampling requirements is missing and should be added.

We added some references of the above teams (Young team: Mullikin et al. 1994, for Huang, Bewersdorf and Lidke we already cite two of their papers Liu et al. 2018, Huang et al. 2014, for Rieger team we added Thorsen et al. 2018, Rieger and Stallinga 2014).

REVIEWER #2

In the revised version, the authors provided a major reorganization of the manuscript and added additional data. Unfortunately, it is my opinion that the work remains inappropriate for publication in JCB for several reasons:

The manuscript assumes an understanding of microscopy that, in my experience, is beyond the majority of cell biologists. The manuscript contains inaccurate statements and poor references. The writing is generally hard to read and the manuscript is very long with many figures. The effort to characterize so many microscopes is impressive, but I'm not convinced that the data is useful to a broad audience, and the measurements and methods they suggest are not novel. They do use a novel ImageJ plugin to analyze the data, but introduction and validation of the tool is not the focus of the manuscript.

While I appreciate the extensive work the authors put into the revision, I do not think they have adequately addressed the issues above. I recommend that the authors send this work to a specialized journal, such as Journal of Microscopy. To effectively reach a more general audience, the authors would need to dramatically increase their effort to define, explain and reference the background knowledge that is necessary to appreciate and implement their suggestions.

I also remain unconvinced that the suggested QC time intervals and acceptable limits are applicable to the wide range of imaging experiments performed by cell biologists. I think this audience would benefit more from a thorough treatment of the problems that may be lurking in their microscope and how they should be addressed, in addition to how to test for them.

By publishing this work in JCB, the authors wish to raise the awareness of the importance of monitoring over time the QC measurement points that we propose among the community of biologists who use core-facilities and biologists who regularly use microscopes. We think that JCB is the perfect journal to start with.

This work is a founding part of what is emerging with the QUAREP-LiMi consortium, which is beginning to gain a lot of importance as evidenced by the number of participants (more than 400 members now) and the impressive number of recent publications (<https://quarep.org/resources/publications/>) and the ones to come (<https://quarep.org/nature-method-published-two-stories/>). The authors are convinced that the reproducibility of scientific measurements made with a light microscope is of primary importance and strongly believe that this work will contribute to this momentum generated by QUAREP-LiMi by questioning the community of biologists on their laboratory practices.

To improve the manuscript, the authors added biological examples in the figures and in the text. The general introductions and conclusions as well as those of the subsections now contain concrete examples of the importance of these measurements and of the sometimes dramatic impact that a deviation from the norm can have on their results.

As in the previous version, the authors offer time intervals and acceptable limits that are based on the data they collected from their microscopes and their general knowledge as core facility staff, but the rationale for the time intervals and acceptable limits is not discussed. They do say that "The recommended frequency depends on multiple parameters like the per year occupation time of the microscope, the number of its different users (major difference if the system is in a core facility with dozens of users per year or in a research team, used by the same people the whole year), the human

resources that take care of the microscope, and the kind of biological experiments carried out." In my opinion, this statement argues against attempts to define general time intervals and acceptable limits for QC. When some rationale is given, it is most often inadequate (e.g., "From the experimental results shown in Figure 3B, we propose to set the tolerance value at a ratio of 1.5 between measured and theoretical FWHM. 69% of the lateral FWHM and 82% of the axial measurements are located inside the 1.5 ratio green square.").

Both measurement time intervals and acceptable limits are empiric values and recommendations. We often explained in the manuscript for which kind of applications the limit values are crucial. In the new revised version we emphasize more the importance of each measured parameter and its acceptable limit values in biology by giving at the end of each 'chapter' a short paragraph that describes exactly the importance in biological imaging. We also added examples of biological imaging in the 'chapters' of stage drift, stage positioning repeatability and camera noise in the effort to really make the measured parameters and limit values understandable by biologists. Regarding the measuring time interval, we added this part in the first revised version of the manuscript after the advice of the third reviewer. We think that it is useful information for the reader. For both acceptable limits and time intervals our aim is to raise awareness in the biologists community who perform imaging and provide useful guidelines.

I also think that it is critical to this type of work to not only provide a method of assessing a problem, but also - whenever possible - solutions. The authors list troubleshooting suggestions in Table 1, but many of them will not be helpful to novice microscopists (e.g., "beam size at the back focal plane of the objective" and "cosmic rays") and there are no references provided.

We agree that a troubleshooting list offering solutions was missing as we were focused on the origin of the problem. That is why we fusionned the last column of Table 1 including now both the 'reason of the out of limit values' and the 'proposed action', giving a long and useful to our opinion list of troubleshooting for all the chapters of the manuscript.

They also make statements such as "The user of the microscope can nevertheless continue monitoring the PSF quality of the objective while keeping in mind that imaging of small structures or applying super-resolution techniques should be avoided (Fig 3F, G, H)." Vague phrases that are open to interpretation (such as "small structures") are not helpful to most biologists, and depending on the extent of the aberration(s) may be risky advice.

In the new revised version we give a short paragraph at the end of each 'chapter' which presents the importance of the metric and the limit values in biological imaging. We give more specific examples (examples added in the chapter of stage and camera characterization) and in general try to be more precise. We also modified the phrase 'imaging of small structures' to become more precise.

Below I give a few specific examples to support my general assessments. While this is not a complete list, I hope these examples will clarify my general opinion and aid the authors in future attempts to publish this work.

Examples:

"One has to collect images with a sampling density high enough in the x-y and z dimension to get the most accurate PSF Gaussian fitting." - Most biologists 1) usually do not think in terms of sampling when collecting images, and may not understand what is meant by "collecting images with a sampling density high enough" and 2) don't know that the PSF central maxima can be approximated by a Gaussian, making it unclear why Gaussian fitting is used.

1) we need to give to biologists recommendations and thinking of the sampling rate when imaging, that is why we refer to it here. We simplified the phrase with the following: "Another important consideration for PSF evaluation is the sampling density (i.e. pixel size). One has to collect images with a sampling density high enough (at least more than two times the resolution according to the Shannon-Nyquist criterion) in the x-y and z dimension to get the most accurate PSF Gaussian fitting. We consider that PSF can be sufficiently fitted by a Gaussian function (ref. Zhang et al.)"

2) we specified that a Gaussian fit is sufficient for calculating FWHM of PSF and we give now a reference (Zhang et al., Gaussian approximations of fluorescence microscope point-spread function models, Applied Optics, 2007)

"We showed that when using detector arrays, the sensor size influenced the observed U values." - 'Detector arrays' is not a common phrase understood by biologists, and the inconsistent use of detector vs sensor is potentially confusing. Here the authors are using 'detector' to refer to the photodiode and 'sensor' to refer to the chip. Elsewhere, detector is used to refer to the chip. The authors use different terms for the same concept/part throughout the review (e.g., Poisson noise and shot noise). Switching between terms can be particularly confusing to non-experts, especially when the terms are not defined.

To gain in clarity, we homogenized the text in the new revised version by using the term "detectors" to refer to photodiodes as well as all the family of the matrix and point detectors. The term sensor is now used to refer to the camera sensor chip. The notion of noise is now simplified by removing Poisson and shot noise terms. We added some new references for a better understanding, if the reader wishes more details.

The authors use the phrasing "short, mid, and long time scale" to define lengths of experiments, an important parameter in determining whether a detected problem may cause problems for a particular type of experiment (e.g., stage drift). In my first review I asked the authors to define these time scales. In the revised versions they "added examples like "time-lapse experiments" for mid-term or "z-stack acquisitions" for short-term." These examples are not useful. One may need to collect a seconds-minutes timescale timelapse using camera streaming to image fast dynamics, or a very large z-stack to image a cleared organ. The authors should explain how to determine if the error may be a problem for a given experiment, so that the researchers can decide for themselves if the error may affect their conclusions.

We added examples in the first revised version for the different time scales as you proposed, and we believe that they are pertinent. Of course a z stack in a cleared sample is longer than a typical z stack of a cellular monolayer, but with confocal , spinning disk techniques we do not (and often cannot) perform in routine this kind of imaging at high depths. Then the reader can classify her/his experiment in the appropriate time scale depending on the real time of the experiment, as we define for the short scale the 5min duration and the mid scale 2h. Regarding if the error is a problem to the experiment, we added a phrase at the end of the section, that sums up the importance of illumination power for quantitative biological imaging. If the error is a real problem for a given experiment, then the biologist is the only person who can decide on that, depending on the degree of quantification she/he desires to perform.

"Ideally, all pixels should have values very close to each other, considering the intrinsic Poisson noise of sensors." - Poisson noise is inherent to the stochastic and quantum nature of light, it is not an intrinsic property of sensors.

We modified this sentence to : "Ideally, all of the pixels in the image of a uniform sample would have the same grey level value across the field of view, considering the intrinsic noise of detectors [ref 9]."

In their letter, the authors state (in response to my comment that they should recommend collecting multiple images) that "In our case we focus on characterizing the field illumination and a single image is sufficient." - I would argue that making any measurement once is not good practice, regardless of whether the measurement is made for QC purposes or for image processing/corrections.

As we often explain in the text and for other sections (like the coregistration one) we do not aim to characterize the illumination field of a microscope for calibration reasons and post-processing corrections. We simply want to assess values and monitor them in time. From our experience of many years QC at our respective core facilities, a single image acquisition is enough for field illumination monitoring.

"Coma aberrations can also be present, although their influence on the measured x-y FWHM stays low." - Coma causes elongation of the lateral PSF. Other aberrations, such as field curvature and astigmatism, are not mentioned.

To our experience coma aberration induces a small elongation of the lateral PSF. It is more the 'banana' shape PSF of that aberration that is difficult to characterize with a metric and is not well represented by only measuring the xyz values. We added the astigmatism aberration and explained more the spherical aberrations in the Discussion section.

"Hence, whenever homogeneity illumination is affected in the case of a wide sensor chip, besides choosing better corrected objectives, no corrective action can be undertaken for the microscope input, apart from cropping the image to a central area of the sensor or using digital shading correction." - I understand what the authors are getting at here, but in the case of a widefield system adjusting the collector lens or liquid light guide position can make a big difference.

This is right. The text is modified accordingly.

"It is important to note that camera manufacturers apply protocols to correct homogeneity of the chip through the DSNU (column or pixel correction for sCMOS), average hot pixels or adjust the linearization of the sensor." The new reference provided is from 1995. Camera technology has dramatically changed since then, and sCMOS detectors were introduced to the biological imaging community much later.

This more recent reference now replaces the older one for that point: <https://www.nature.com/articles/s41467-019-13841-8>

With regards to measuring stage drift: "A fluorescent bead is convenient, but bright field observation of various features, like dust, is possible as well" - This is not true if the dust is on the camera faceplate.

Indeed. We added a sentence to warn the potentially inattentive user.

"WF techniques, bulb-type sources like metal-halide, mercury or xenon arc lamps are not very homogeneous and additional optical elements like diffusers are often necessary" - The new reference provided is a paper from the Manley lab in which they build a unique system (FIFI) for decreasing illumination inhomogeneity in a custom-built localization microscopy system using laser illumination. I'm not sure how that citation is appropriate here.

This more global reference replaces the older one:
<https://opg.optica.org/oe/fulltext.cfm?uri=oe-28-15-22036>

"The results showed not unexpectedly that most of the objectives are better corrected for the G-R combinations, while the combinations with the blue channel (DAPI) are more often associated with poor co-registration. This ensures that the tested microscopes were most adapted for imaging at that range of the visible spectrum, often used for live-cell imaging." The blue light forming the DAPI image is visible. DAPI can be excited with UV light below the visible spectrum, but the excitation wavelength does not effect co-registration (i.e., you would see the same misregistration when imaging a fluorescent bead and a luminescent bead of the same wavelengths).

This is true. The confusion comes from the 'visible spectrum' formulation. To avoid any trouble, we originally specified 'at that range of the visible spectrum', as our data showed that green and red are better corrected than blue or even UV. Since that doesn't seem to be clear enough, we've added the ref 22.

"Spatial and time-resolved quantification of a labelled cell component involves fluorescence quantification. Such emission quantification relies on the fluorochrome photophysical properties and is linearly related to the dye concentration and the intrinsic power of the light source used under conventional excitation." - Until ground state depletion is reached, which is quite easy with the high intensity laser of a point scanning confocal.

The authors believe that "under conventional excitation" defines their point quite well. Moreover, the authors wish to globally simplify the text and they are not sure that the notion of ground state depletion introduced by Reviewer #2 will help to improve the general understanding of the work presented.

Finally, a somewhat off-topic comment with regards to the statement "The authors would like to warmly thank Reviewer 2 for his very pertinent remarks throughout this work." I would like to kindly suggest that the authors do not make assumptions with regards to the gender of the reviewers of their manuscripts.

We now take into account this comment and reply accordingly in the cover letter and replies to the comments.

August 12, 2022

RE: JCB Manuscript #202107093RR

Dr. Orestis Faklaris
CNRS
Biocampus
1919 route de Mende
Montpellier 34293
France

Dear Dr. Faklaris:

Thank you for submitting your revised manuscript entitled "Quality assessment in light microscopy for routine use through simple tools and robust metrics". We would be happy to publish your paper in JCB pending final revisions necessary to meet our formatting guidelines (see details below).

A. MANUSCRIPT ORGANIZATION AND FORMATTING:

- 1) Text limits: Character count for Articles and Tools is normally < 40,000, not including spaces. Count includes the abstract, introduction, results, discussion, and acknowledgments. Count does not include title page, materials and methods, figure legends, references, tables, or supplemental legends. Needless to say, the manuscript is well over this limit but, given the unusual nature of the paper (e.g. embedding methodology into each individual section, etc.), we can forego the limits in this case.
- 2) Figures limits: Articles and Tools may have up to 10 main text figures. You currently meet this limitation but bear it in mind when revising.
- 3) Figure formatting: Scale bars must be present on all microscopy images, including inset magnifications. Molecular weight or nucleic acid size markers must be included on all gel electrophoresis.
- 4) Statistical analysis: Error bars on graphic representations of numerical data must be clearly described in the figure legend. The number of independent data points (n) represented in a graph must be indicated in the legend. Statistical methods should be explained in full in the materials and methods. For figures presenting pooled data the statistical measure should be defined in the figure legends. Please also be sure to indicate the statistical tests used in each of your experiments (both in the figure legend itself and in a separate methods section) as well as the parameters of the test (for example, if you ran a t-test, please indicate if it was one- or two-sided, etc.).
****Also, since you used parametric tests in your study (e.g. t-tests, ANOVA, etc.), you should have first determined whether the data was normally distributed before selecting that test. In the stats section of the methods, please indicate how you tested for normality. If you did not test for normality, you must state something to the effect that "Data distribution was assumed to be normal but this was not formally tested."****
- 5) Please be sure to provide the sequences for all of your primers/oligos and RNAi constructs in the materials and methods. You must also indicate in the methods the source, species, and catalog numbers (where appropriate) for all of your antibodies.
- 6) Microscope image acquisition: The following information must be provided about the acquisition and processing of images:
 - a. Make and model of microscope
 - b. Type, magnification, and numerical aperture of the objective lenses
 - c. Temperature
 - d. imaging medium
 - e. Fluorochromes
 - f. Camera make and model
 - g. Acquisition software
 - h. Any software used for image processing subsequent to data acquisition. Please include details and types of operations involved (e.g., type of deconvolution, 3D reconstitutions, surface or volume rendering, gamma adjustments, etc.).

7) References: There is no limit to the number of references cited in a manuscript. References should be cited parenthetically in the text by author and year of publication. Abbreviate the names of journals according to PubMed.

8) Supplemental materials: There are normally strict limits on the allowable amount of supplemental data. Articles/Tools may have up to 5 supplemental figures. Obviously, the paper is well over this limit but, given the circumstances, we will be able to give you the extra space. However, please do not add to the current number of supplementary figures. Please also note that tables, like figures, should be provided as individual, editable files. A summary of all supplemental material (that is, in addition to the supplementary figure legends) should appear at the end of the Materials and methods section (or, in this unusual case, at the end of the main text but before the references). Please see any recent JCB paper for an example of what is meant by this.

9) eTOC summary: A ~40-50 word summary that describes the context and significance of the findings for a general readership should be included on the title page. The statement should be written in the present tense and refer to the work in the third person. It should begin with "First author name(s) et al..." to match our preferred style.

10) Conflict of interest statement: JCB requires inclusion of a statement in the acknowledgements regarding competing financial interests. If no competing financial interests exist, please include the following statement: "The authors declare no competing financial interests." If competing interests are declared, please follow your statement of these competing interests with the following statement: "The authors declare no further competing financial interests."

11) A separate author contribution section is required following the Acknowledgments in all research manuscripts. All authors should be mentioned and designated by their first and middle initials and full surnames. We encourage use of the CRediT nomenclature (<https://casrai.org/credit/>).

12) ORCID IDs: ORCID IDs are unique identifiers allowing researchers to create a record of their various scholarly contributions in a single place. At resubmission of your final files, please consider providing an ORCID ID for as many contributing authors as possible.

B. FINAL FILES:

Please contact the journal office with any questions, cellbio@rockefeller.edu.

Thank you for this interesting contribution, we look forward to publishing your paper in Journal of Cell Biology.

Sincerely,

Joerg Bewersdorf, PhD
Monitoring Editor
Journal of Cell Biology

Tim Spencer, PhD
Executive Editor
Journal of Cell Biology